# DF40: Toward Next-Generation Deepfake Detection

**Zhiyuan Yan**[1], **Taiping Yao**[2][†], **Shen Chen**[2], **Yandan Zhao**[2], **Xinghe Fu**[2], **Junwei Zhu**[2], **Donghao Luo**[2], **Chengjie Wang**[2], **Shouhong Ding**[2], **Yunsheng Wu**[2], **Li Yuan**[1][†]

[1]School of Electronic and Computer Engineering, Peking University
[2]Tencent Youtu Lab
{zhiyuanyan@stu.,yuanli-ece@}pku.edu.cn

## Abstract

We propose a new comprehensive benchmark to revolutionize the current deepfake detection field to the next generation. Predominantly, existing works identify top-notch detection algorithms and models by adhering to the common practice: training detectors on one specific dataset (*e.g.*, FF++ [62]) and testing them on other prevalent deepfake datasets. This protocol is often regarded as a "golden compass" for navigating SoTA detectors. But can these stand-out "winners" be truly applied to tackle the myriad of realistic and diverse deepfakes lurking in the real world? If not, what underlying factors contribute to this gap? In this work, we found the **dataset** (both train and test) can be the "primary culprit" due to the following: (1) *forgery diversity*: Deepfake techniques are commonly referred to as both face forgery (face-swapping and face-reenactment) and entire face synthesis (especially face). Most existing datasets only contain partial types of them, with limited forgery methods implemented (*e.g.*, 2 swapping and 2 reenactment methods in FF++); (2) *forgery realism*: The dominated training dataset, FF++, contains out-of-date forgery techniques from the past four years. "Honing skills" on these forgeries makes it difficult to guarantee effective detection generalization toward nowadays' SoTA deepfakes; (3) *evaluation protocol*: Most detection works perform evaluations on one type, *e.g.*, training and testing on face-swapping types only, which hinders the development of universal deepfake detectors.

To address this dilemma, we construct a highly diverse and large-scale deepfake detection dataset called **DF40**, which comprises **40** distinct deepfake techniques (10 times larger than FF++). We then conduct comprehensive evaluations using **4** standard evaluation protocols and **8** representative detection methods, resulting in over **2,000** evaluations. Through these evaluations, we provide an extensive analysis from various perspectives, leading to **7** new insightful findings contributing to the field. We also open up **4** valuable yet previously underexplored research questions to inspire future works. We release our dataset, code, and checkpoints at https://github.com/YZY-stack/DF40.

## 1 Introduction

We are currently in 2024, an explosive era of Artificial Intelligence Generated Content (AIGC), a world where you can effortlessly make anyone say anything at any time, a realm where reality and fiction blend seamlessly, creating a mesmerizing tapestry of digital artistry and potential deception.

---

[†] Corresponding Author

38th Conference on Neural Information Processing Systems (NeurIPS 2024) Track on Datasets and Benchmarks.

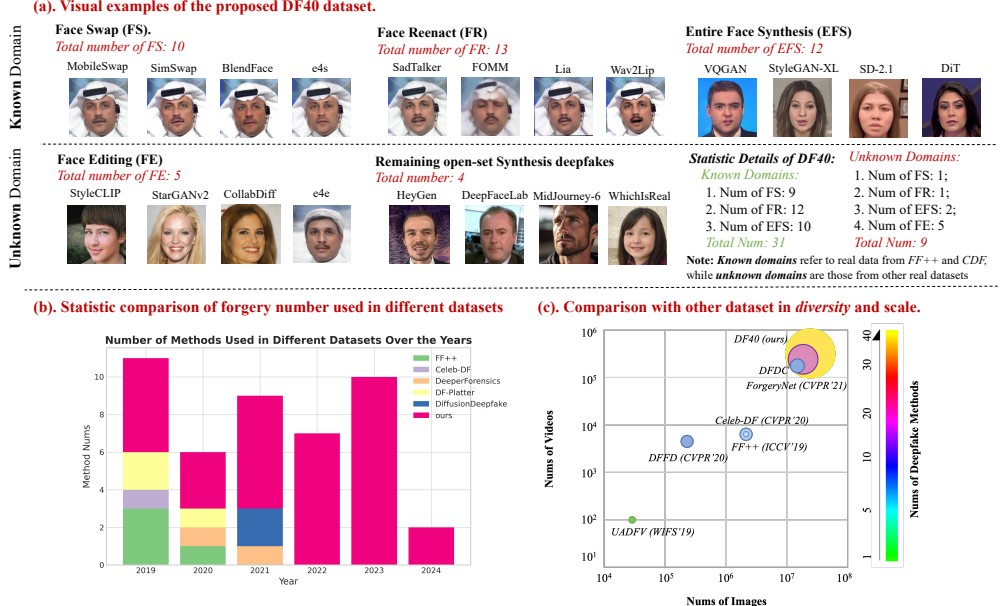

Figure 1: Overview of our DF40 dataset. DF40 shows advantages in data diversity, synthesis quality, and deepfake realism. Note *all* the above figures are deepfake, which does not exist in the real world.

Amidst this AIGC revolution, the ease of generating *deepfakes*[1] has become a "double-edged sword." Unfortunately, deepfake is often misused for manipulating one person's identity (face-swapping) or controlling facial expressions and movements in a portrait (face-reenactment). It can be particularly harmful as it may lead to severe digital crimes and undermine social trust. Therefore, there is an urgent need to develop a reliable system for detecting deepfakes.

Unfortunately, most existing detection methods can only handle a subset of deepfake types [65; 87]. To illustrate, in face-swapping deepfake detection, blending-based detectors [65; 43] have achieved SoTA results on existing face-swapping datasets (*e.g.*, CDF [42]). But all these methods heavily rely on the assumption that existing face-swapping should share a blending step[2]. However, recent advancements in face-swapping (*e.g.*, Simswap [9], FaceDancer [61]) directly generate all content (including the background) without blending. So "looking for" the specific blending artifacts for detecting the "non-blending" forgeries such as Simswap could be unrealistic. Regrettably, this crucial observation may not be well-known in the field, as most existing face-swapping deepfake datasets [62; 17; 42] involve blending in their fake data. Similarly, many existing face-reenactment generators (*e.g.*, LIA [82]) and all entire-image-synthesis generators (*e.g.*, DDPM [24]) are also without any blending. Hence, developing detectors on these blend-only datasets might limit the model's potential ability to detect a broad range of deepfake types. In a real-world scenario with unpredictability and complexity, creating such a "unified" detector is essential. However, this issue has persistently been challenging, as most existing datasets either contain limited types or are specific to forgery types (see Tab. 1).

Therefore, we realize that a *diverse* and *comprehensive* dataset is the true key to "unlock" the potential of ideal deepfake detection. To this end, we "jump" from the previous protocol settings (*e.g.*, train and test solely on face-swapping fakes) and propose a new comprehensive dataset called **DF40**. Our main contribution can be generally summarized as two folds: **(1)** Our DF40 implements **40** different deepfake techniques, including face-swapping, face-reenactment, entire face synthesis, and face editing. We even include the just-released DiT [54], PixArt-$\alpha$ [8], and highly popular software DeepFaceLab [14] and HeyGen [23] to simulate real-world deepfakes; **(2)** We introduce 4 standard protocols for evaluations and show 8 insightful findings to the field. We also open up new research questions and topics to inspire future research toward next-generation deepfake detection.

---

[1]Deepfake is a term derived from "deep" (the use of deep nets) and "fake" (does not exist). Our scope mainly focuses on face deepfakes but shows the promising in detecting other non-face deepfakes such as arts.

[2]Blending the altered (deepfake-generated) face into an existing background image.

Table 1: Comparison of existing/previous deepfake datasets. DF40 surpasses any other dataset in diversity, scale, and modeling. Deepfake data in DF40 is created by **40** deepfake techniques, including **10** FS methods, **13** FR methods, **12** EFS methods, and **5** FE methods.

| Dataset | Publication | Latest Fake | Methods | FS | FR | EFS | FE | Fake Videos | Fake Images | Pretraining |
|---|---|---|---|---|---|---|---|---|---|---|
| DF-TIMIT [37] | ArXiv'18 | faceswap-GAN [64] (2018) | 2 | 2 | - | - | - | 640 | - | - |
| UADFV [91] | ICASSP'19 | Unknown | 1 | 1 | - | - | - | 49 | 252 | - |
| FaceForensics++ [62] | ICCV'19 | NeuralTextures [71] | 4 | 2 | 2 | - | - | 4K | - | - |
| DeepFakeDetection [17] | None | Unknown | 5 | 5 | - | - | - | 3068 | - | - |
| CDF [42] | CVPR'20 | Unknown | 1 | 1 | - | - | - | 5,639 | - | - |
| DFFD [13] | CVPR'20 | StyleGAN [34] (2018) | 7 | 7 | - | - | - | 3000 | 0.2M+ | - |
| DeeperForensics-1.0 [30] | CVPR'20 | DF-VAE [30] (2020) | 1 | 1 | - | - | - | 10K | - | - |
| DFDC [18] | ArXiv'20 | StyleGAN [34] (2018) | 8 | 5 | 1 | 2 | - | 0.1M+ | - | - |
| ForgeryNet [22] | CVPR'21 | StarGANv2 [11] (2020) | 15 | 6 | 4 | 2 | 3 | 0.1M+ | 1M+ | ✓ |
| FakeAVCeleb [36] | NeurIPS'21 | Wav2Lip [55] (2021) | 4 | 2 | 2 | - | - | 9.5K | - | - |
| KoDF [38] | ICCV'21 | Wav2Lip [55] (2021) | 6 | 3 | 3 | - | - | 0.1M+ | - | - |
| FFIW [94] | CVPR'21 | FSGAN [52] (2019) | 3 | 3 | - | - | - | 10K | - | - |
| DF3 [32] | TMM'22 | StyleGAN3 [33] (2021) | 6 | - | - | 6 | - | 15k+ | - | - |
| DeepFakeFace [69] | ArXiv'23 | Stable-Diffusion [78] (2021) | 3 | 1 | - | 2 | - | - | 90K | - |
| DF-Platter [51] | CVPR'23 | FaceShifter [39] (2020) | 3 | 3 | - | - | - | 0.1M+ | - | - |
| DiffusionDeepfake [4] | ArXiv'24 | Stable-Diffusion [78] (2021) | 2 | - | - | 2 | - | - | 0.1M+ | - |
| **DF40 (Ours)** | - | PixArt-$\alpha$ [8] (2024) | **40** | **10** | **13** | **12** | **5** | **0.1M+** | **1M+** | ✓ |

i) **Latest Fake**: The year when the *latest* deepfake method was added to the dataset, *e.g.*, the latest method in ours is PixArt-$\alpha$ [8] (2024).

ii) **Methods**: The *number* of different deepfake methods used to generate fakes in the dataset. The methods are generally classified into: face-swapping (*FS*), face-reenactment (*FR*), entire face synthesis (*EFS*), and face editing (*FE*).

iii) **Fake Videos** and **Fake Images**: The number of *fake* videos and images involved in the dataset.

iv) **Pretraining**: Indicates if pretraining was conducted and pretraining weights were released in the dataset.

## 2 Background

### 2.1 Deepfake Generation.

Based on previous survey [50], deepfake techniques can be typically classified into four types: face-swapping (*FS*), face-reenactment (*FR*), entire face synthesis (*EFS*), and face editing (*FE*). **(1) Face-swapping:** This paper classifies the face-swapping technique into two domains: *DF-family* and *FS-family*. *(i) DF-family* involves creating a mask around the facial region (some even include the neck [46]) and blending the generated deepfake face back into the background image using that mask. Most existing and famous face forgery datasets, such as FF-DF [62], CDF [42] and DFDC [16], belong to this line. *(ii) FS-family* methods represent another significant category in face-swapping deepfake generation. These methods typically involve the use of an identity-background encoder. It disentangles a face image's identity and background information during encoding. Notably, these methods directly generate all content, even the background. Many recent face-swapping research works [85; 9] are within this line. **(2) Face-reenactment:** Generally, this technique can be used to modify source faces, imitating the actions or expressions of another face. Differing from face-swapping, face-reenactment techniques are *rarely* considered in existing datasets. Two commonly used reenactment-based forgeries are Face2Face [72] and NeuralTextures [71]. These two forgeries are implemented in the FF++ dataset. Due to the amazingly rapid development of the AIGC technologies (*e.g.*, Digital Human), these relatively old-fashioned methods cannot represent the modern' SoTA reenactment methods. Our DF40 implements 13 face-reenactment methods in total, including the classical animation [67; 68], SoTA's audio-based driven methods [92; 55], image-based driven methods [82; 26; 5]. We also include the well-known best face generation technique, HeyGen [23], in our dataset for evaluation. **(3) Entire Face Synthesis:** This technique can be generally treated as "Face AIGC." With the rapid development of AIGCs, this technique has achieved remarkably notable improvement. The two widely used technologies to generate synthesis faces are GAN (*e.g.*, VQGAN [20]) and Diffusion models (*e.g.*, StableDiffusion [59]). **(4) Face Editing:** This technique aims to modify the facial attributes (*e.g.*, age and gender) of the given face images. Most of these works utilize the latent code of StyleGAN [35] to perform editing during GAN inversion.

### 2.2 Existing Deepfake Datasets.

Most earlier public deepfake datasets published before 2021 [37; 91; 17; 42; 62; 13] generate deepfakes using only the face-swapping technique that involves a blending process. Also, these datasets contain only single or no more than 4 specific manipulation approaches. Notably, DFDC [16] implements 7 deepfake approaches and extends the data scale to the million-level. After that, ForgeryNet [22] applies 15 methods to create forgery data with a million-level data scale. However, with the advent of AIGCs such as diffusion-based generators, producing realistic content has become increasingly popular and widely seen on social media. Consequently, whether a detector trained on older forgery methods can generalize to today's SoTA deepfakes is uncertain. In response, many recent works [32; 69] have begun to focus on current generative models. A very recent

Table 2: Involved/Implemented deepfake methods of the proposed DF40 dataset. We use the following 40 distinct methods to create deepfake videos and images for evaluation.

| Type | ID-Number | Method | Sub-Types | Venue | Real Data Source | Data Format | Data Used | Data Scale | Code Link |
|---|---|---|---|---|---|---|---|---|---|
| Face-swapping (FS) | 1 | FSGAN [52] | Parsing mask | ArXiv 2019 | FF++ & CDF | Video | Train & Test | 1500+ | Hyper-link |
| | 2 | FaceSwap [1] | Graphic based | None | FF++ & CDF | Video | Train & Test | 1500+ | Hyper-link |
| | 3 | SimSwap [9] | Disentangle | ICCV 2019 | FF++ & CDF | Video | Train & Test | 1500+ | Hyper-link |
| | 4 | InSwapper [28] | Used in Roop [60] | None | FF++ & CDF | Video | Train & Test | 1500+ | Hyper-link |
| | 5 | BlendFace [66] | Disentangle | ICCV 2023 | FF++ & CDF | Video | Train & Test | 1500+ | Hyper-link |
| | 6 | UniFace [85] | Disentangle | ECCV 2022 | FF++ & CDF | Video | Train & Test | 1500+ | Hyper-link |
| | 7 | MobileSwap [86] | Lightweight | AAAI 2022 | FF++ & CDF | Video | Train & Test | 1500+ | Hyper-link |
| | 8 | e4s [46] | Disentangle | CVPR 2023 | FF++ & CDF | Video | Train & Test | 1500+ | Hyper-link |
| | 9 | FaceDancer [61] | Disentangle | WACV 2023 | FF++ & CDF | Video | Train & Test | 1500+ | Hyper-link |
| | 10 | DeepFaceLab [14] | High quality | PR 2023 | UADFV [91] | Video | Test Only | 100 | Hyper-link |
| Face-reenactment (FR) | 11 | FOMM [67] | Image Driven | NeurIPS 2019 | FF++ & CDF | Video | Train & Test | 1,500+ | Hyper-link |
| | 12 | FS_vid2vid [79] | Landmark Driven | ArXiv 2019 | FF++ & CDF | Video | Train & Test | 1,500+ | Hyper-link |
| | 13 | Wav2Lip [55] | Audio Driven | MM 2020 | FF++ & CDF | Video | Train & Test | 1,500+ | Hyper-link |
| | 14 | MRAA [68] | Image Driven | CVPR 2021 | FF++ & CDF | Video | Train & Test | 1,500+ | Hyper-link |
| | 15 | OneShot [80] | Image Driven | CVPR 2021 | FF++ & CDF | Video | Train & Test | 1,500+ | Hyper-link |
| | 16 | PIRender [58] | Image Driven | ICCV 2021 | FF++ & CDF | Video | Train & Test | 1,500+ | Hyper-link |
| | 17 | TPSMM [93] | Image Driven | CVPR 2022 | FF++ & CDF | Video | Train & Test | 1,500+ | Hyper-link |
| | 18 | LIA [82] | Image Driven | ICLR 2022 | FF++ & CDF | Video | Train & Test | 1,500+ | Hyper-link |
| | 19 | DaGAN [26] | Image Driven | CVPR 2022 | FF++ & CDF | Video | Train & Test | 1,500+ | Hyper-link |
| | 20 | SadTalker [92] | Audio Driven | CVPR 2023 | FF++ & CDF | Video | Train & Test | 1,500+ | Hyper-link |
| | 21 | MCNet [25] | Image Driven | ICCV 2023 | FF++ & CDF | Video | Train & Test | 1,500+ | Hyper-link |
| | 22 | HyperReenact [5] | Image Driven | ICCV 2023 | FF++ & CDF | Video | Train & Test | 1,500+ | Hyper-link |
| | 23 | HeyGen [23] | Text Driven | None | VFHQ [84] | Video | Test Only | 50 | Hyper-link |
| Entire Face Synthesis (EFS) | 24 | VQGAN [20] | GAN based | CVPR 2021 | FF++ & CDF | Image | Train & Test | 48,000+ | Hyper-link |
| | 25 | StyleGAN2 [35] | GAN based | ArXiv 2019 | FF++ & CDF | Image | Train & Test | 48,000+ | Hyper-link |
| | 26 | StyleGAN3 [33] | GAN based | NeurIPS 2021 | FF++ & CDF | Image | Train & Test | 48,000+ | Hyper-link |
| | 27 | StyleGAN-XL [63] | GAN based | SIGGRAPH 2022 | FF++ & CDF | Image | Train & Test | 48,000+ | Hyper-link |
| | 28 | SD-2.1 [59] | GAN based | CVPR 2022 | FF++ & CDF | Image | Train & Test | 48,000+ | Hyper-link |
| | 29 | DDPM [24] | Latent Diffusion | NeurIPS 2020 | FF++ & CDF | Image | Train & Test | 48,000+ | Hyper-link |
| | 30 | RDDM [45] | Latent Diffusion | ArXiv 2023 | FF++ & CDF | Image | Train & Test | 48,000+ | Hyper-link |
| | 31 | PixArt-α [8] | Latent Diffusion | ICLR 2024 | FF++ & CDF | Image | Train & Test | 48,000+ | Hyper-link |
| | 32 | DiT-XL/2 [54] | Latent Diffusion | ICCV 2023 | FF++ & CDF | Image | Train & Test | 48,000+ | Hyper-link |
| | 33 | SiT-XL/2 [3] | Latent Diffusion | ArXiv 2024 | FF++ & CDF | Image | Train & Test | 48,000+ | Hyper-link |
| | 34 | MidJounery6 [49] | Popular Application | None | FFHQ [34] | Image | Test Only | 1,000 | Hyper-link |
| | 35 | WhichisReal [83] | GAN based | None | FFHQ [34] | Image | Test Only | 1,000 | Hyper-link |
| Face Editing (FE) | 36 | CollabDiff [27] | Diffusion based | CVPR 2023 | CelebA [47] | Image | Test Only | 48,000+ | Hyper-link |
| | 37 | e4e [73] | StyleGAN based | SIGGRAPH 2021 | FF++ & CDF | Image | Train & Test | 48,000+ | Hyper-link |
| | 38 | StarGAN [10] | StarGAN based | CVPR 2018 | CelebA [47] | Image | Test Only | 2,000 | Hyper-link |
| | 39 | StarGANv2 [11] | StarGAN based | CVPR 2020 | CelebA [47] | Image | Test Only | 2,000 | Hyper-link |
| | 40 | StyleCLIP [53] | StyleGAN based | ICCV 2021 | CelebA [47] | Image | Test Only | 2,000 | Hyper-link |

i) **Note-1**: The nine "test only" data is utilized as the **unknown** domain data, simulating the domain shift of real people distributions.
ii) **Note-2**: For HeyGen and DeepFaceLab, these two SoTA deepfakes are widely used and highly realistic. However, since they belong to *one-to-one* type [50], their implementation requires significantly more time and resources. Thus, we only create 50 and 100 for each.

study, DiffusionDeepfake [4], proposes a diffusion-specific dataset containing two mainstream SoTA diffusion generators: Stable-Diffusion [59] and MidJourney [49]. Furthermore, to detect these AIGC-generated images (not limited to face), GenImage [95] introduces a million-scale dataset created by 8 different generative models. However, both [4] and [95] are within only the entire image synthesis scope and may not guarantee successful detection of other deepfakes, *e.g.*, face-swapping. A more comprehensive comparison of existing datasets can be referred to Tab. 1.

## 3 DF40 Benchmark

**Research Scope and Summary of DF40.**   In this work, our scope specifically focuses on *face* deepfake detection, including face-swapping (FS), face-reenactment (FR), entire face synthesis (EFS), and face editing (FE). Natural image synthesis (such as art) is not within our scope, but we show the potential of using model training on face deepfake to detect these AIGC-generated images. DF40 dataset provides 40 deepfake approaches with 4 different types: FS, FR, EFS, and FE. For FS and FR, we provide video format data (over 0.1M video clips in total), and image format for EFS and FE (over 1M+ images in total). We also introduce several latest/popular generation techniques (*e.g.*, PixArt-$\alpha$ [8] in 2024) and online software (*e.g.*, HeyGen [23], DeepFaceLab [14; 29]). Furthermore, some classical and representative methods are also included, *e.g.*, FOMM [67]. Our dataset shows the advantages in both data diversity and scale (see Tab. 1).

**Original Data Collection and Deepfake Data Generation.**   **(1) Real Data:** We consider two mainstream and popular deepfake datasets: FF++ [62] (c23 version) and CDF [42] as our original data. The rationale behind this selection is that most previous research [40; 65; 90] follows the evaluation protocol of training on FF++ and testing on CDF. By utilizing these datasets for training and evaluation, we can adhere to previous works and verify if their conclusions and methods still hold in the context of our new dataset. We also provide real data from other existing datasets (*e.g.*, CelebA [47]) to facilitate the unknown domain evaluations (see Tab. 2 for details). We show more details of these original face datasets in the Appendix. **(2) Fake Data:** To guarantee the *diversity* of deepfake approaches in the proposed DF40, we introduce and implement **40** distinct deepfake techniques, which are listed in Tab. 2. A detailed description of each deepfake method is provided in the Appendix. Formally, we denote $x_t(i_t, a_t, b_t)$ as the *target* subject to be manipulated, which possesses attributes ($i$: person identity, $a$: identity-agnostic content, $b$:

Table 3: Same data domain, different forgery types (**Protocol-1**): **Cross-forgery evaluation** of different models on the FF domain. FS (FF) denotes all FS data within the FF domain, with FR (FF), FS (CDF), *etc*, being similar. The results of the within-forgery evaluations are in gray (as the validation set only).

| Training Set | Model | Testing Set (FF) | | | |
| --- | --- | --- | --- | --- | --- |
| | | FS (FF) | FR (FF) | EFS (FF) | Avg. (FF) |
| FS (FF) | Xception [12] | 0.991 | 0.892 | 0.810 | 0.898 |
| | CLIP [57] | 0.996 | 0.908 | 0.837 | 0.914 |
| | SRM [48] | 0.988 | 0.867 | 0.703 | 0.853 |
| | SPSL [44] | 0.987 | 0.849 | 0.735 | 0.857 |
| | RECCE [6] | 0.991 | 0.855 | 0.758 | 0.868 |
| | RFM [75] | 0.992 | 0.884 | 0.821 | 0.899 |
| FR (FF) | Xception [12] | 0.838 | 0.996 | 0.670 | 0.835 |
| | CLIP [57] | 0.932 | 0.999 | 0.798 | 0.910 |
| | SRM [48] | 0.893 | 0.998 | 0.698 | 0.863 |
| | SPSL [44] | 0.901 | 0.998 | 0.695 | 0.865 |
| | RECCE [6] | 0.865 | 0.997 | 0.716 | 0.859 |
| | RFM [75] | 0.892 | 0.999 | 0.776 | 0.889 |
| EFS (FF) | Xception [12] | 0.665 | 0.807 | 0.999 | 0.824 |
| | CLIP [57] | 0.688 | 0.889 | 0.999 | 0.859 |
| | SRM [48] | 0.596 | 0.776 | 0.999 | 0.790 |
| | SPSL [44] | 0.659 | 0.811 | 0.999 | 0.823 |
| | RECCE [6] | 0.691 | 0.801 | 0.999 | 0.830 |
| | RFM [75] | 0.653 | 0.795 | 0.999 | 0.816 |
| BI (FF) | SBI [65] | 0.810 | 0.714 | 0.678 | 0.734 |

Table 4: Same forgery types, different data domains (**Protocol-2**): **Cross-domain evaluation** of different models on CDF domain. FS (FF) denotes all FS data within the FF domain, with FR (FF), FS (CDF), *etc*, having similar meanings. FS (FF) and FS (CDF) represent the same forgery methods used but created on different data domains.

| Training Set | Model | Testing Set (CDF) | | | |
| --- | --- | --- | --- | --- | --- |
| | | FS (CDF) | FR (CDF) | EFS (CDF) | Avg. (CDF) |
| FS (FF) | Xception [12] | 0.922 | 0.657 | 0.642 | 0.740 |
| | CLIP [57] | 0.967 | 0.744 | 0.730 | 0.814 |
| | SRM [48] | 0.919 | 0.621 | 0.603 | 0.714 |
| | SPSL [44] | 0.938 | 0.656 | 0.648 | 0.747 |
| | RECCE [6] | 0.926 | 0.632 | 0.610 | 0.723 |
| | RFM [75] | 0.939 | 0.637 | 0.628 | 0.735 |
| FR (FF) | Xception [12] | 0.481 | 0.857 | 0.369 | 0.569 |
| | CLIP [57] | 0.638 | 0.933 | 0.209 | 0.593 |
| | SRM [48] | 0.454 | 0.869 | 0.326 | 0.550 |
| | SPSL [44] | 0.479 | 0.852 | 0.256 | 0.529 |
| | RECCE [6] | 0.452 | 0.881 | 0.332 | 0.555 |
| | RFM [75] | 0.492 | 0.882 | 0.359 | 0.578 |
| EFS (FF) | Xception [12] | 0.586 | 0.594 | 0.983 | 0.721 |
| | CLIP [57] | 0.617 | 0.735 | 0.988 | 0.780 |
| | SRM [48] | 0.589 | 0.620 | 0.964 | 0.724 |
| | SPSL [44] | 0.635 | 0.651 | 0.975 | 0.754 |
| | RECCE [6] | 0.623 | 0.603 | 0.984 | 0.737 |
| | RFM [75] | 0.644 | 0.666 | 0.981 | 0.764 |
| BI (FF) | SBI [65] | 0.679 | 0.609 | 0.723 | 0.670 |

external attributes) that can uniquely determine itself, while the *source* $x_s(i_s, a_s, b_s)$ is regarded as the *conditional media* ($c$), driving the *target* to change either identity or attributes or even both. We use $f_{sw}, f_{re}, f_{sy}, f_{ed}$ to represent *FS*, *FR*, *EFS*, and *FE*, respectively. Fig. 1 provides some visual examples for each category in DF40, and Fig. 14 generally illustrates them from the method perspective. *(i) Face-Swapping (FS):* From Fig. 14(a), FS aims to replace the content of $x_t$ with that of $x_s$ preserving the identity $i_s$; *(ii) Face-Reenactment (FR):* From Fig. 14(b), FR on $x_t(i_t, a_t, b_t)$ preserves its identity $i_t$ but has its *intrinsic* attributes $a_t$, *e.g.*, pose, mouth and expression manipulated by a driven variable $c_a$ and forms $\tilde{x}_t(i_t, \tilde{a}_s, b_t)$; *(iii) Entire Face Synthesis (EFS):* EFS generates a entirely synthesis face $\tilde{x}_t(\tilde{i}_t, \tilde{a}_t, \tilde{b}_t)$. We fine-tune the gen-

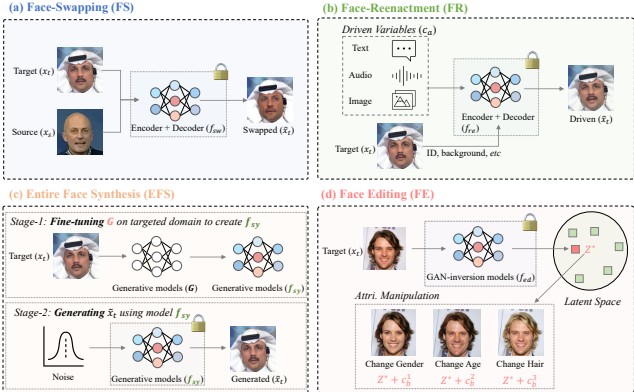

Figure 2: The general fake data generation pipeline of the proposed *DF40* dataset.

erative model $G$ using the real data from FF++ and CDF, and then obtain a face-synthesis model $f_{sy}$. Then, we can synthesize new synthesis faces from noise $n$. The general generation process is in Fig. 14(c); *(iv) Face Editing (FE):* From Fig. 14(d), FE on $x_t(i_t, a_t, b_t)$ has its *external* attributes $b_t$ altered, such as facial hair and age, controlled by conditional source $c_b$, to obtain $\tilde{x}_t(i_t, a_t, \tilde{b}_s)$.

## 4 Evaluations and Analysis

### 4.1 Experimental Setup

All pre-processing and training codebases in this work adhere to *DeepfakeBench* [90] to align the *standardized* settings. The details of dataset configuration, algorithm implementation, and full training can be seen in the Appendix. For detection selection, we choose the classical baseline Xception [12] with four SoTA detectors (SRM [48], SPSL [44], RECCE [6], RFM [75]) that use Xception as the backbone. In this manner, we aim to revisit the extra improvement over the baseline under our setting. We also consider the blending-based detector that uses the blending image (BI) for training, which creates pseudo-fakes by image blending, without using the actual deepfake data. We use SBI [65] to implement the blending-based detector. Note that the original SBI paper only uses BI for training, so we do not re-train it using our DF40 in evaluations.

Table 5: Different forgery types, different data domains (**Protocol-3**): toward real-world open-set evaluation of different models on the unknown domain. FS (FF) denotes all FS data within the FF domain, with FR (FF) and EFS (FF) being similar. Here, we use ❶ to mark the FS method, ❷ for the FR, ❸ for the EFS, and ❹ for the FE.

| Training Set | Model | Testing Set | | | | | | | | | |
|---|---|---|---|---|---|---|---|---|---|---|---|
| | | DeepFaceLab (❶) | HeyGen (❷) | MidJourney-6 (❸) | Whichisreal (❸) | StarGAN (❹) | StarGAN2 (❹) | StyleCLIP (❹) | e4e (❹) | CollabDiff (❹) | Avg. |
| FS (FF) | Xception [12] | 0.882 | 0.394 | 0.384 | 0.535 | 0.577 | 0.616 | 0.426 | 0.553 | 0.546 | 0.546 |
| | CLIP [57] | 0.930 | 0.539 | 0.540 | 0.439 | 0.896 | 0.746 | 0.730 | 0.738 | 0.674 | 0.692 |
| | SRM [48] | 0.866 | 0.473 | 0.298 | 0.538 | 0.606 | 0.617 | 0.572 | 0.410 | 0.699 | 0.564 |
| | SPSL [44] | 0.930 | 0.370 | 0.414 | 0.557 | 0.559 | 0.590 | 0.536 | 0.574 | 0.584 | 0.565 |
| | RECCE [6] | 0.899 | 0.537 | 0.293 | 0.509 | 0.580 | 0.599 | 0.399 | 0.520 | 0.492 | 0.536 |
| | RFM [75] | 0.918 | 0.719 | 0.286 | 0.496 | 0.652 | 0.570 | 0.705 | 0.689 | 0.798 | 0.648 |
| FR (FF) | Xception [12] | 0.705 | 0.473 | 0.459 | 0.323 | 0.492 | 0.456 | 0.006 | 0.175 | 0.050 | 0.349 |
| | CLIP [57] | 0.845 | 0.614 | 0.632 | 0.466 | 0.762 | 0.436 | 0.298 | 0.631 | 0.611 | 0.588 |
| | SRM [48] | 0.786 | 0.604 | 0.510 | 0.357 | 0.473 | 0.434 | 0.044 | 0.428 | 0.080 | 0.413 |
| | SPSL [44] | 0.704 | 0.543 | 0.446 | 0.272 | 0.348 | 0.423 | 0.002 | 0.585 | 0.060 | 0.376 |
| | RECCE [6] | 0.724 | 0.576 | 0.314 | 0.278 | 0.529 | 0.374 | 0.005 | 0.177 | 0.060 | 0.337 |
| | RFM [75] | 0.739 | 0.588 | 0.511 | 0.325 | 0.407 | 0.423 | 0.009 | 0.201 | 0.030 | 0.360 |
| EFS (FF) | Xception [12] | 0.497 | 0.325 | 0.472 | 0.772 | 0.777 | 0.677 | 0.984 | 0.611 | 0.997 | 0.679 |
| | CLIP [57] | 0.745 | 0.506 | 0.534 | 0.828 | 0.946 | 0.823 | 0.929 | 0.923 | 0.983 | 0.802 |
| | SRM [48] | 0.527 | 0.358 | 0.338 | 0.794 | 0.769 | 0.703 | 0.982 | 0.509 | 0.997 | 0.664 |
| | SPSL [44] | 0.641 | 0.383 | 0.427 | 0.694 | 0.699 | 0.723 | 0.922 | 0.602 | 0.967 | 0.673 |
| | RECCE [6] | 0.583 | 0.505 | 0.442 | 0.753 | 0.769 | 0.724 | 0.964 | 0.643 | 0.979 | 0.707 |
| | RFM [75] | 0.619 | 0.349 | 0.551 | 0.623 | 0.730 | 0.636 | 0.966 | 0.665 | 0.979 | 0.680 |
| BI (FF) | SBI [65] | 0.764 | 0.402 | 0.342 | 0.426 | 0.591 | 0.586 | 0.564 | 0.379 | 0.570 | 0.514 |

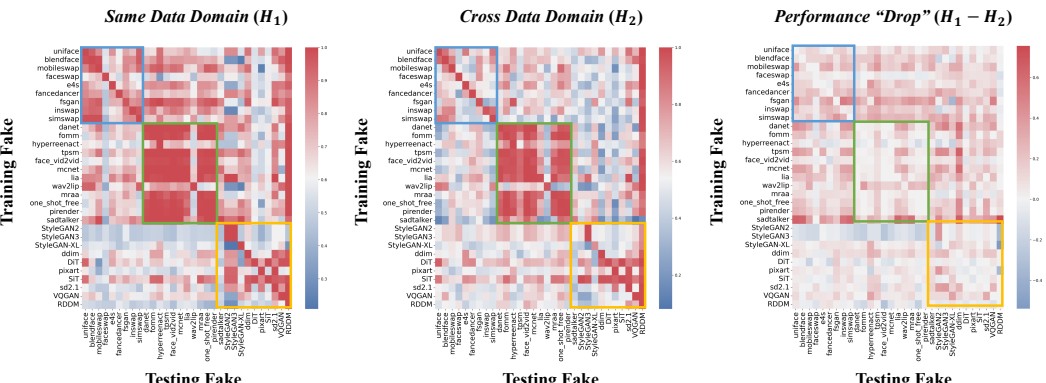

Figure 3: One-Verse-All (OvA) evaluation (**Protocol-4**): Training the baseline (*i.e.*, Xception) on one fake and testing it on other remaining fakes. We show the cross-forgery evaluations on both the FF++ domain and CDF domain. We also show the performance "drop" from the FF++ to the CDF. **Blue** donates all FS methods, **Green** for FR, and **Yellow** for EFS. In each heatmap, more "red" indicates higher values; "White" means 0.5 AUC (by chance), and "blue" indicates values below 0.5.

## 4.2 Evaluations, Findings, and Analysis

In this work, we conduct evaluations under four standard protocols: Cross-forgery evaluation (**Protocol-1**), Cross-domain evaluation (**Protocol-2**), Toward unknown forgery or domain evaluation (**Protocol-3**), and One-Verse-All (OvA) evaluation (**Protocol-4**). From these evaluations, we list 8 critical findings via empirical observations or in-depth analysis, as follows.

**Finding-1: Asymmetric performance drop among different forgery types (fake regions matter).** From Tab. 3, we observe that the performance drop between FS and FR is moderate but more significant between these two and EFS. Specifically, model training on FS yields higher results (around 0.8) on EFS, while training on EFS only achieves about 0.6 on FS. To understand the underlying reason for this finding, we provide the following detailed explanations: *One possible explanation is that FS can exhibit both localized and global forgeries, while EFS is restricted to global forgeries, making EFS less diverse in scope compared to FS.* As a result, training a model on EFS might not be sufficient to capture the localized fake artifacts present in FS. For example, some localized FS (such as DeepFakes of FF++ [62]) only contain fake artifacts within the facial region, but a model trained on EFS might be biased to "expect" artifacts across the entire image, including the background, resulting in a lower result on FS. To verify our claim, we consider using t-SNE for visualizing the latent distribution of 'real', 'whole-fake', and 'face-fake'. The t-SNE (Fig. 4) results show that a well-trained EFS detection model can easily distinguish the whole-fake from real images. However, the localized version, including face-fake and mouth-fake, appears much more similar to real images, making it more challenging to detect compared to the whole-fake. This experiment qualitatively verifies the significance of the fake region in detection.

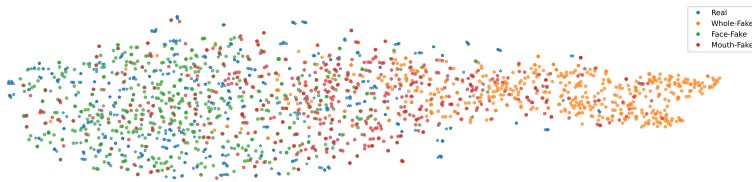

Figure 4: t-SNE visualization for real, whole-fake, face-fake, and mouth-fake images. The results show that a well-trained EFS detector (Xception) can effectively distinguish between whole-fake and real images, but struggles to identify fakes with only face or mouth manipulation. This observation highlights the significant influence of the manipulated region on detection performance.

**Finding-2: Existing well-designed (SoTA) detectors may not have obvious advantages over the baseline.** In our experiments, SRM, SPSL, RECCE, and RFM, all state-of-the-art deepfake detectors using Xception as their backbone, achieve results similar to the simple baseline Xception. This implies that these SoTA models might only learn sub-optimal forgery features and lack clear advantages over the baseline, although these SoTAs show higher results on previous datasets, $e.g.$, DFDC. In other words, although these well-designed SoTA detectors achieve higher scores than the Xception (baseline) when testing on CDF [42], they cannot maintain a consistent advantage over the baseline when testing on other forgeries. This highlights the remaining generalization issue.

**Finding-3: CLIP excels in deepfake detection than other baselines (such as Xception).** Outperforming other SoTA detectors and the Xception baseline across all scenarios, CLIP (both base and large versions) demonstrates the power of pre-training in deepfake detection. This highlights the benefits of pre-training before fine-tuning on the deepfake detection task. However, why can CLIP outperform the Xception by a notable margin? We analyze this point in the following content from the view of real face distribution. We analyze the learned features of the three models (CLIP-base, CLIP-large, Xception), as visualized in Fig. 7. **(1)** We see that both versions of CLIP learn more informative features that can gather some real samples into several different "groups" (see HeyGen and MidJourney), while the real features of Xception mix all of them together. Leveraging large-scale pre-training, CLIP can capture more informative facial features ($e.g.$, ID) about real faces and thus "know" that these features are *unrelated* to deepfakes, while Xception could be overfitted to ID, as evidenced by [19]. **(2)** Between the large and base versions, CLIP-large's real samples are closer to each other (see WhichisReal), suggesting it learns more comprehensive features of real faces. We also see that CLIP-large can *implicitly* "separate and align" different forgeries and data domains ("orthogonal" in WhichisReal), with no need for explicit constraints as [70].

**Finding-4: Forgery methods and data domains together contribute to discriminative forgery artifacts.** A significant performance drop is observed when both forgeries and domains change. For example, Xception training on FS (FF) achieves only 0.657 and 0.642 on FR (CDF) and EFS (CDF), respectively. These results are significantly lower (nearly a 20% drop) compared to results on FS (CDF) and FR (FF), which involve crossing only one of the domains or forgeries. This suggests that both factors together contribute to discriminative forgery artifacts for distinguishing real and fake.

To intuitively convey our conclusions, we draw a **causal graph** to illustrate the causal relationship of how domain (D) and forgery method (F) influence the model's generalization (R). The causal graph is presented in Fig. 5. Mathematically, the model's generalization result $R$ can be expressed as a function of these intermediate variables: $R = f(X_1, X_2, X_3)$, where $X_1 = g(D)$ represents the domain-specific influence, $X_3 = h(F)$ represents the forgery method-specific influence, and $X_2 = k(D, F)$ represents the combined influence of both domain and forgery method. Each of these variables $X_1$, $X_2$, and $X_3$ captures different aspects of how D and F interact to impact the overall performance $R$. The functions $g$, $h$, and $k$ are mappings that quantify the respective influences: $X_1 = g(D)$, $X_3 = h(F)$, and $X_2 = k(D, F)$. Finally, the result $R$ is determined by integrating these influences: $R = f(g(D), k(D, F), h(F))$. Fig. 6(a) supports this point, showing increased overlap between real and fake logits as domains and methods change incrementally. Also, crossing domains can result in "wrong confidence" for real predictions and vice versa.

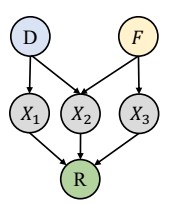

Figure 5: Causal graph.

**Finding-5: FR forgeries may share transferable patterns for detecting other FR instances.** Most FR forgery methods demonstrate high AUC within the heatmap, even across data domains. However, Wav2Lip, an audio-driven reenactment forgery that only modifies the mouth region, is an exception.

**(a). Xception**

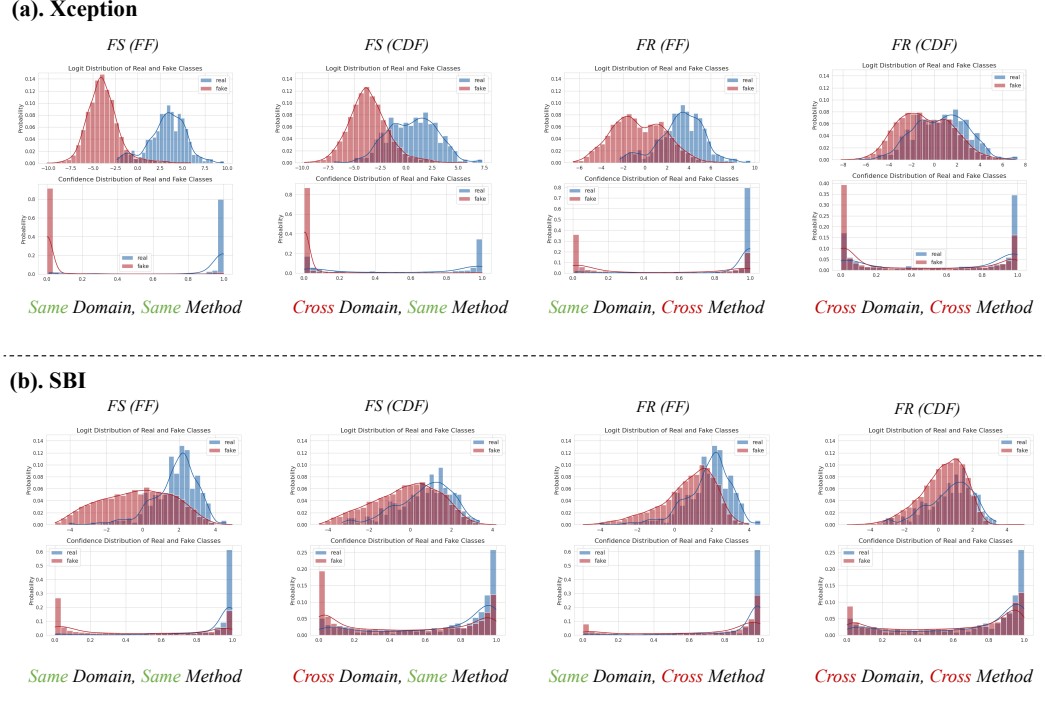

Figure 6: Logits and confidence analysis for Xception and SBI models.

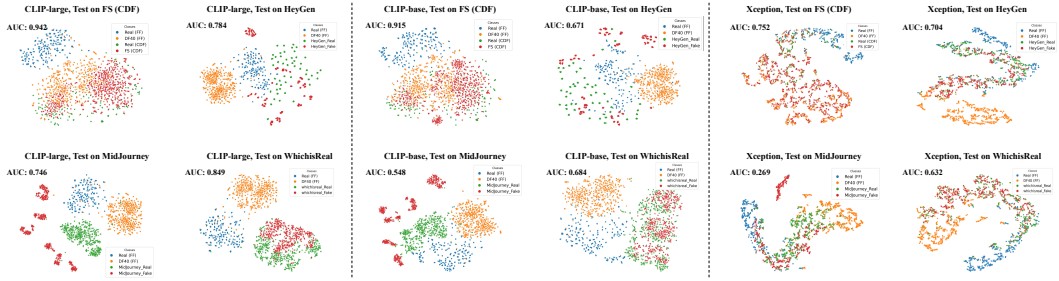

Figure 7: t-SNE Visualizations for three models: Xception, CLIP-base, and CLIP-large.

Its artifacts are more localized compared to other FRs that generate all content, making it different. Note that **Finding-5** does not mean that collecting (many) FR methods is without value. Specifically, even though these FR methods may share some transferable patterns to detect other FR methods, the fake similarity between FR and other types (like EFS) can also vary greatly depending on the specific FR method used (see Fig. 3).

**Finding-6: SBI could be viewed as an "anomaly detection" model.** SBI [65] (a blending-based detection) could be considered an "anomaly detection" model, which might utilize the real features for classifying fakes. Results in Fig. 6(a) show that when forgery samples appear significantly different from real samples, SBI classifies them as anomalies (forgery samples), likely because the pseudo-fake samples generated by SBI closely resemble their original real counterparts. In other words, SBI creates more realistic fake samples that are more similar to the real, encouraging the detection model to learn a more robust real representation than the baseline.

**Finding-7: CLIP-large shows the potential to generalize to some non-face deepfakes when trained only on face data.** In additional to evaluate on the face deepfakes, we also evaluate non-face-domain deepfakes (natural image synthesis) using the widely-used GenImag edataset to determine if models trained on face-domain data can transfer to non-face-domain detection. In Tab. 19, surprisingly, CLIP-large achieves a 0.746 AUC on previously unseen non-face deepfakes, while Xception only reaches a 0.535 AUC, near chance levels. This highlights that CLIP-large might learn some transferable (EFS) forgery features that are not related to the face content.

Table 6: Comparison of different models/baselines used for pre-training. FS (CDF) denotes all FS data within the CDF domain. DF40 (FF) is the combination of FS (FF), FR (FF), and EFS (FF).

| Training Set | Model | Testing Set | | | | | | | | | | | | |
|---|---|---|---|---|---|---|---|---|---|---|---|---|---|---|
| | | FS (CDF) | FR (CDF) | EFS (CDF) | DeepFaceLab | HeyGen | MidJourney-6 | Whichisreal | StarGAN | StarGAN2 | StyleCLIP | CollabDiff | e4e | Avg. |
| DF40 (FF) | Xception | 0.752 | 0.831 | 0.681 | 0.851 | 0.704 | 0.269 | 0.632 | 0.721 | 0.569 | 0.495 | 0.675 | 0.542 | 0.644 |
| | CLIP-base | 0.915 | 0.926 | 0.843 | 0.907 | 0.671 | 0.548 | 0.684 | 0.913 | 0.782 | 0.813 | 0.948 | 0.823 | 0.814 |
| | CLIP-large | **0.942** | **0.896** | **0.858** | **0.948** | **0.784** | **0.746** | **0.849** | **0.974** | **0.909** | **0.929** | **0.977** | **0.967** | **0.898** |

Table 7: Similar to Tab. 6, we evaluate on GenImage [95] (**non-face domain deepfakes**).

| Training Set | Model | Testing Set | | | | | | | | |
|---|---|---|---|---|---|---|---|---|---|---|
| | | ADM | BigGAN | GLide | MidJourney | SD-v4 | SD-v5 | Vqdm | Wukong | Avg. |
| DF40 (FF) | Xception | 0.723 | 0.529 | 0.514 | 0.558 | 0.490 | 0.494 | 0.469 | 0.505 | 0.535 |
| | CLIP-base | 0.940 | 0.850 | 0.666 | 0.447 | 0.494 | 0.494 | 0.682 | 0.542 | 0.639 |
| | CLIP-large | 0.911 | 0.967 | 0.736 | 0.571 | 0.630 | 0.614 | 0.882 | 0.660 | 0.746 |

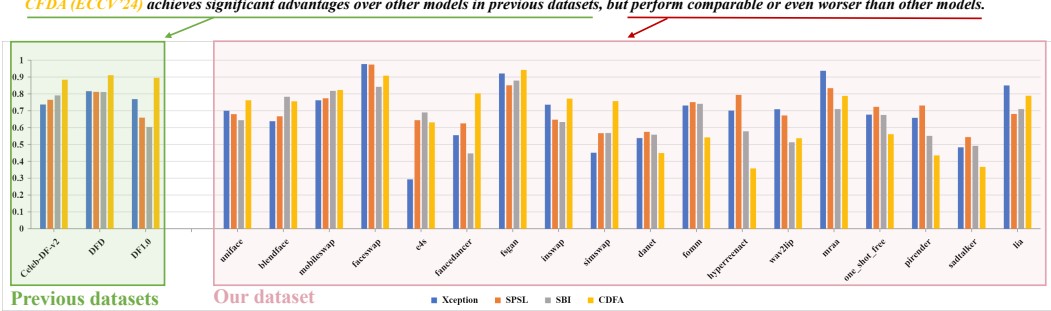

Figure 8: Evaluation of existing detectors on previous datasets and our DF40.

## 4.3 Discussion & Further Analysis

**Why A More Diverse Deepfake Dataset Is Crucial for Current Deepfake Detector?** In our work, we propose the DF40 dataset, with highly diverse deepfake approaches. Here, we aim to qualitatively verify why such a highly diverse dataset is extremely crucial for current deepfake detectors. The results in Fig. 8 show that CFDA [43] (latest SoTA) achieves significant advantages over other methods on existing datasets, but it performs comparably or even worse than other detectors. This emphasizes the necessity of constructing a dataset that encompasses a greater variety of deepfake methods and types. Without such a highly diverse dataset for evaluation, we would only be seeing "the tip of the iceberg." In other words, the current SoTA may not be the definitive "winner."

**Are Super-Resolution Images Fake?** Here, we explore whether a super-resolution (SR) image will be considered fake or real by the detection model. This question is important for two reasons: **(1)** many deepfake software programs involve an SR operation to enhance the resolution of generated faces (*e.g.,* FaceFusion[3]), and **(2)** Most existing SR methods [81] are based on deep generative models such as GAN. Therefore, the output of SR methods can also contain generative artifacts that might be similar to those in deepfake images. We use GFPGAN to perform SR on the self-reconstruction images (SRI), where we use SimSwap [9] to perform face-swap to the same ID. Results in Tab. 8 show that SR has a significant impact on fake images, particularly for EFS and FE methods. As demonstrated in the table, models trained on FS and EFS exhibit more than a 20% point improvement for the SRI after applying super-resolution. This suggests that SR operations may introduce noticeable generative artifacts that can be detected by models trained on EFS and FE.

**Exploration of Frequency Artifacts in Different Types of Deepfakes.** Here, we explore whether there exist common fake patterns in the frequency domain. following [77], we compute the average frequency spectra of high-pass filtered images. Results in Fig. 9 show that *although the deepfake types differ, they can show similar patterns/artifacts in the average spectra*. For instance, methods from FS (*i.e.,* SimSwap [9], BlendFace [66], InSwap) show similar "checkboard" patterns that can also be observed in EFS (*i.e.,* SD1.5 [59], VQGAN [20]) and FE (*i.e.,* e4e [73]). This observation highlights the need to further explore whether and why forgeries with different types can exhibit similar patterns in frequency.

---

[3]https://github.com/facefusion/facefusion/blob/master/facefusion/processors/
modules/face_enhancer.py

Table 8: Ablation study regarding the impact of **super-resolution** to the fake images. We use self-reconstruction images (SRI) as the fake and apply GFPGAN [81] to perform the super-resolution.

| Test / Train | FS | | FR | | EFS | | FE |
|---|---|---|---|---|---|---|---|
| | FSGAN | BlendFace | LIA | Wav2Lip | DiT | DDIM | e4e |
| SRI | 0.772 | 0.835 | 0.746 | 0.564 | 0.687 | 0.713 | 0.543 |
| SRI + Super-Resolution | 0.983 | 0.825 | 0.988 | 0.833 | 0.997 | 0.946 | 0.978 |

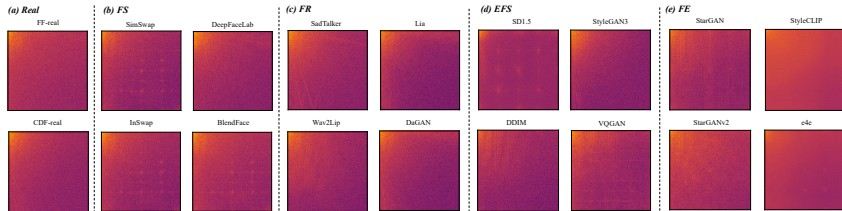

Figure 9: Frequency analysis on selected fake data within the proposed DF40.

**A Model Bias Toward the Resolution Gap Between Real and Fake.** In our previous experiments (see Tab. 5), we observe anti-intuitive phenomena: models training on FR achieve very low AUC (much lower than 0.5) on StyleCLIP. After investigating the underlying reason, we found that *the unaligned resolution between the real and fake classes introduces a model bias, that is, higher resolution, higher probability to be real.* Specifically, we observe that the training real data (FF++) obviously contains more high-frequency components (higher resolution) compared to the training fake data (FR). This noticeable resolution gap between real and fake data could be inevitably captured by the model [56], leading the model to potentially adopt a trivial solution: "Low resolution implies fake." In contrast, during testing, the fake data (StyleCLIP) contains significantly more high-frequency components (higher resolution) than the testing real data used for testing. We provide a frequency visualization as evidence to validate this claim (see our Appendix). This bias leads the model to make an inconsistent decision, resulting in the anomaly in AUC.

### 4.4 Open Questions and Potential Topics for Future Research

(1) **Regrading Blending:** Many existing detectors have utilized blending data to enhance the model's generalization. However, we have found that even the latest blending detector [43] is still limited in generalizing to all types of deepfakes (see Fig. 8). This indicates that blending might not be "all you need." But the question remains: *What is the role of blending data in training deepfake detectors? And what kinds of deepfakes can be addressed by blending?* (**Question-1**).

(2) **Regrading Forgery Diversity:** As the diversity of deepfakes increasingly goes up, **Question-2** raise: *how to design a new framework to learn (many) different forgeries effectively and jointly, without overfitting to limited specific fakes?*

(3) **Regrading Forgery Type:** Previous research [89; 88] classified deepfake types at the instance level. For example, if four deepfakes are applied for training, they will be regarded as four *distinct* deepfakes. However, there are so many forgery techniques available that it is hard to elaborate on them all. So, **Question-3:** *can we classify deepfakes based on FS, FR, EFS, and FE?*

(4) **Domain-Invariant Detector:** As shown in our previous discussion, we have found that many factors (such as *resolution* and *fake region*) can impede the model from learning domain-invariant features. Thus, **Question-4:** *How can we develop an invariant detector?*

## 5 Conclusions, Board Impacts, and Limitations

**(1) Conclusions:** We have developed *DF40*, a highly diverse and groundbreaking benchmark, comprising 40 distinct deepfake techniques to support the detection. Leveraging DF40, we conducted over 2,000+ evaluations using 8 representative detectors under 4 standard evaluation protocols, creating 7 new findings and 4 open questions for future works. We hope the proposed DF40 could revolutionize the whole field for the next generation. **(2) Board Impacts:** DF40 offers high-quality and realistic deepfake techniques, facilitating the detection of today's real-world deepfakes. Also, our benchmark assists in safeguarding societal trust and promoting the responsible use of such technology. **(3) Limitations:** One limitation is the lack of comprehensive analysis for *video-level detectors*. Actually, we provide evaluations using video models, *e.g.*, I3D [7] in the Appendix. However, we have not delved deeply into discussing and analyzing certain issues, *e.g.*, assessing and visualizing whether video models can effectively capture both temporal and spatial artifacts. In future work, we plan to broaden our benchmark's scope and address these concerns in detail.

## Acknowledgments and Disclosure of Funding

This work was supported in part by the Natural Science Foundation of China (No. 62202014, 62332002, 62425101), Shenzhen Basic Research Program (No.JCYJ20220813151736001).

We would like to express our profound gratitude to all the individuals (all collaborators) who have contributed to the successful completion of this work. Our heartfelt appreciation goes out to Yue Han and Chengming Xu from Tencent Youtu Lab, as well as Wentang Song from Shenzhen University, for their invaluable support in code implementation and reproduction. Furthermore, we extend our gratitude to Ke Sun from Xiamen University, Mingli Zhu from the Chinese University of Hong Kong, Shenzhen (CUHK-Shenzhen), Shaokui Wei from CUHK-Shenzhen, Mingda Zhang from CUHK-Shenzhen, and Yuhao Luo from CUHK-Shenzhen for their insightful discussions and contributions to this work. Our sincere thanks go to the code developers of *DeepfakeBench* for their easy-to-use codebase and unified training/testing framework. This includes Prof. Baoyuan Wu (project leader), Xinghang Yuan (previously in CUHK-Shenzhen), Yize Chen (CUHK-Shenzhen), Kangran Zhao (CUHK-Shenzhen), and Jikang Cheng (Wuhan University). Once again, we thank everyone involved for their dedication and efforts in making this big project successful.

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

# A    Appendix

## A.1    Content Structure in Appendix

We have organized additional important content in the Appendix due to the limited space. We present a brief outline of the content structure of the Appendix to facilitate readers to find the corresponding content, as follows:

- **Section A.2: Dataset Generation Methods & Original Data:**
    - Section A.2.1: Brief Introduction of Generation Methods;
    - Section A.2.2: Details of 40 Implemented Synthesis Methods;
    - Section A.2.3: Formulation of Manipulation Methods;
    - Section A.2.4: Rationale of Fine-tuning EFS Methods;
    - Section A.2.5: Details of Original Datasets;
- **Section A.3: Introduction of the Used Detection Methods:**
- **Section A.4: Experimental Setup & Full Results:**
    - Section A.4.1: Experimental Setup and Details;
    - Section A.4.2: Full Experimental Results and Further Discussion;
- **Section A.5: Further Analysis Results:**
    - Section A.5.1: More Results of Different Fake Regions;
    - Section A.5.2: Generalization to Non-face Domain AIGCs;
    - Section A.5.3: Generalization to External/Previous Deepfake Datasets;
    - Section A.5.4: Comparative Analysis of CLIP-base and CLIP-large for Deepfake Detection;
    - Section A.5.5: Analysis of "Anomalous Values" in Experiments;
    - Section A.5.6: Comparison of Video Model and Image Model;
    - Section A.5.7: Visualizations of the Real People Features;
    - Section A.5.8: Artifacts of deepfake forgeries in Frequency;
- **Section A.6 Additional information on Dataset Publication:**
    - Section A.6.1: Hosting Platform and Links;
    - Section A.6.2: Controlled Access;
    - Section A.6.3: Discrimination, Bias, and Fairness.

## A.2    Dataset Generation Methods & Original Data

### A.2.1    Brief Introduction of Generation Methods

Based on previous survey [50], deepfake techniques can be typically classified into four types: face-swapping (*FS*), face-reenactment (*FR*), entire face synthesis (*EFS*), and face editing (*FE*).

**(1) Face-swapping:** This paper classifies the face-swapping technique into two domains: *DF-family* and *FS-family*. *(i) DF-family* involves creating a mask around the facial region (some even include the neck [46]) and blending the generated deepfake face back into the background image using that mask. Most existing and famous face forgery datasets, such as FF-DF [62], Celeb-DF [42] and DFDC [16], belong to this line. *(ii) FS-family* methods represent another significant category in face-swapping deepfake generation. These methods typically involve the use of an identity-background encoder. It disentangles a face image's identity and background information during encoding. Notably, these methods directly generate all content, even the background. Much recent face-swapping research works [85; 9] and popular software (*e.g.*, Roop [60]) are within this line.

**(2) Face-reenactment:** Generally, this technique can be used to modify source faces, imitating the actions or expressions of another face. Differing from face-swapping, face-reenactment techniques are *rarely* considered in existing datasets. Two commonly used reenactment-based forgeries are Face2Face [72] and NeuralTextures [71]. These two forgeries are implemented in the FF++ dataset. Face2Face employs pairs of original and target faces, using key facial points to generate varied expressions, while NeuralTexture uses rendered images from a 3D face model to migrate expressions. Although these forgeries achieve more realistic visual synthetic results compared to their face-swapping counterparts in FF++, Due to the amazingly rapid development of the AIGC technologies (*e.g.*, Digital Human), these relatively old-fashioned methods cannot represent the modern' SoTA reenactment methods. Our DF40 implements 13 face-reenactment methods in total, including the classical animation [67; 68], SoTA's audio-based driven methods [92; 55], image-based driven methods [82; 26; 5]. We also include the well-known best face generation technique, HeyGen, in our dataset for evaluation.

**(3) Entire Face Synthesis:** This technique can be generally treated as "Face AIGC." With the rapid development of AIGCs, this technique has achieved remarkably notable improvement. The two widely used technologies to generate synthesis faces are GAN (*e.g.*, StarGAN [10]) and Diffusion models (*e.g.*, StableDiffusion [59]).

**(4) Face Editing:** This technique aims to modify the facial attributes (*e.g.*, age and gender) of the given face images. Most of these works utilize the latent code of StyleGAN [35] to perform editing during GAN inversion.

### A.2.2 Details of Implemented Forgery Methods in DF40

We used a total of **40** distinct deepfake generation/synthesis methods (see Tab. 2 of the manuscript for details). We will briefly explain each synthesis method below:

**1. FSGAN [52]** FSGAN is proposed by Nirkin et al. [52], which is the latest face-swapping method that has become popular recently. The key feature of this method is that it performs reenactment along with the face swap. First, it applies reenactment on the target video based on the source video's pose, angle, and expression by selecting multiple frames from the source having the most correspondence to the target video. Then, it transfers the missing parts and blends them with the target video. This process makes it much easier to train and does not take much time to generate face-swapped video. We use the code from the official FSGAN GitHub repository [52]. We used the best quality swapping model recommended by the authors of FSGAN to prepare our dataset by fine-tuning the input video pairs and generating better-quality results. We adopt this method because of its efficiency and better quality of the results. It is also used in the DFDC, ForgeryNet, FFIW, and FakeAVCeleb datasets. We use the official code from `https://github.com/NVlabs/imaginaire/` for implementation.

**2. FaceSwap [1]** FaceSwap is a pure landmark and graphics-based face-swapping method, not employing any neural network. The outcome is relatively poor and considered as the prototype of FS in the FF++ dataset. Despite its limitations, it still provides a basic understanding of face-swapping techniques. FaceSwap relies on traditional computer vision methods, such as facial landmark detection and image warping, to achieve face-swapping results. Although the quality of the results may not be as high as those produced by more advanced methods, FaceSwap remains an essential reference point in the development of face-swapping technology. Due to its popularity, it was used in FaceForensics++ datasets [62] to generate the face-swapped dataset. We use the official code from `https://github.com/MarekKowalski/FaceSwap/` for implementation.

**3. SimSwap [9]** SimSwap is one of the state-of-the-art (SOTA) methods for single-frame face swapping. It has not been applied to academic datasets yet. The facefusion software provides a good encapsulation of SimSwap, which can be directly called for use. SimSwap leverages deep learning techniques, such as attention mechanisms and GANs, to achieve high-quality face-swapping results in a single frame. Its ability to produce visually appealing and realistic face swaps has made it a popular choice for both research and practical applications. We use the official code from `https://github.com/neuralchen/SimSwap/` for implementation.

**4. InSwapper [28]** InSwapper is another SOTA method for single-frame face swapping that is also utilized in Roop [60]. Similar to SimSwap, it has not been widely applied to existing academic

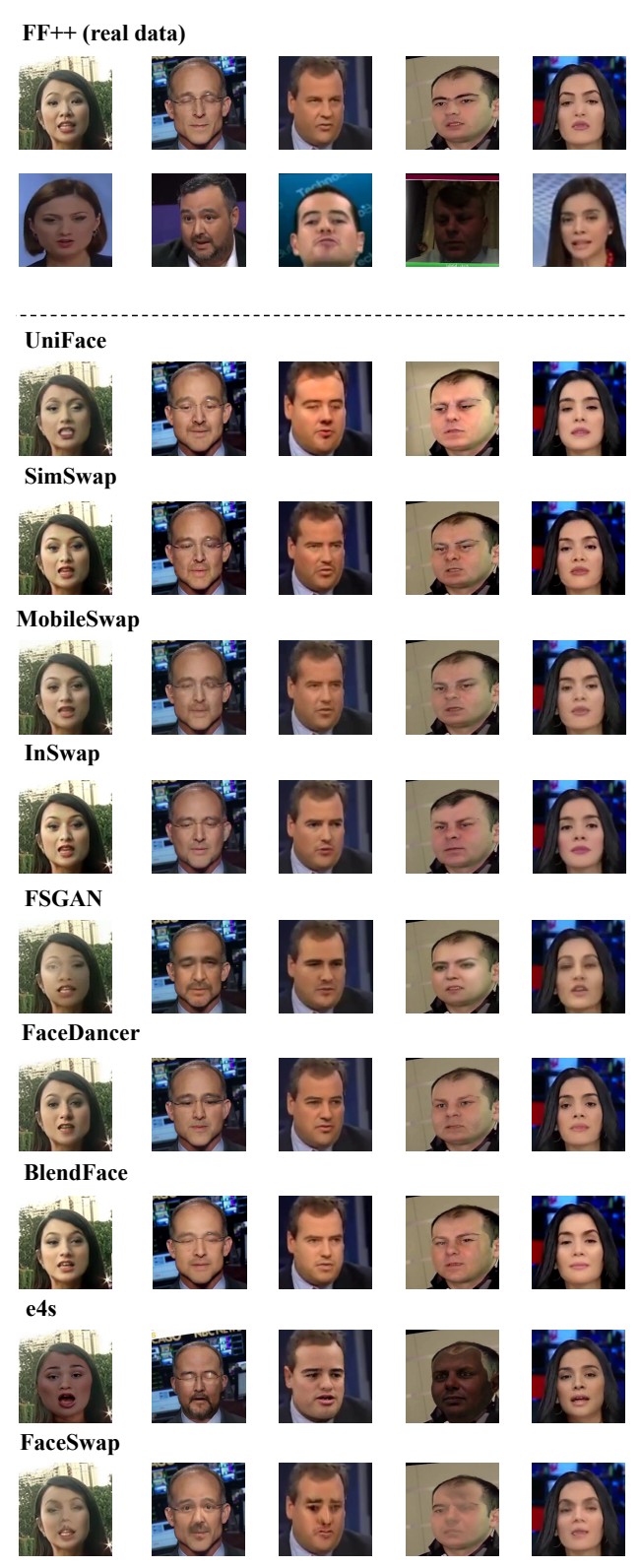

Figure 10: Illustration of visual examples for all FS (face-swapping) generated data on the FF domain.

deepfake datasets, but many applications in the real world due to its convenience and effectiveness. InSwapper can achieve good single-frame face swap results, but it performs average in terms of

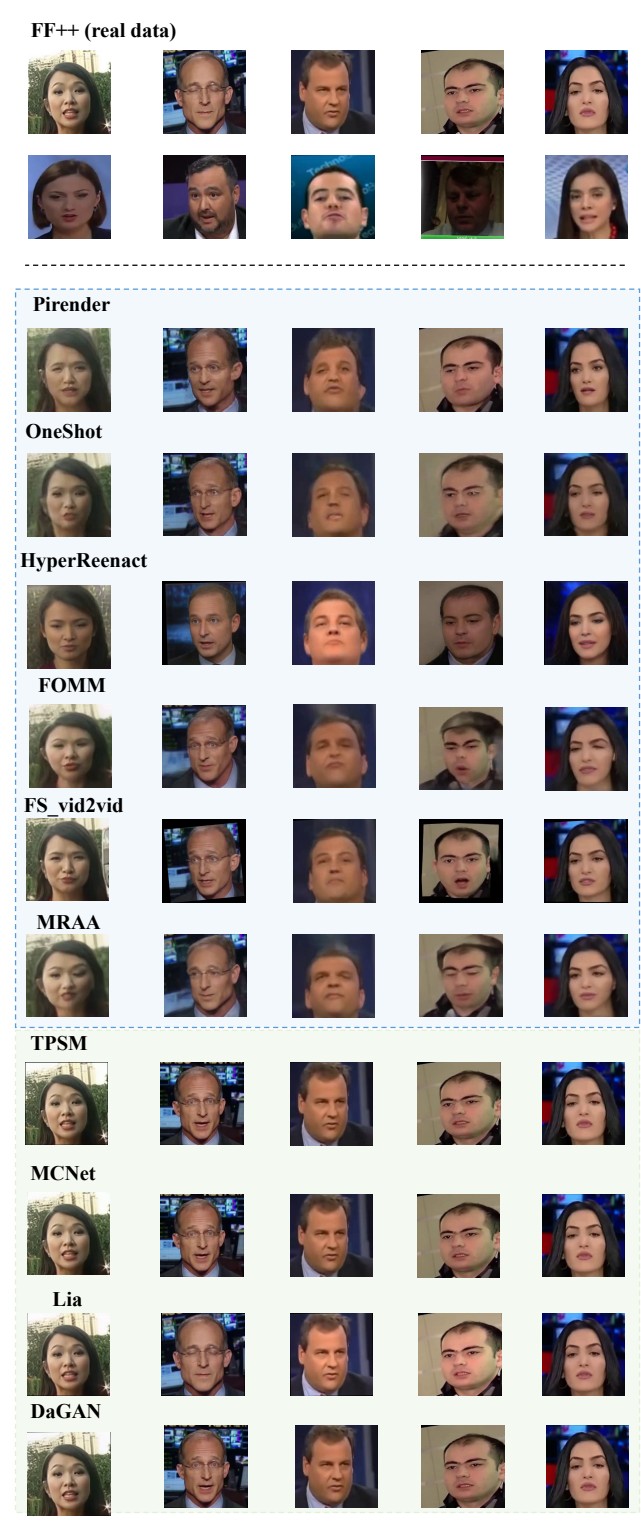

Figure 11: Illustration of visual examples for all FR (face-reenactment) generated data on the FF domain.

frame-to-frame consistency. InSwapper uses advanced deep learning techniques, including GANs and attention mechanisms, to generate realistic face swaps. Its main focus is on preserving the identity

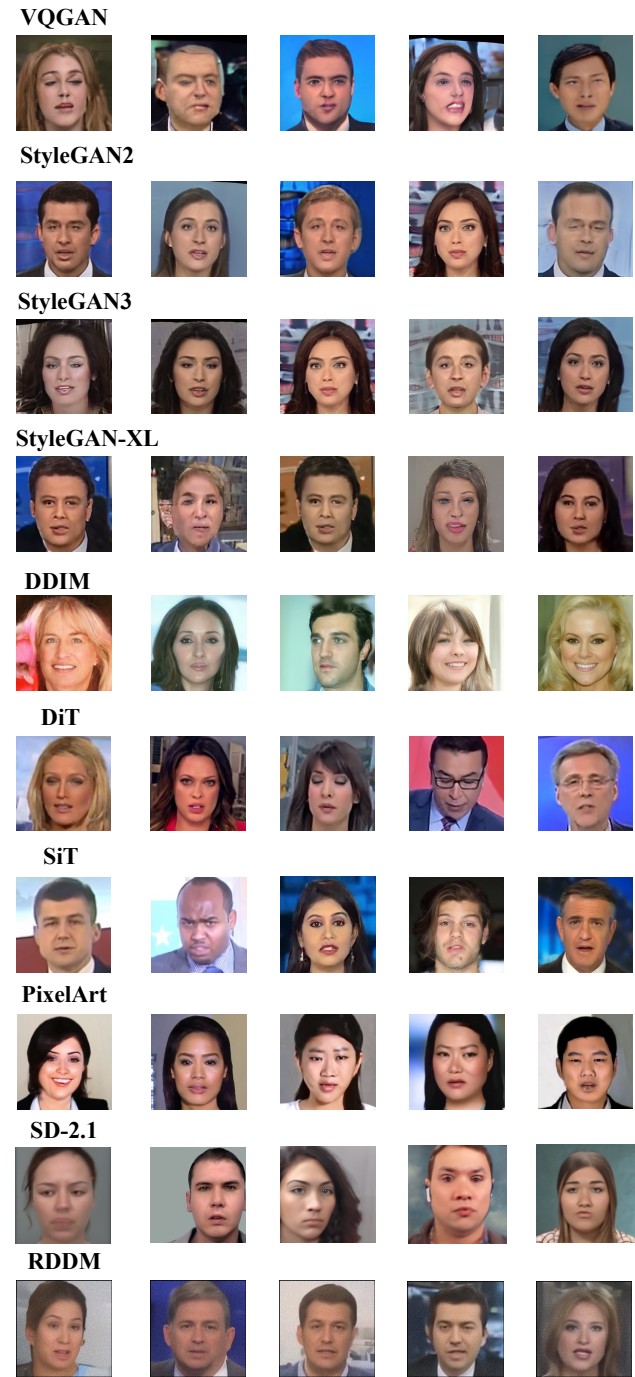

Figure 12: Illustration of visual examples for all EFS (entire face synthesis) generated data on the FF domain.

of the target face while seamlessly blending the source face's features. We use the popular code repository from https://github.com/haofanwang/inswapper/ for implementation.

**5. BlendFace [66]**   BlendFace is the latest SOTA face-swapping method that was published in ICCV 2023. It has a good performance in single-frame face swapping, but the continuous frames in the video tend to jitter noticeably. BlendFace employs advanced deep learning techniques, such as GANs, to generate high-quality face swaps. It also focuses on maintaining the target face's identity while

**DeepFaceLab (FS)**

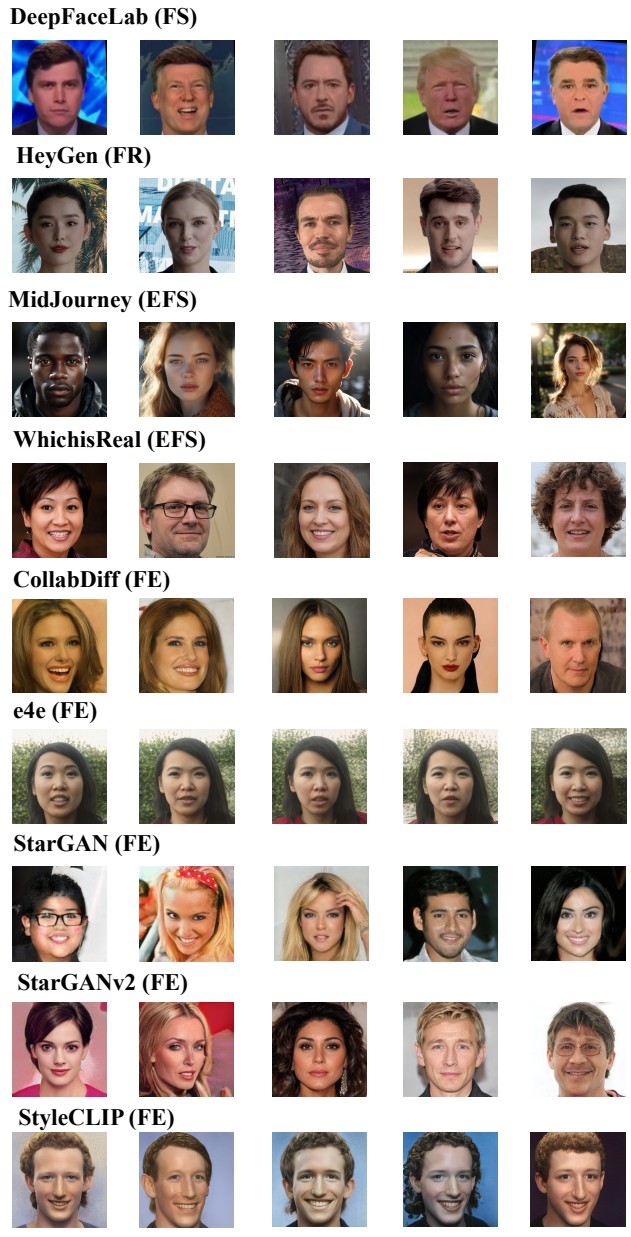

**HeyGen (FR)**

**MidJourney (EFS)**

**WhichisReal (EFS)**

**CollabDiff (FE)**

**e4e (FE)**

**StarGAN (FE)**

**StarGANv2 (FE)**

**StyleCLIP (FE)**

Figure 13: Illustration of visual examples for all unknown forgery data, including all types like FS, FR, EFS, and FE, simulating the real-world SoTA deepfakes.

blending the source face's features seamlessly. The main challenge faced by BlendFace is to achieve a stable and consistent face swap across video frames, which remains an area for future research. We use the official code from https://github.com/mapooon/BlendFace/ for implementation.

**6. UniFace [85]** UniFace is another recent face-swapping technique that was published in ECCV 2022. It leverages disentangled representations to transfer identity and attributes, utilizing Feature Disentanglement to separate identity and attribute embeddings unsupervised. It employs Attribute Transfer and Identity Transfer modules, powered by learned Feature Displacement Fields, to granularly manipulate attributes and model adaptive identity fusion, respectively, with an emphasis on maintaining consistency in reenactment and attribute preservation in swapping. The model is designed to handle challenging conditions like occlusions, extreme lighting, and large poses without reliance on pre-estimated structures. The main drawback of this method is the visible rectangular frame

around the swapped face, which affects the overall quality and realism of the face swap. Improving the blending and transition between the source and target faces remains a challenge for UniFace. We use the official code from https://github.com/xc-csc101/UniFace/ for implementation.

**7. MobileFaceSwap [86]**    MobileFaceSwap reduces the computational load of the model and realizes real-time face swapping on mobile devices. It introduces a new idea of integrating IDs into the network. MobileFaceSwap employs lightweight deep learning architectures, such as MobileNet, to achieve real-time performance on resource-constrained devices. The method focuses on maintaining the quality of the face swap while reducing the computational complexity, making it suitable for mobile applications and other scenarios where computational resources are limited. We use the official code from https://github.com/Seanseattle/MobileFaceSwap/ for implementation.

**8. e4s [46]**    e4s proposes a framework for explicit disentanglement of shape and texture based on facial components, considering both geometry and texture details. This method aims to create more realistic face swaps by separating and recombining facial features more accurately. By focusing on the individual components of the face, e4s can better preserve the identity of the target face while seamlessly blending the source face's features. This approach has the potential to improve the overall quality and realism of face swaps in various applications. The primary drawback of e4s is its noticeable temporal inconsistency, which makes it easy to detect using temporal models. We use the official code from https://github.com/e4s2022/e4s/ for implementation.

**9. FaceDancer [61]**    FaceDancer is a very recent work that was published in CVPR 2023, which proposes a single-stage, identity-based method for face swapping of unknown identities. This method aims to create high-quality face swaps even when the source and target faces have not been previously encountered during training. FaceDancer employs advanced deep learning techniques, such as GANs and attention mechanisms, to achieve realistic results while maintaining the identity of the target face. This method has the potential to broaden the applicability of face-swapping technology to a wider range of scenarios and use cases. This technology provides highly realistic visual results at both spatial and temporal levels. We use the official code from https://github.com/felixrosberg/FaceDancer/ for implementation.

**10. DeepFaceLab [14]**    DeepFaceLab [14] is a leading deepfake generation method. According to them, the majority of deepfake videos are generated by DeepFaceLab or other similar techniques (auto-encoder-based DF-family face-swapping techniques) such as DeepFake [15] used in FF++ and face-swap GAN [64]. They provide a complete, easy-to-use pipeline and provide end-to-end code with a windows software tool as well. They have also shared some synthesis models that can be used to generate deepfakes based on our requirements. Their method is a modification of the original Faceswap model in which they added an intervening network between the encoder and decoders. It helps the network to extract common features between source and target videos. Moreover, their loss function includes a mean squared error along with a structural dissimilarity index. We used this deepfake generation method in our paper to include the most recent and widely used method in our DF40. We use its official code repository from https://github.com/iperov/DeepFaceLab/ to create fake videos one by one.

**11. FOMM [67]**    FOMM is a highly classic method of image animation and face reenactment, previously used in the ForgeryNet dataset [22]. It employs deep learning techniques, such as GANs and attention mechanisms, to generate high-quality image animations and face reenactment results. FOMM has been widely used in various applications, including video editing, virtual reality, and digital content creation. Its ability to produce visually appealing and realistic animations has made it a popular choice for both research and practical applications. Thus, we propose to implement this classical synthesis method in our benchmark. We use the official code from https://github.com/AliaksandrSiarohin/first-order-model/ for implementation.

**12. FS_Vid2vid [79]**    FS_Vid2vid is another popular and classical synthesis method proposed by NVIDIA. It can generate unseen domain videos. However, when the test domain is significantly different from the training domain or the keypoint estimation is inaccurate, the effect is poor. FS_Vid2vid employs advanced deep learning techniques, such as GANs and attention mechanisms, to generate high-quality face swaps across different domains. The main challenge

faced by this method is to achieve stable and consistent face swaps when dealing with significant domain differences or inaccurate keypoint estimation. Improving the robustness and generalizability of FS_Vid2vid remains an area for future research. We use the official code from https://github.com/NVlabs/few-shot-vid2vid/ for implementation.

**13. Wav2Lip [55]**   Recently, audio-based facial reenactment techniques along with lip-syncing have been proposed by researchers. In lip-sync, the source person controls the mouth movement, and in face reenactment, facial features are manipulated in the target video. One of the most recent audio-driven facial reenactment methods is Wav2Lip, which aims to lip-sync the video with respect to any desired speech signal by reenacting the face. Wav2Lip has been widely used in various applications, including video editing, virtual reality, and digital content creation. Its ability to produce visually appealing and realistic lip-sync animations has made it a popular choice for both research and practical applications. We used this facial reenactment method because of the efficiency of its synthesis process for generating lip-synced video. We use the official code from https://github.com/Rudrabha/Wav2Lip/ for implementation. Also, the audio and text data to be used for driving is randomly sampled from LRS3 [2].

**14. MRAA [68]**   MRAA is also a highly classic image animation method, but there seems to be no existing deepfake dataset applied to this dataset so far. It identifies object parts, tracks them in driving videos, and estimates motion using principal axes within regions, leading to more stable and semantically meaningful representations. It also models non-object motion with an affine transformation to decouple foreground from background and disentangle shape and pose in the region space for improved animation fidelity. The framework is self-supervised, label-free, and optimized by reconstruction losses. MRAA has the potential to improve the overall quality and realism of image animations in various applications. We use the official code from https://github.com/snap-research/articulated-animation/ for implementation.

**15. OneShotFree [80]**   OneShotFree presents a reenactment-based approach for animating articulated objects with distinct parts, addressing the limitations of previous unsupervised methods that struggle with motion representation and articulation. It identifies object parts, tracks them in driving videos, and estimates motion using principal axes within regions, leading to more stable and semantically meaningful representations. It also models non-object motion with an affine transformation to decouple foreground from background and disentangle shape and pose in the region space for improved animation fidelity. The framework is self-supervised, label-free, optimized by reconstruction losses, and surpasses SoTA benchmarks, particularly for articulated objects like human bodies. We use the official code from https://github.com/zhanglonghao1992/One-Shot_Free-View_Neural_Talking_Head_Synthesis/ for implementation.

**16. PIRender [58]**   PIender is a neural rendering system for generating controllable portrait images by manipulating the motion of existing faces through intuitive and semantically meaningful parameters. It leverages 3D Morphable Models (3DMM) parameters for facial control, enabling precise adjustments to expressions and poses. Unlike many existing techniques that offer limited or indirect editing capabilities, PInder provides direct, fine-grained control for intuitive image editing, allowing realistic modifications to facial aspects like posture and expressions without the need for specialized software skills. Additionally, it's extended to address the audio-driven facial reenactment, synthesizing coherent videos from a single reference image and audio stream, showcasing the potential for intuitive controls in portrait manipulation and synthesis. We use the official code from https://github.com/RenYurui/PIRender/ for implementation.

**17. TPSM [93]**   TPSM (or called TPSMM) presents a thin-plate spline motion model for image animation, an unsupervised method aimed at addressing the challenge of animating objects with large pose discrepancies between source and driving images. It introduces a new framework, starting with thin-plate splines to estimate more flexible optical flow, enabling smoother warping of feature maps from source to drive image domain. Multi-resolution occlusion masks are employed to enhance feature fusion to improve occluded region restoration. Additionally, dedicated auxiliary losses are designed to ensure a clear division of labor among network modules, promoting high-quality image generation. The method is demonstrated to animate various objects, including faces, bodies, and animations, showing improved performance on benchmarks and highlighting its potential for

handling unseen manipulations. We use the official code from `https://github.com/yoyo-nb/Thin-Plate-Spline-Motion-Model/` for implementation.

**18. LIA [82]**   LIA is a recent SoTA face reenactment that was published in ICLR 2022, which edits the Latent code by learning linear changes in the Latent space, $i.e.$, learning a set of orthogonal Motion directions. This method aims to create more realistic face reenactments and animations by editing the Latent code directly. LIA is evaluated on real-world videos, demonstrating the capability to animate still images without bias, eliminating the need for structure. This approach has the potential to improve the overall quality and realism of face swaps and animations in various applications. We use the official code from `https://github.com/wyhsirius/LIA/` for implementation.

**19. DaGAN [26]**   This method is a depth-aware generative adversarial network (DaGAN) for talking head generation, aiming to create realistic synthetic videos that encapsulate the identity of a person from a source image and the pose from a driving video. It uniquely incorporates unsupervised depth recovery to learn dense facial geometry (depth) from videos, which is vital for accurate face synthesis and distinguishing facial structures amidst cluttered backgrounds without requiring expensive annotations. Depth maps facilitate the estimation of sparse key points capturing essential head movements and guide the generation of motion fields for warping source representations via depth-aware attention. Comprehensive experiments validate the model's capacity to generate high-quality videos, achieving significant advancements in unseen faces and showcasing the effectiveness of depth awareness in talking head synthesis. We use the official code from `https://github.com/harlanhong/CVPR2022-DaGAN/` for implementation.

**20. SadTalker [92]**   SadTalker is another recent popular talking head synthesis method that was published in CVPR 2023. The system animates talking heads in videos through audio-driven images using realistic facial expressions and head movements. It decomposes the problem into two stages: audio-motion mapping and 3D motion modeling. The ExpNet learns to extract facial expressions from audio with the aid of an initial identity reference frame's expression coefficients, ensuring lip synchronization and reducing uncertainty. PoseVAE models head movements in diverse styles, ensuring coherence with the beat alignment to audio. Evaluations show that SadTalker surpasses previous methods in generating higher-quality videos with accurate lip-sync and motion and reduced blurring, especially for challenging tasks like head pose transfer and talking-head videos. The system demonstrates the potential for creating coherent videos with natural-looking animations from audio and reference images, advancing video conferencing and digital media applications. We use the official code from `https://github.com/OpenTalker/SadTalker/` for implementation.

**21. MCNet [25]**   MCNet is another recent SoTA reenactment method that was published in ICCV 2023. This work presents a novel approach for generating high-fidelity talking head videos, aiming to animate a static source image with dynamic poses and expressions derived from a separate driving video while preserving the original person's identity. This is achieved through the Implicit Identity Representation Conditioned Memory Compensation Network (MCNet), which addresses the issue of ambiguous generation caused by large, complex motions in the driving video. MCNet introduces a unique mechanism that learns a global facial representation space and employs an implicit identity representation conditioned memory bank to provide rich structural and appearance priors. These priors help compensate for insufficient information in occluded regions or subtle expression variations, significantly reducing artifacts and enhancing video quality. The system incorporates a memory compensation module (MCM) and an implicit identity representation conditioned memory module (IICM) to query and utilize the learned memory effectively. Extensive experimentation affirms its capability to generate realistic talking head videos even under challenging conditions. We use the official code from `https://github.com/harlanhong/ICCV2023-MCNET/` for implementation.

**22. HyperReenact [5]**   HyperReenact is another recent SoTA reenactment method that was published in ICCV 2023. It is a one-shot neural face reenactment method capable of synthesizing realistic talking head sequences of a source individual driven by a target's facial pose, including 3D head orientation and expression. HyperReenact is designed to handle both self and cross-subject reenactment scenarios, requiring only a single source image. It excels in minimizing visual artifacts, even when there are drastic differences in head pose between the source and target images or when conducting cross-subject reenactment. By leveraging the photorealism and disentangled

properties of a pre-trained StyleGAN2 model, HyperReenact first inverts real images into the latent space before refining and retargeting them. The method is compared favorably with several state-of-the-art reenactment techniques, showcasing its proficiency in generating artifact-free details around facial areas such as the mouth and eyes, even under extreme head poses. Additionally, the paper includes extensive quantitative and qualitative evaluations, demonstrating HyperReenact's efficiency and effectiveness across various metrics and benchmark tests. We use the official code from `https://github.com/StelaBou/HyperReenact/` for implementation.

**23. HeyGen [23]**   HeyGen is an AI-powered video creation platform that simplifies the process of making professional-quality videos. With its user-friendly interface and advanced features, HeyGen has become a popular choice for users looking to create high-quality, engaging content without the need for extensive editing skills and time investment. One of the standout features of HeyGen is its AI-powered text-to-speech functionality, which allows users to effortlessly convert written text into natural-sounding speech. This feature enables users to create engaging voiceovers for their videos without the need for professional voice actors or expensive recording equipment. In addition to its text-to-speech capabilities, HeyGen also offers customizable avatars that can be used to represent speakers in the videos. These avatars can be tailored to match the desired appearance and style of the user, adding a personal touch to the video content. Furthermore, HeyGen's automated professional video editor streamlines the video creation process by intelligently combining video clips, images, text, and audio to produce polished and visually appealing videos. This feature saves users significant time and effort typically required for manual video editing. HeyGen's combination of AI-powered features and user-friendly interface makes it an ideal solution for users looking to create professional-quality videos with minimal effort. Its versatility and efficiency have made it popular for various applications, including content creation, digital marketing, e-learning, and social media promotion. We use the software from its official website `https://www.heygen.com/` for creating fake videos.

**24. VQGAN [20]**   VQGAN is a classical entire image synthesis method, which introduces a method that enhances transformers' capability to synthesize high-resolution images, overcoming their traditional inefficiency for long sequences typical of pixel-dense data. The approach enables efficient modeling and synthesis of high-resolution images by integrating CNN inductive bias for local interactions with transformers' expressiveness. It involves two stages: initially employing CNNs to learn a rich vocabulary of image components, followed by transformers arranging these components within high-resolution images. This method is adaptable to conditional synthesis tasks, allowing both non-spatial information, like object categories, and spatial data, such as segmentation, to guide image generation. Notably, it presents the first instance of transformers generating semantically-guided megapixel images and attains state-of-the-art performance among autoregressive models for class-conditional ImageNet synthesis. We use the official code from `https://github.com/CompVis/taming-transformers/` for implementation.

**25. StyleGAN2 [35]**   StyleGAN2 is an improved version of StyleGAN proposed by NVIDIA, enhancing the image quality and reducing the artifacts in the generated images. The authors propose architectural and training modifications to the StyleGAN, focusing on normalizing the generator's mapping network, progressively growing, and introducing regularization for improved conditioning between latent codes and images. Their approach enhances image quality, reduces artifacts, and increases the generator's invertibility while preserving its expressiveness. By improving upon the original StyleGAN, StyleGAN2 can generate more realistic and high-quality human face images with specific attributes. We use the official code from `https://github.com/NVlabs/stylegan2/` for implementation.

**26. StyleGAN3 [33]**   StyleGAN3 is an improved version of StyleGAN2 and StyleGAN proposed by NVIDIA, enhancing the image quality and reducing the artifacts in the generated images. StyleGAN3 addresses the issue of StyleGAN's performance degradation when dealing with unstructured datasets like ImageNet, typically designed for controllable synthesis. The authors find the limitation lies not in the training strategy but rather in the model's inherent design. They leverage the Projected GAN framework to apply powerful priors and a progressive growth strategy, training StyleGAN3 on ImageNet. The outcome, StyleGAN3, achieves a new state-of-the-art large-scale synthesis, generating 1024-resolution images for the first time. It demonstrates inversion and editing beyond

portraits or specific classes, showcasing broad applicability. We use the official code from `https://github.com/NVlabs/stylegan3/` for implementation.

**27. StyleGAN-XL [63]**  StyleGAN-XL is another version of the original StyleGAN that has been improved. It tackles the scalability of StyleGAN to datasets, particularly ImageNet's vastness and diversity, where it traditionally struggled. The authors refute the notion of StyleGAN's design being unsuitable for diversity and instead blame the training methodology. Introducing Projected GAN and progressive training, they successfully scale StyleGAN3 to ImageNet, birthing StyleGAN-XL, a new state-of-the-art in large-scale synthesis at 1024². StyleGAN-XL's flexibility inverts and edits images outside narrow domains, validating its robustness. We use the official code from `https://github.com/autonomousvision/stylegan-xl/` for implementation.

**28. Stable-Diffusion-2.1 [59]**  Stable-Diffusion is currently among the most popular methods for generating images from text. In addition to text-to-image, it can also perform image-to-image transformations. Stable-Diffusion employs advanced deep learning techniques, such as GANs and attention mechanisms, to achieve high-quality results. By supporting both text-to-image and image-to-image transformations, Stable-Diffusion can generate more diverse and high-quality images and animations. This method has been widely used in various applications, including content creation, data augmentation, and artistic expression, making it a popular choice for both research and practical applications. We use the official code from `https://github.com/Stability-AI/stablediffusion/` for implementation.

**29. DDPM [24]**  DDPM, or the Denoising Diffusion Probabilistic Model, is a pioneering work in the field of diffusion models. It generates high-quality images by gradually denoising them. DDPM employs advanced deep learning techniques, such as GANs and attention mechanisms, to achieve high-quality results. DDPM can generate visually appealing and realistic images with minimal artifacts by using a step-by-step denoising process. This method has been instrumental in advancing the field of diffusion models and has inspired the development of more advanced techniques, such as Stable-Diffusion and Collaborative-Diffusion. We use the official code from `https://github.com/lucidrains/denoising-diffusion-pytorch/` for implementation.

**30. RDDM [45]**  RDDM, or Residual Denoising Diffusion Model, is a very recent advanced diffusion model that was published in CVPR 2024. It aims to improve the robustness and stability of the image generation process. By separating the conventional single denoising diffusion process into residual and noise diffusion components, RDDM establishes a dual diffusion framework. This framework employs residual diffusion to model directed degradation from a target image to a degraded input, guiding restoration processes, while noise diffusion accounts for stochastic perturbations. RDDM's innovation lies in its ability to balance certainty (via residuals) and diversity (through noise), thereby unifying tasks with diverse requirements like image generation and restoration within a single, coherent model. We use the official code from `https://github.com/nachifur/RDDM/` for implementation.

**31. PixelArt-$\alpha$ [8]**  PixelArt-$\alpha$ is a very recent image synthesis method that was published in ICLR 2024. This work presents PixelArt-$\alpha$, a Transformer-based text-to-image synthesis model that achieves competitive image generation quality similar to leading models like Midjourney [49], with a focus on high-resolution outputs up to 1024x1024 pixels, all while maintaining a significantly reduced training cost. To overcome the hurdles of extensive computational expenses associated with advanced text-to-image models, PixelArt-$\alpha$ employs a three-stage training strategy. This strategy separately optimizes pixel dependencies, aligns text and images, and enhances image aesthetics. Furthermore, a new auto-labeling system enriches the dataset with high-concept-density captions, enhancing text-image alignment efficiency. Experimental results highlight PixelArt-$\alpha$'s capability to generate images of exceptional quality with remarkable adherence to textual descriptions, affirming its potential as a more environmentally and economically sustainable solution for photorealistic text-to-image synthesis. We use the official code from `https://github.com/PixArt-alpha/PixArt-alpha/` for implementation.

**32. DiT [54]**  DiT is a very popular architecture for realistic image synthesis recently. The work presents a novel class of diffusion models that leverage transformer architectures, termed Diffusion Transformers (DiTs), for image synthesis tasks. These models replace the traditionally used U-Net

backbone with transformers operating on latent image patches, demonstrating improved scalability and performance. The study analyzes the scalability of DiTs based on their forward pass complexity, measured in Gflops, revealing that models with higher computational capacity yield consistently lower FID scores, indicative of better image quality. Notably, the largest variants of DiTs, named DiT-XL/2, have surpassed previous diffusion models' performance on class-conditional ImageNet benchmarks at resolutions of 512x512 and 256x256, achieving a SoTA FID score of 2.27 at the latter resolution. The success of DiTs suggests a promising avenue for further scaling and potential integration into text-to-image synthesis pipelines, capitalizing on transformers' strong capabilities in generative modeling. We use the official code from `https://github.com/facebookresearch/DiT/` for implementation. We adopt the DiT-XL/2 version for generation in our benchmark.

**33. SiT [3]** SiT is another very popular architecture for realistic image synthesis recently. The paper introduces Scalable Interpolant Transformers (SiT), a novel family of generative models that build upon Diffusion Transformers (DiTs), focusing on enhancing diffusion-based image synthesis. SiT employs an interpolant framework, enabling more flexible distribution connections and facilitating a modular exploration of design elements in generative models such as learning dynamics, objectives, interpolant choices, and sampling strategies. By meticulously integrating these components, SiT outperforms DiT models across various sizes on the conditional ImageNet 256x256 benchmark, utilizing the same backbone architecture, parameter count, and computational complexity (GFLOPs). SiT's adaptability in tuning diffusion coefficients independently from the learning process further pushes its performance, achieving an FID-50K score of 2.06. This work underscores the potential of transformer-based architectures in advancing generative models and paves the way for future developments in scalable and efficient image generation. We use the official code from `https://github.com/willisma/SiT/` for implementation.

**34. MidJourney [49]** MidJourney is a really popular application, software, and image synthesis method that focuses on generating highly realistic content. Midjourney is a generative artificial intelligence program and service created and hosted by the San Francisco–based independent research lab Midjourney, Inc. Midjourney generates images from natural language descriptions, called prompts, similar to OpenAI's DALL-E and Stability AI's Stable Diffusion. By generating intermediate representations, MidJourney can be used for various applications, such as creative content generation. Its ability to produce realistic synthesis images makes it a popular choice for both research and practical applications. We use the software from its official website `https://www.midjourney.com/home/` for creating fake images. We adopt the latest version, MidJourney-6, for all generations.

**35. WhichFaceIsReal [83]** WhichFaceIsReal (or called WhichisReal) is a really popular software to create high realistic GAN-generated images and has been widely used in many academic research such as [77]. This software has been developed by Jevin West and Carl Bergstrom at the University of Washington as part of the Calling Bullshit project. All images are either computer-generated from thispersondoesnotexist.com using the StyleGAN software or real photographs from the FFHQ dataset of Creative Commons and public domain images. We use the software from its official website `https://www.whichfaceisreal.com/` for creating fake images. We create 2,000 fake images using this software in our benchmark to simulate the real-world realistic GAN-generated face content.

**36. Collaborative-Diffusion [27]** Collaborative-Diffusion (or called CollabDiff) is a very recent face editing method that was published in CVPR 2023. The proposed framework is designed for multi-modal face generation and editing. Unlike traditional diffusion models that primarily operate under single-modality control, this approach enables the simultaneous utilization of various modalities—such as text descriptions and facial masks—to guide the creation and modification of facial images. By leveraging pre-trained uni-modal diffusion models in conjunction, the system synthesizes high-quality, coherent images that adhere to the specified conditions without necessitating additional training. The implementation relies on Latent Diffusion Models (LDM) balanced for quality and efficiency, working within an autoencoder's latent space to manage computational demands. A Variational Autoencoder (VAE) is trained on the CelebA-HQ dataset to handle the transformation between image and latent representations. This collaborative synthesis method opens new avenues for creative expression in face manipulation tasks by empowering users to define age, facial features, or other attributes through a mix of textual and visual directives. We use the official code from `https://github.com/ziqihuangg/Collaborative-Diffusion/` for implementation.

**37. e4e [73]**    e4e, similar to pspNet, aims to improve the accuracy of projecting real images into the StyleGAN latent space. It mainly focuses on real-image editing using StyleGAN inversion, introducing an encoder architecture, referred to as e4e, tailored for effective editing post-inversion. The authors delve into the latent space characteristics of StyleGAN, revealing a tradeoff between distortion (reconstruction fidelity) and editability, as well as between distortion and perceptual quality. They outline design principles for encoders to balance these tradeoffs, enabling inversions that are conducive to meaningful manipulation while maintaining visual authenticity. Extensive qualitative and quantitative evaluations, along with a user study, demonstrate that the proposed inversion method followed by common editing techniques produces high-quality edits on real images from various challenging domains like cars and horses, with minimal loss in reconstruction accuracy. The paper further supplements these findings with detailed implementation specifics and additional experimental results in an appendix. We use the official code from https://github.com/omertov/encoder4editing/ for implementation.

**38. StarGAN [10]**    StarGAN is a very classical face editing method and has been applied in many academic datasets such as ForgeryNet [22]. StarGAN is a unified generative adversarial network model designed for image-to-image translation across multiple domains using a single generator and discriminator network. This innovative approach addresses the limitations of previous methods, which required separate models for each pair of domains, by enabling simultaneous training on diverse datasets within one framework. Consequently, StarGAN achieves enhanced image translation quality and versatility, allowing it to transform input images adaptively into any target domain specified. The system incorporates an auxiliary classifier to ensure images are correctly classified post-translation and employs a reconstruction loss to maintain content consistency between the input and output images. The authors express the hope that StarGAN will facilitate the development of various intriguing image translation applications spanning multiple domains. We use the official code from https://github.com/yunjey/stargan/ for implementation.

**39. StarGANv2 [11]**    StarGANv2 is an improved version of StarGAN. It is an advanced image-to-image translation model capable of producing diverse images across multiple domains with a single framework. It addresses two pivotal challenges: generating images of varied styles within a target domain and managing translations across numerous domains. StarGAN v2 surpasses preceding models' ability to synthesize images featuring rich styles across diverse domains, as validated through experiments on CelebA-HQ and a newly introduced animal faces dataset (AFHQ), which exhibits extensive inter- and intra-domain variability. The innovation lies in its style code generation process, style space flexibility, and the effective use of training data from multiple domains, resulting in a model that generalizes well to unseen images. The paper also emphasizes the model's capability for both latent-guided and reference-guided synthesis We use the official code from https://github.com/clovaai/stargan-v2/ for implementation.

**40. StyleCLIP [53]**    StyleCLIP is a popular face editing method that integrates the power of Contrastive Language-Image Pre-training (CLIP) models with StyleGAN to achieve text-driven manipulation of imagery without requiring manual intervention or predefined manipulation paths. StyleCLIP offers three key techniques: text-guided latent optimization using CLIP as a loss function for flexible yet time-consuming edits, a latent mapper trained for specific text prompts to provide local latent space adjustments swiftly, and a method mapping text prompts to global, input-agnostic style space directions within StyleGAN, enabling control over manipulation intensity and disentanglement. The work underscores the synergy of CLIP's broad visual concept understanding and StyleGAN's generative capabilities, opening up new possibilities for intuitive and creative image manipulation tasks. We use the official code from https://github.com/orpatashnik/StyleCLIP/ for implementation.

### A.2.3    Formulation of Manipulation Methods

To guarantee the *diversity* of deepfake approaches in the proposed DF40, we introduce and implement **40** distinct deepfake techniques, which are listed in Tab. 2. A detailed description of each deepfake method is provided in the Appendix. They are selected according to perspectives of modeling types, conditional sources, forgery effects, and functions. Formally, we denote $x_t(i_t, a_t, b_t)$ as the *target* subject to be manipulated, which possesses attributes ($i$: person identity, $a$: identity-agnostic content, $b$: external attributes) that can uniquely determine itself, while the *source* $x_s(i_s, a_s, b_s)$ is regarded

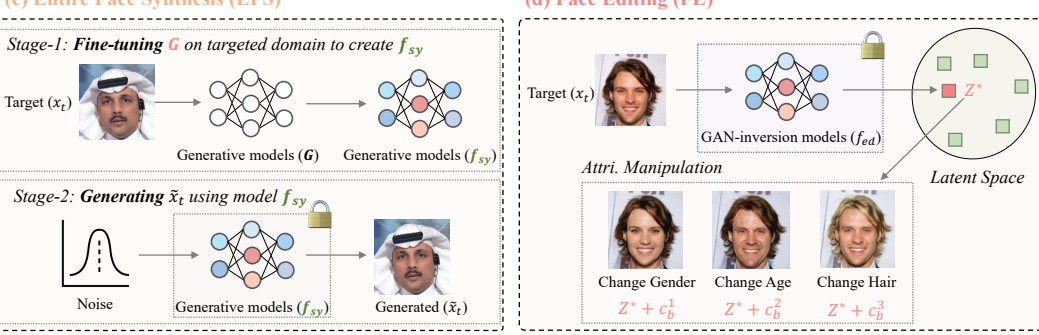

Figure 14: The general fake data generation pipeline of the proposed *DF40* dataset.

as the *conditional media* ($c$), driving the *target* to change either identity or attributes or even both. We use $f_{sw}, f_{re}, f_{sy}, f_{ed}$ to represent four types of deepfakes in our work: *FS*, *FR*, *EFS*, and *FE*, respectively. Fig. 1 provides some visual examples for each category in DF40, and Fig. 14 generally illustrates them from the method perspective. *(i) FS Approach:* From Fig. 14(a), FS aims to replace the content of $x_t$ with that of $x_s$ preserving the identity $i_s$. Formally, $\tilde{x}_t(\tilde{i}_s, a_t, b_t)$ only swaps identity $i$ from the source $x_s$ to the target $x_t$, and the identity-agnostic content $a$ are preserved. The swapping process can be expressed as:

$$f_{sw}(x_t(i_t, a_t, b_t), x_s(i_s, a_s, b_s)|\text{swap}) = \tilde{x}_t(\tilde{i}_s, a_t, b_t) \tag{1}$$

*(ii) FR Approach:* From Fig. 14(b), FR on $x_t(i_t, a_t, b_t)$ preserves its identity $i_t$ but has its *intrinsic* attributes $a_t$, *e.g.*, pose, mouth and expression manipulated by a driven variable $c_a$ and forms $\tilde{x}_t(i_t, \tilde{a}_s, b_t)$. Mathematically, we can drive the following equation:

$$f_{re}(x_t(i_t, a_t, b_t)|c_a) = \tilde{x}_t(i_t, \tilde{a}_s, b_t). \tag{2}$$

*(iii) EFS Approach:* EFS generates a completely synthesis face $\tilde{x}_t(\tilde{i}_t, \tilde{a}_t, \tilde{b}_t)$. In order to bridge the personal identity gap between generative models and other deepfake methods, we fine-tune the generative model $G$ with the same real data as other forgery methods (*e.g.*, e4s [46]) and obtain a face-synthesis model $f_{sy}$. After that, we can synthesize new synthesis faces from noise $n$ as follows:

$$f_{sy}(n) = \tilde{x}_t(\tilde{i}_t, \tilde{a}_t, \tilde{b}_t). \tag{3}$$

The general generation process can be seen in Fig. 14(c). *(iv) FE Approach:* From Fig. 14(d), FE on $x_t(i_t, a_t, b_t)$ has its *external* attributes $b_t$ altered, such as facial hair, age, gender, and ethnicity, controlled by conditional source $c_b$, to obtain $\tilde{x}_t(i_t, a_t, \tilde{b}_s)$. We also include multiple attribute manipulation with two editing approaches, *e.g.*, both hair and eyebrows are manipulated. We can formulate the transformation as follows:

$$f_{ed}(x_t(i_t, a_t, b_t)|c_b) = \tilde{x}_t(i_t, a_t, \tilde{b}_s), \tag{4}$$

where $\tilde{\cdot}$ means that the element is manipulated by forgery algorithms.

### A.2.4 Rationale of Fine-tuning EFS Methods

**Why did we fine-tune the generative model of EFS?** The primary motivation is to create an aligned data domain and fake methods for evaluation. We have two main reasons for pursuing this: **(1)**

---
**Algorithm 1** FS Method
---
**Input:** Target face $x_t(i_t, a_t, b_t)$, Source face $x_s(i_s, a_s, b_s)$
**Output:** Manipulated face $\tilde{x}_t$
 1: *Define* $x_t$ as the target face with attributes $i_t$, $a_t$, $b_t$
 2: *Define* $x_s$ as the source face with attributes $i_s$, $a_s$, $b_s$
 3: Apply FS method to swap identity $i$ from the source $x_s$ to the target $x_t$, preserving the identity-agnostic content $a$
 4: $\tilde{x}_t(\tilde{i}_s, a_t, b_t) \leftarrow f_{sw}(x_t(i_t, a_t, b_t), x_s(i_s, a_s, b_s)|\text{swap})$
 5: **return** $\tilde{x}_t$
---

---
**Algorithm 2** FR Method
---
**Input:** Target face $x_t(i_t, a_t, b_t)$, Conditional source $c_a$
**Output:** Manipulated face $\tilde{x}_t$
 1: *Define* $x_t$ as the target face with attributes $i_t$, $a_t$, $b_t$
 2: *Define* $c_a$ as the conditional source to manipulate the intrinsic attributes $a_t$
 3: Apply FR method to manipulate the intrinsic attributes $a_t$ of the target face $x_t$, preserving its identity $i_t$
 4: $\tilde{x}_t(i_t, \tilde{a}_s, b_t) \leftarrow f_{re}(x_t(i_t, a_t, b_t)|c_a)$
 5: **return** $\tilde{x}_t$
---

**Alignment with previous evaluation protocols:** Most previous deepfake detection studies have followed a widely-used protocol for selecting state-of-the-art detectors: training models on four deepfake methods from FF++ and testing them on one fake method from CDF. This led us to question whether we could expand this approach by implementing more fake methods on FF++ (beyond the original four) and CDF (testing on more than just one specific method). This consideration motivated us to choose FF++ and CDF as the real data sources for generating corresponding fake data. **(2) Isolating the individual effects on detection results:** Simply following the protocol of training on FF++ and testing on CDF would not allow us to isolate the individual influences of data domains and fake methods on the final detection results. Additionally, we noticed that many previous studies have performed cross-manipulation evaluations—training on one specific method and testing on others within the same data domain (FF++). However, the limited diversity of fake methods in previous datasets restricted cross-manipulation evaluations to just 4-5 different methods, most of which were face-swapping techniques. This limitation motivated us to create a significantly more diverse dataset to enable more comprehensive cross-manipulation evaluations.

**How did we fine-tune the generative model of EFS?** Please note the fine-tuning details for each EFS method can vary. In Algorithm.3, we provide a general understanding of the fine-tuning process. In details, we also release the specific fine-tuning code for reproduction (https://github.com/YZY-stack/DF40/tree/main/EFS_finetune_code/diffusion_based).

### A.2.5 Details of Original Datasets

In this study, we utilize several existing datasets to generate synthetic data for research purposes. The face data in these datasets mainly come from existing academic datasets, as shown in Table 2. In the following, we will introduce the leading institutions, dataset links, and official copyright issues for each dataset:

**FF++ Dataset [62]** The FaceForensics++ (FF++) dataset is designed for the detection and analysis of manipulated facial images and videos, specifically focusing on deepfake techniques. It contains over 1000 original videos and their manipulated versions generated using four different manipulation methods, namely Deepfakes [15], Face2Face [72], FaceSwap [21], and NeuralTextures [71]. The dataset is widely used for developing and benchmarking deepfake detection algorithms. Note some research works treat FaceShifter as another forgery method of the FF++, such as [19]. However, here we only consider the four mentioned methods that are included in the original FF++ paper [62]. Also, there are three versions of FF++ in terms of compression level, *i.e.*, raw, lightly compressed (c23), and heavily compressed (c40). The FF++ dataset is led by the University of Munich in Germany. The dataset can be accessed through the following

**Algorithm 3** EFS Method

---

**Input:** Noise $n$, Real Data $D$, Generative Model $G$
**Output:** Manipulated face $\tilde{x}_t$
1: *Define* $n$ as the noise used to generate a completely synthetic face
2: *Fine-tune* the generative model $G$ with the real data $D$ to obtain a face-synthesis model $f_{sy}$
3: Apply EFS method to generate a completely synthetic face from noise $n$ using the fine-tuned model $f_{sy}$
4: $\tilde{x}_t(\tilde{i}_t, \tilde{a}_t, \tilde{b}_t) \leftarrow f_{sy}(n)$
5: **return** $\tilde{x}_t$

---

**Algorithm 4** FE Method

---

**Input:** Target face $x_t(i_t, a_t, b_t)$, Conditional source $c_b$
**Output:** Manipulated face $\tilde{x}_t$
1: *Define* $x_t$ as the target face with attributes $i_t$, $a_t$, $b_t$
2: *Define* $c_b$ as the conditional source to manipulate the external attributes $b_t$
3: Apply FE method to alter the external attributes $b_t$ of the target face $x_t$, preserving its identity $i_t$ and intrinsic attributes $a_t$
4: $\tilde{x}_t(i_t, a_t, \tilde{b}_s) \leftarrow f_{ed}(x_t(i_t, a_t, b_t)|c_b)$
5: **return** $\tilde{x}_t$

---

link: https://github.com/ondyari/FaceForensics. The copyright information can be found at https://github.com/ondyari/FaceForensics/blob/master/LICENSE. The real human data sources for this dataset are not explicitly mentioned.

**Celeb-DF (CDF) Dataset [42]**   The Celeb-DF dataset is a large-scale dataset containing both real and manipulated celebrity videos, aimed at facilitating the development of deepfake detection algorithms. CDF dataset has two different versions: CDF-v1 and CDF-v2. Mostly, CDF-v2 is the default version of CDF. Here, we also adopt the CDF-v2 version. It consists of more than 5900 videos, with 563 real videos and over 5300 deepfake videos generated using a modified version of the DeepFakes approach. The dataset is designed to evaluate the performance of deepfake detection methods in real-world scenarios. The Celeb-DF dataset is led by the State University of New York at Buffalo. Specifically, the dataset can be accessed through the following official website: https://github.com/yuezunli/celeb-deepfakeforensics. The real human data sources for this dataset are not explicitly mentioned. The copyright information can be found at https://docs.google.com/forms/d/e/1FAIpQLScoXint8ndZXyJi2Rcy4MvDHkkZLyBFKN43lTeyiG88wrGOrA/viewform?pli=1.

**UADFV Dataset [41]**   The UADFV dataset is also led by the State University of New York at Buffalo. The dataset can be accessed through the following link: https://docs.google.com/forms/d/e/1FAIpQLScKPoOv15TIZ9Mn0nGScIVgKRM9tFWOmjh9eHKx57Yp-XcnxA/viewform. The real human data in this dataset seems to be directly crawled from the internet, as no official website is provided. The copyright information can be found at https://docs.google.com/forms/d/e/1FAIpQLScKPoOv15TIZ9Mn0nGScIVgKRM9tFWOmjh9eHKx57Yp-XcnxA/viewform.

**CelebA Dataset [47]**   The CelebA dataset is a large-scale face attributes dataset containing over 200,000 celebrity images, annotated with 40 attribute labels, such as gender, age, and facial expressions. It is primarily used for face attribute recognition tasks, including attribute prediction, face recognition, and face editing. The dataset has been widely adopted by the research community for developing and evaluating various machine learning models, including deep learning approaches. The CelebA dataset is led by the Chinese University of Hong Kong. The dataset can be accessed through the following link: https://mmlab.ie.cuhk.edu.hk/projects/CelebA.html. The real human data sources for this dataset are not explicitly mentioned. The complete copyright information for this dataset is not provided.

**FFHQ Dataset [34]**   The Flickr-Faces-HQ (FFHQ) dataset is a high-quality dataset of human faces designed for training generative adversarial networks (GANs) and other machine-learning models

to generate realistic human faces. It contains 70,000 high-resolution images (1024x1024 pixels) of diverse human faces collected from Flickr and annotated with age, gender, and ethnicity information. The dataset has been used in various applications, such as style-based GANs, super-resolution, and facial attribute manipulation. The FFHQ dataset is led by NVIDIA. The dataset can be accessed through the following link: https://github.com/NVlabs/ffhq-dataset. The real human data sources for this dataset come from Flickr: https://www.flickr.com/. The copyright information for this dataset is not explicitly mentioned.

**VFHQ Dataset [84]**   The Varied Faces-HQ (VFHQ) dataset is a high-quality dataset of diverse human faces designed for training and evaluating generative models, such as GANs, in generating realistic and diverse human faces. It consists of 28,000 high-resolution images (1024x1024 pixels) collected from the internet, with a focus on diversity in terms of age, gender, ethnicity, and facial expressions. The dataset is widely used for developing and benchmarking generative models and face-related applications. The VFHQ dataset is led by the Shenzhen Institutes of Advanced Technology, the Chinese Academy of Sciences, and the ARC Lab of Tencent PCG. The dataset can be accessed through the following link: https://liangbinxie.github.io/projects/vfhq/. The copyright information can be found at https://liangbinxie.github.io/projects/vfhq/license.txt. The real human data sources for this dataset are not explicitly mentioned.

## A.3   Introduction of the Used Detection Models

**Detector Implementation**   Here, we describe the general idea of the 7 detection algorithms used in our evaluation as follows.

1) **Xception**[62]: Xception is a deepfake detection method based on the XceptionNet model[12] trained on the FaceForensics++ dataset [62]. It has three variants: Xception-raw, Xception-c23, and Xception-c40, which are trained on raw videos, and H.264 videos with medium (23) and high degrees (40) of compression, respectively. In our work, we use the c23 version by default. Many detection works utilize Xception as their backbone or baseline, making it a popular choice in the field.

2) **RECCE** [6]: RECCE is a reconstruction-based detector that constructs a graph over encoder and decoder features in a multi-scale manner. It further utilizes the reconstruction differences as the forgery traces on the graph output as a guide to the final representation, which is fed into a classifier for forgery detection. The model performs end-to-end optimization for reconstruction and classification learning, and uses Xception as the backbone for extracting features from the given image.

3) **SPSL**[44]: SPSL is a frequency-based detector that combines spatial image and phase spectrum to capture the up-sampling artifacts of face forgery, improving the transferability (generalization ability) for face forgery detection. The paper provides a theoretical analysis of the validity of utilizing the phase spectrum and highlights the importance of local texture information over high-level semantic information for face forgery detection. This detector utilizes the Xception backbone[12] for feature extraction. The code for this detector is not publicly available, so we use the implementation version in DeepfakeBench [90]. This model uses Xception as the backbone for extracting features from the given image.

4) **SRM**[48]: SRM is also a frequency-based detector that extracts high-frequency noise features and fuses two different representations from RGB and frequency domains to improve the generalization ability for face forgery detection. This detector utilizes the Xception backbone[12] for feature extraction, and its code is not publicly available, so we use the implementation version in DeepfakeBench [90]. The code for this detector is not publicly available, so we use the implementation version in DeepfakeBench [90]. This model uses Xception as the backbone for extracting features from the given image.

5) **SBI** [65]: introduces a novel approach for deepfake detection using self-blended images (SBIs), which are synthetically created by blending pseudo source and target images derived from single pristine images. This technique reproduces common forgery artifacts like blending boundaries and statistical inconsistencies, thereby generating more generalized and less recognizable fake samples. Unlike conventional methods that often overfit to specific manipulation artifacts seen during training, SBIs encourage classifiers to learn broader, more robust representations that generalize better to unseen manipulations and scenes. Extensive experimentation demonstrates the

Table 9: The results of our experiments are generated using the following settings.

| Training Config | CLIP | Other Detectors |
|---|---|---|
| Image Size | 224, 224 | 256, 256 |
| Weight Initialization | CLIP Pre-trained | ImageNet Pre-trained |
| Optimizer | Adam | Adam |
| Base Learning Rate | 1e-5 | 2e-4 |
| Weight Decay | 5e-4 | 5e-4 |
| Optimizer Momentum | B1, B2=0.9, 0.999 | B1, B2=0.9, 0.999 |
| Batch Size | 32 | 32 |
| Training Epochs | 15 | 15 |
| Learning Rate Schedule | None, Constant | None, Constant |
| Flip Probability | 0.5 | 0.5 |
| Rotate Probability | 0.5 | 0.5 |
| Rotate Limit | [-10, 10] | [-10, 10] |
| Blur Probability | 0.5 | 0.5 |
| Blur Limit | [3, 7] | [3, 7] |
| Brightness Probability | 0.5 | 0.5 |
| Brightness Limit | [-0.1, 0.1] | [-0.1, 0.1] |
| Contrast Limit | [-0.1, 0.1] | [-0.1, 0.1] |
| Quality Lower | 40 | 40 |
| Quality Upper | 100 | 100 |

efficacy of the proposed method, particularly in closing the domain gap between training and test sets, as evidenced by significant performance improvements on the DFDC and DFDCP datasets. To align the settings with other detectors, we use the implementation version in DeepfakeBench [90] (v2 version).

6) **RFM** [76]: proposes an attention-based data augmentation framework, Representative Forgery Mining (RFM), which selectively occludes the top-sensitive facial areas, compelling the detector to explore previously overlooked regions for more representative forgery cues. This strategy is designed to refine and expand the detector's attention, enhancing its capacity to identify manipulated facial images generated by diverse techniques. To align the settings with other detectors, we use the implementation version in DeepfakeBench [90] (v2 version).

7) **CLIP** [57]: stands for Contrastive Language-Image Pretraining, is an innovative artificial intelligence model developed by OpenAI. This model is designed to understand and generate a meaningful connection between images and texts. Unlike traditional models that are trained for one specific task, CLIP is trained on a variety of data from the internet, enabling it to handle a wide range of tasks without requiring task-specific training data. It achieves this by learning to associate images and texts in a similar way to how humans do, making it a significant step forward in the field of AI. CLIP has broad applications, ranging from creating art to assisting in scientific research. To align the settings with other detectors, we use the implementation version in DeepfakeBench [90] (v2 version).

## A.4 Experimental Setup and Full Experimental Results

### A.4.1 Experimental Setup

**Training Details.** In the training module, we utilize the Adam optimization algorithm with a learning rate of 0.0002, except for CLIP. The batch size is set to 32 for both training and testing. For CLIP, we use the pre-trained weights from the default CLIP from OpenAI. For other detectors, we initialize the pre-trained weights from ImageNet. We summarize all the configurations used in our experiments (see Tab. 9). All experiments are conducted in a standardized environment using the NVIDIA V100 GPU to ensure fair and consistent evaluation. More software library dependencies can be seen on our website.

**Evaluation Metrics.** Regarding evaluation, we compute the frame-level Area Under the Curve (AUC) as our primary evaluation metric. We report video-level AUC for the video detector (*i.e.*, I3D). Note we can only perform training and testing I3D on FS and FR due to EFS and FE do not have video-format data. Furthermore, unlike DeepfakeBench, which does not involve using the validation

Table 10: Summary of the datasets used for deepfake detection. The table provides information on the number of real and fake videos, the total number of videos, whether rights have been cleared, the number of agreeing subjects, the total number of subjects, the number of synthesis methods, and the number of perturbations.

| Dataset | Real Videos | Fake Videos | Total Videos | Rights Cleared | Total Subjects | Synthesis Methods | Perturbations | Download Link |
|---|---|---|---|---|---|---|---|---|
| FF++ [62] | 1000 | 4000 | 5000 | NO | N/A | 4 | 2 | Hyper-link |
| FaceShifter [39] | 1000 | 1000 | 2000 | NO | N/A | 1 | - | Hyper-link |
| CDF-v2 [42] | 590 | 5639 | 6229 | NO | 59 | 1 | - | Hyper-link |
| DF-1.0 [30] | 50,000 | 10,000 | 60,000 | YES | 100 | 1 | 7 | Hyper-link |

set, we use the validation set to select the best-performing models and apply this checkpoint to detect *all* other testing data.

**Training and Testing Data Split. (1) For FF++:** We adhere to the *official* data split method, which uses 720 selected videos for training and 140 for testing and validation. When training models using the FF domain, such as FS (FF), we use the 720 corresponding fake videos for training and the original 720 real videos as real samples. **(2) For CDF:** We only use the 518 testing videos for all generations. Specifically, the original CDF testing data consists of 340 fake videos and 178 real videos. We select all video pairs (*e.g.*, ID1_ID2) from the 340 fake videos and create 680 fake videos, including the case of ID2_ID1. Thus, we have 680 fake videos for each method in the CDF domain. **(3) For DeepFaceLab:** We choose the 50 real videos from the original UADFV dataset [41] and then construct 100 pairs (*e.g.*, ID1_ID2 and ID2_ID1) for creating fake videos.

**Data Augmentation**   Our benchmark employs a series of widely used data augmentation methods for image processing, which are described as follows:

1. **Rotation:** Randomly rotates the image within a range of -10 to 10 degrees with a 0.5 probability, introducing diversity in object orientations and enhancing model robustness.

2. **Isotropic Resize:** Resizes the image while maintaining isotropy, preserving the aspect ratio, and accommodating different object sizes and proportions.

3. **Random Brightness and Contrast:** Randomly adjusts brightness and contrast with a 0.5 probability, helping the model generalize better to different lighting conditions.

4. **FancyPCA:** Applies the FancyPCA algorithm with a 0.5 probability, altering the principal components of the image to change the color distribution and create diverse training samples.

5. **Hue Saturation Value (HSV) Adjustment:** Randomly adjusts the hue, saturation, and value of the image, allowing for variations in color representation.

6. **Image Compression:** Applies image compression with a 0.5 probability, introducing artifacts and simulating real-world scenarios with lower-quality images or compression artifacts, improving model robustness in practical applications.

Regarding video methods, it is worth noting that the I3D model does not implement any augmentations. This is a limitation of the current implementation, and future work could explore incorporating video-specific augmentations to enhance the model's performance and robustness further.

**External Testing Datasets**   Our benchmark currently incorporates a collection of 4 widely recognized and extensively used datasets in the realm of deepfake forensics: FaceForensics++ (FF++/FF) [62], CelebDF-v2 (CDF-v2/CDF) [42], FaceShifter (Fsh) [39], and DeeperForensics-1.0 (DF-1.0) [30]. The detailed descriptions of each dataset are presented in Tab. 10.

Specifically, FF++ [62] is a large-scale database comprising over 1.8 million forged images from 1000 pristine videos. Forged images are generated by four face manipulation algorithms using the same set of pristine videos, *i.e.,* DeepFakes (FF-DF) [15], Face2Face (FF-F2F) [72], FaceSwap (FF-FS) [21], and NeuralTexture (FF-NT) [71]. Note that there are three versions of FF++ in terms of compression level, *i.e.*, raw, lightly compressed (c23), and heavily compressed (c40). Following previous works [90], we adopt the c23 version for our experiments. CDF is another popular face-swapping dataset that includes 5,639 synthetic videos and 890 real videos downloaded from YouTube. In our experiments, 518 testing videos are used for evaluation. Fsh and DF-1.0 generate high-fidelity face-swapping videos based on real videos from FF++.

In addition to these face forgery detection datasets, we also consider evaluating models trained on our DF40 into other *non-face* domain datasets (see results in Tab. 19 of the manuscript). Here, we select the recent representative dataset GenImage [95]. This dataset comprises over a million pairs of images, including both AI-synthesized fake images and genuine collected images, covering diverse content categories. It leverages SoTA generative techniques, including both diffusion models and GANs.

### A.4.2 Full Experimental Results and Discussion

In the main paper, we conduct evaluations under four standard protocols: Cross-forgery evaluation (**Protocol-1**), Cross-domain evaluation (**Protocol-2**), Toward unknown forgery and domain evaluation (**Protocol-3**), and One-Verse-All (OvA) evaluation (**Protocol-4**). For the OvA, we only show the heatmap to give an intuitive understanding of the performance. Here, we present the full results of OvA, including 2*31*31 evaluations. The results can be seen in Tab. 11 (testing on FS, FF domain), Tab. 12 (testing on FR, FF domain), Tab. 13 (testing on EFS, FF domain), Tab. 14 (testing on FS, CDF domain), Tab. 15 (testing on FR, CDF domain), Tab. 16 (testing on EFS, CDF domain).

**Further Discussion of Protocol-4: Training on DF40 (FF) and Testing on FS (FF).**   The key observation and finding in Tab. 11 is as follow:

**(1)** UniFace, BlendFace, and MobileSwap detection methods show high AUC values when detecting SimSwap deepfakes, indicating that these methods are particularly effective at detecting SimSwap-generated deepfakes. Conversely, FaceSwap deepfakes are most effectively detected by their own detection method (FaceSwap), with an AUC of 0.992.

**(2)** Among face-reenactment deepfake detection methods, Wav2Lip shows high AUC values in detecting UniFace, BlendFace, MobileSwap, and SimSwap deepfakes. This suggests that the Wav2Lip detection method is more versatile and effective in detecting various face-swapping deepfakes. However, it performs poorly in detecting FaceSwap deepfakes, with an AUC of 0.380.

**(3)** In the entire face synthesis category, DiT and SiT detection methods exhibit higher AUC values when detecting UniFace, BlendFace, and MobileSwap deepfakes, compared to other entire face synthesis detection methods. This suggests that these two methods are more effective in detecting face-swapping deepfakes, while other methods like StyleGAN2, StyleGAN3, and StyleGAN-XL have lower AUC values, making them less effective.

**(4)** The detection methods generally show a decline in performance when tested on face editing deepfakes, as indicated by lower AUC values in the last few rows of the table. This suggests that face editing deepfakes are more challenging to detect, possibly due to their subtler manipulations and greater diversity in editing techniques.

**(5)** The AUC values tend to be lower when detection methods are tested on deepfakes from different data domains (FF and CDF). This underlines the importance of incorporating cross-data-domain evaluations in the development and assessment of deepfake detection methods. It also highlights the need to improve the generalization capabilities of these methods to handle diverse deepfake techniques and data domains effectively.

In summary, the table reveals the varying effectiveness of detection methods across different deepfake types and data domains. It emphasizes the need for more robust detection methods that can generalize well across diverse deepfake techniques and data sources, as well as the importance of cross-data-domain evaluations in assessing detection performance.

**Further Discussion of Protocol-4: Training on DF40 (FF) and Testing on FR (FF).**   The key observation and finding in Tab. 12 is as follow:

**(1)** FSGAN detection method consistently achieves high AUC values across most of the face-reenactment deepfakes, indicating that it is particularly effective in detecting these deepfakes. On the other hand, the Hyperreenact detection method demonstrates a varying performance, with high AUC values for detecting MobileSwap and FOMM deepfakes but lower AUC values for detecting FaceSwap and Sadtalker deepfakes.

**(2)** Among the face-swapping deepfake detection methods, UniFace, BlendFace, and MobileSwap show relatively high AUC values when detecting FSGAN deepfakes. This suggests that these methods

are more effective at detecting FSGAN-generated deepfakes compared to other face-swapping deepfakes.

**(3)** The Wav2Lip detection method, which belongs to the face-reenactment category, exhibits high AUC values when detecting SimSwap deepfakes, indicating its effectiveness in detecting this specific face-swapping deepfake. However, its performance is inconsistent across other face-swapping deepfakes, with lower AUC values for detecting FaceSwap and e4s deepfakes.

**(4)** The detection methods trained on DF40 (FF) generally show a decline in performance when tested on face editing deepfakes, as indicated by lower AUC values in the last few rows of the table. This suggests that face-editing deepfakes are more challenging to detect, possibly due to their subtler manipulations and greater diversity in editing techniques.

**(5)** The AUC values tend to be lower when detection methods are tested on deepfakes from different data domains (FF and CDF). This underlines the importance of incorporating cross-data-domain evaluations in the development and assessment of deepfake detection methods. It also highlights the need to improve the generalization capabilities of these methods to handle diverse deepfake techniques and data domains effectively.

In summary, the table reveals the varying effectiveness of detection methods across different deepfake types and data domains when trained on DF40 (FF) and tested on FR (FF). It emphasizes the need for more robust detection methods that can generalize well across diverse deepfake techniques and data sources, as well as the importance of cross-data-domain evaluations in assessing detection performance.

**Further Discussion of Protocol-4: Training on DF40 (FF) and Testing on EFS (FF).** The key observation and finding in Tab. 13 is as follow:

**(1)** The LIA detection method consistently achieves high AUC values across most of the entire face synthesis (EFS) deepfakes, indicating that it is particularly effective in detecting these deepfakes. On the other hand, the Sadtalker detection method demonstrates a varying performance, with high AUC values for detecting StyleGAN2 and StyleGAN3 deepfakes but lower AUC values for detecting StyleGAN-XL and RDDM deepfakes.

**(2)** Among the face-swapping deepfake detection methods, UniFace and BlendFace show relatively high AUC values when detecting StyleGAN2 and StyleGAN3 deepfakes. This suggests that these methods are more effective at detecting these specific entire face synthesis deepfakes compared to other face-swapping deepfakes.

**(3)** The Wav2Lip detection method, which belongs to the face-reenactment category, exhibits high AUC values when detecting DDIM deepfakes, indicating its effectiveness in detecting this specific entire face synthesis deepfake. However, its performance is inconsistent across other entire face synthesis deepfakes, with lower AUC values for detecting StyleGAN2 and StyleGAN3 deepfakes.

**(4)** The detection methods trained on DF40 (FF) generally show a decline in performance when tested on face editing deepfakes, as indicated by lower AUC values in the last few rows of the table. This suggests that face editing deepfakes are more challenging to detect, possibly due to their subtler manipulations and greater diversity in editing techniques.

**(5)** The AUC values tend to be lower when detection methods are tested on deepfakes from different data domains (FF and CDF). This underlines the importance of incorporating cross-data-domain evaluations in the development and assessment of deepfake detection methods. It also highlights the need to improve the generalization capabilities of these methods to handle diverse deepfake techniques and data domains effectively.

In summary, the table reveals the varying effectiveness of detection methods across different deepfake types and data domains when trained on DF40 (FF) and tested on EFS (FF). It emphasizes the need for more robust detection methods that can generalize well across diverse deepfake techniques and data sources, as well as the importance of cross-data-domain evaluations in assessing detection performance.

Table 11: Part of quantitative results of OvA (Protocol-4): **Testing on FS (FF)** and Training on DF40 (FF). FS (FF) denotes all FS data within the FF domain, and DF40 (FF) denotes the combination results of FS (FF), FR (FF), and EFS (FF).

| | UniFace | BlendFace | MobileSwap | FaceSwap | e4s | FaceDancer | FSGAN | InSwap | SimSwap |
|---|---|---|---|---|---|---|---|---|---|
| UniFace | 0.999 | 0.953 | 0.842 | 0.553 | 0.829 | 0.598 | 0.784 | 0.725 | 0.946 |
| BlendFace | 0.982 | 0.996 | 0.984 | 0.571 | 0.711 | 0.611 | 0.845 | 0.838 | 0.955 |
| MobileSwap | 0.864 | 0.963 | 1.000 | 0.531 | 0.720 | 0.659 | 0.856 | 0.800 | 0.916 |
| FaceSwap | 0.614 | 0.681 | 0.725 | 0.992 | 0.634 | 0.518 | 0.78 | 0.591 | 0.576 |
| e4s | 0.829 | 0.736 | 0.688 | 0.531 | 1.000 | 0.637 | 0.763 | 0.668 | 0.610 |
| FaceDancer | 0.851 | 0.803 | 0.801 | 0.463 | 0.791 | 0.992 | 0.841 | 0.737 | 0.810 |
| FSGAN | 0.891 | 0.926 | 0.952 | 0.461 | 0.757 | 0.749 | 0.969 | 0.824 | 0.851 |
| InSwap | 0.905 | 0.930 | 0.925 | 0.516 | 0.623 | 0.662 | 0.866 | 0.999 | 0.963 |
| SimSwap | 0.984 | 0.984 | 0.982 | 0.417 | 0.668 | 0.684 | 0.833 | 0.840 | 1.000 |
| DaGAN | 0.667 | 0.756 | 0.894 | 0.430 | 0.744 | 0.644 | 0.726 | 0.705 | 0.703 |
| FOMM | 0.519 | 0.671 | 0.765 | 0.491 | 0.711 | 0.643 | 0.738 | 0.724 | 0.560 |
| Hyperreenact | 0.638 | 0.612 | 0.643 | 0.517 | 0.600 | 0.510 | 0.614 | 0.522 | 0.546 |
| TPSM | 0.693 | 0.750 | 0.890 | 0.468 | 0.763 | 0.645 | 0.740 | 0.714 | 0.671 |
| FS_vid2vid | 0.557 | 0.591 | 0.621 | 0.498 | 0.624 | 0.607 | 0.696 | 0.647 | 0.536 |
| MCNet | 0.700 | 0.768 | 0.862 | 0.482 | 0.709 | 0.675 | 0.783 | 0.767 | 0.673 |
| LIA | 0.860 | 0.852 | 0.835 | 0.525 | 0.858 | 0.759 | 0.899 | 0.840 | 0.650 |
| Wav2Lip | 0.936 | 0.944 | 0.746 | 0.380 | 0.620 | 0.611 | 0.723 | 0.698 | 0.939 |
| MRAA | 0.575 | 0.656 | 0.686 | 0.531 | 0.718 | 0.666 | 0.805 | 0.700 | 0.483 |
| OneShot | 0.515 | 0.638 | 0.789 | 0.527 | 0.760 | 0.672 | 0.681 | 0.681 | 0.543 |
| Pirender | 0.578 | 0.643 | 0.788 | 0.483 | 0.794 | 0.658 | 0.693 | 0.646 | 0.554 |
| Sadtalker | 0.78 | 0.847 | 0.985 | 0.371 | 0.699 | 0.738 | 0.754 | 0.865 | 0.804 |
| StyleGAN2 | 0.466 | 0.507 | 0.47 | 0.536 | 0.479 | 0.455 | 0.578 | 0.469 | 0.476 |
| StyleGAN3 | 0.465 | 0.507 | 0.472 | 0.536 | 0.479 | 0.454 | 0.578 | 0.470 | 0.476 |
| StyleGAN-XL | 0.525 | 0.572 | 0.575 | 0.567 | 0.533 | 0.462 | 0.504 | 0.572 | 0.551 |
| DDIM | 0.677 | 0.589 | 0.601 | 0.503 | 0.758 | 0.644 | 0.605 | 0.598 | 0.634 |
| DiT | 0.935 | 0.670 | 0.316 | 0.519 | 0.777 | 0.659 | 0.707 | 0.636 | 0.717 |
| pixart | 0.628 | 0.525 | 0.397 | 0.515 | 0.577 | 0.549 | 0.584 | 0.542 | 0.583 |
| SiT | 0.905 | 0.693 | 0.551 | 0.577 | 0.797 | 0.694 | 0.778 | 0.700 | 0.679 |
| SD2.1 | 0.607 | 0.526 | 0.470 | 0.477 | 0.597 | 0.555 | 0.554 | 0.532 | 0.465 |
| VQGAN | 0.668 | 0.718 | 0.635 | 0.460 | 0.650 | 0.521 | 0.655 | 0.618 | 0.591 |
| RDDM | 0.535 | 0.520 | 0.494 | 0.485 | 0.482 | 0.580 | 0.547 | 0.472 | 0.495 |

Table 12: Part of quantitative results of OvA (Protocol-4): **Testing on FR (FF)** and Training on DF40 (FF). FR (FF) denotes all FR data within the FF domain, and DF40 (FF) denotes the combination results of FS (FF), FR (FF), and EFS (FF).

| | DaGAN | FOMM | Hyperreenact | TPSM | FS_vid2vid | MCNet | LIA | Wav2Lip | MRAA | OneShot | Pirender | Sadtalker |
|---|---|---|---|---|---|---|---|---|---|---|---|---|
| UniFace | 0.659 | 0.644 | 0.907 | 0.695 | 0.692 | 0.724 | 0.779 | 0.711 | 0.579 | 0.574 | 0.544 | 0.593 |
| BlendFace | 0.758 | 0.807 | 0.914 | 0.773 | 0.813 | 0.812 | 0.793 | 0.797 | 0.774 | 0.749 | 0.771 | 0.658 |
| MobileSwap | 0.844 | 0.938 | 0.954 | 0.862 | 0.912 | 0.892 | 0.675 | 0.646 | 0.803 | 0.948 | 0.927 | 0.787 |
| faceswap | 0.530 | 0.714 | 0.321 | 0.524 | 0.574 | 0.583 | 0.632 | 0.443 | 0.691 | 0.690 | 0.682 | 0.471 |
| e4s | 0.819 | 0.840 | 0.822 | 0.829 | 0.894 | 0.849 | 0.783 | 0.596 | 0.827 | 0.827 | 0.799 | 0.641 |
| facedancer | 0.654 | 0.746 | 0.738 | 0.677 | 0.723 | 0.703 | 0.709 | 0.611 | 0.806 | 0.766 | 0.705 | 0.609 |
| fsgan | 0.931 | 0.982 | 0.999 | 0.934 | 0.937 | 0.944 | 0.867 | 0.779 | 0.920 | 0.956 | 0.967 | 0.857 |
| InSwap | 0.818 | 0.888 | 0.723 | 0.839 | 0.895 | 0.871 | 0.796 | 0.669 | 0.848 | 0.881 | 0.780 | 0.711 |
| SimSwap | 0.706 | 0.692 | 0.862 | 0.712 | 0.715 | 0.745 | 0.607 | 0.807 | 0.512 | 0.704 | 0.623 | 0.677 |
| DaGAN | 0.999 | 0.999 | 0.999 | 0.999 | 0.999 | 0.999 | 0.999 | 0.696 | 0.987 | 0.998 | 0.998 | 0.907 |
| FOMM | 0.985 | 0.999 | 0.999 | 0.985 | 0.999 | 0.989 | 0.926 | 0.522 | 0.999 | 0.999 | 0.999 | 0.829 |
| HypterReenact | 0.629 | 0.786 | 0.999 | 0.695 | 0.745 | 0.755 | 0.653 | 0.528 | 0.686 | 0.658 | 0.743 | 0.453 |
| TPSM | 0.999 | 0.999 | 0.999 | 0.999 | 0.999 | 0.999 | 0.999 | 0.694 | 0.990 | 0.999 | 0.999 | 0.897 |
| FS_vid2vid | 0.984 | 0.986 | 0.989 | 0.985 | 0.999 | 0.988 | 0.976 | 0.518 | 0.984 | 0.989 | 0.984 | 0.696 |
| MCNet | 0.999 | 0.999 | 0.999 | 0.999 | 0.999 | 0.999 | 0.999 | 0.684 | 0.996 | 0.999 | 0.999 | 0.908 |
| LIA | 0.999 | 0.982 | 0.990 | 0.999 | 0.999 | 0.999 | 0.999 | 0.704 | 0.998 | 0.955 | 0.975 | 0.778 |
| Wav2Lip | 0.633 | 0.785 | 0.817 | 0.818 | 0.758 | 0.808 | 0.786 | 0.994 | 0.532 | 0.563 | 0.623 | 0.702 |
| MRAA | 0.924 | 0.998 | 0.978 | 0.936 | 0.986 | 0.955 | 0.935 | 0.520 | 0.999 | 0.987 | 0.988 | 0.677 |
| OneShot | 0.983 | 0.999 | 0.996 | 0.983 | 0.999 | 0.986 | 0.899 | 0.497 | 0.996 | 0.999 | 0.999 | 0.822 |
| Pirender | 0.989 | 0.999 | 0.999 | 0.992 | 0.999 | 0.994 | 0.954 | 0.524 | 0.999 | 0.999 | 0.999 | 0.820 |
| Sadtalker | 0.995 | 0.999 | 0.999 | 0.997 | 0.999 | 0.998 | 0.810 | 0.735 | 0.966 | 0.999 | 0.999 | 0.994 |
| StyleGAN2 | 0.491 | 0.498 | 0.495 | 0.459 | 0.445 | 0.530 | 0.520 | 0.493 | 0.501 | 0.494 | 0.489 | 0.440 |
| StyleGAN3 | 0.492 | 0.499 | 0.495 | 0.459 | 0.446 | 0.531 | 0.520 | 0.493 | 0.502 | 0.497 | 0.492 | 0.441 |
| StyleGAN-XL | 0.567 | 0.634 | 0.750 | 0.542 | 0.540 | 0.590 | 0.538 | 0.505 | 0.595 | 0.591 | 0.553 | 0.507 |
| DDIM | 0.586 | 0.791 | 0.792 | 0.597 | 0.542 | 0.659 | 0.659 | 0.523 | 0.781 | 0.772 | 0.754 | 0.543 |
| DiT | 0.584 | 0.703 | 0.710 | 0.585 | 0.632 | 0.592 | 0.692 | 0.666 | 0.702 | 0.709 | 0.631 | 0.462 |
| Pixart | 0.516 | 0.567 | 0.532 | 0.532 | 0.435 | 0.552 | 0.618 | 0.587 | 0.627 | 0.552 | 0.542 | 0.592 |
| SiT | 0.661 | 0.921 | 0.926 | 0.684 | 0.733 | 0.710 | 0.786 | 0.597 | 0.936 | 0.866 | 0.840 | 0.523 |
| SD2.1 | 0.616 | 0.577 | 0.684 | 0.592 | 0.461 | 0.656 | 0.808 | 0.543 | 0.643 | 0.481 | 0.414 | 0.469 |
| VQGAN | 0.580 | 0.699 | 0.747 | 0.625 | 0.661 | 0.656 | 0.679 | 0.623 | 0.702 | 0.695 | 0.691 | 0.523 |
| RDDM | 0.520 | 0.467 | 0.562 | 0.524 | 0.441 | 0.556 | 0.572 | 0.555 | 0.451 | 0.407 | 0.404 | 0.500 |

Table 13: Part of quantitative results of OvA (Protocol-4): **Testing on EFS (FF)** and Training on DF40 (FF). EFS (FF) denotes all EFS data within the FF domain, and DF40 (FF) denotes the combination results of FS (FF), FR (FF), and EFS (FF).

| | StyleGAN2 | StyleGAN3 | StyleGAN-XL | ddim | DiT | Pixart | SiT | SD2.1 | VQGAN | RDDM |
|---|---|---|---|---|---|---|---|---|---|---|
| UniFace | 0.844 | 0.844 | 0.549 | 0.897 | 0.547 | 0.794 | 0.558 | 0.913 | 0.820 | 0.998 |
| BlendFace | 0.307 | 0.307 | 0.228 | 0.736 | 0.554 | 0.532 | 0.597 | 0.838 | 0.948 | 0.999 |
| MobileSwap | 0.671 | 0.671 | 0.806 | 0.911 | 0.571 | 0.241 | 0.587 | 0.627 | 0.828 | 0.989 |
| faceswap | 0.618 | 0.618 | 0.508 | 0.733 | 0.562 | 0.504 | 0.622 | 0.559 | 0.681 | 0.880 |
| e4s | 0.367 | 0.367 | 0.523 | 0.778 | 0.560 | 0.531 | 0.578 | 0.890 | 0.880 | 0.950 |
| facedancer | 0.858 | 0.858 | 0.770 | 0.868 | 0.584 | 0.467 | 0.619 | 0.881 | 0.812 | 0.870 |
| fsgan | 0.577 | 0.577 | 0.431 | 0.948 | 0.604 | 0.628 | 0.623 | 0.974 | 0.876 | 0.945 |
| InSwap | 0.263 | 0.263 | 0.576 | 0.658 | 0.478 | 0.441 | 0.532 | 0.629 | 0.665 | 0.940 |
| SimSwap | 0.805 | 0.805 | 0.706 | 0.849 | 0.497 | 0.570 | 0.496 | 0.648 | 0.842 | 0.967 |
| DaGAN | 0.807 | 0.807 | 0.907 | 0.789 | 0.614 | 0.572 | 0.627 | 0.889 | 0.582 | 0.999 |
| FOMM | 0.464 | 0.464 | 0.716 | 0.723 | 0.550 | 0.408 | 0.632 | 0.785 | 0.545 | 0.999 |
| HypterReenact | 0.448 | 0.448 | 0.598 | 0.705 | 0.502 | 0.665 | 0.543 | 0.874 | 0.855 | 0.990 |
| TPSM | 0.828 | 0.828 | 0.921 | 0.839 | 0.569 | 0.520 | 0.600 | 0.877 | 0.581 | 0.999 |
| FS_vid2vid | 0.635 | 0.635 | 0.782 | 0.763 | 0.573 | 0.520 | 0.600 | 0.844 | 0.582 | 0.999 |
| MCNet | 0.695 | 0.695 | 0.778 | 0.806 | 0.533 | 0.524 | 0.590 | 0.853 | 0.386 | 0.999 |
| LIA | 0.999 | 0.999 | 0.758 | 0.775 | 0.512 | 0.729 | 0.634 | 0.998 | 0.590 | 0.999 |
| Wav2Lip | 0.302 | 0.302 | 0.246 | 0.703 | 0.614 | 0.693 | 0.577 | 0.771 | 0.925 | 0.994 |
| MRAA | 0.517 | 0.517 | 0.609 | 0.664 | 0.585 | 0.567 | 0.650 | 0.902 | 0.566 | 0.997 |
| OneShot | 0.682 | 0.682 | 0.875 | 0.772 | 0.600 | 0.337 | 0.625 | 0.701 | 0.754 | 0.994 |
| Pirender | 0.775 | 0.775 | 0.826 | 0.801 | 0.621 | 0.538 | 0.649 | 0.793 | 0.782 | 0.996 |
| Sadtalker | 0.901 | 0.901 | 0.990 | 0.828 | 0.600 | 0.476 | 0.664 | 0.653 | 0.494 | 0.918 |
| StyleGAN2 | 0.999 | 0.999 | 0.473 | 0.671 | 0.506 | 0.395 | 0.486 | 0.609 | 0.602 | 0.464 |
| StyleGAN3 | 0.999 | 0.999 | 0.474 | 0.669 | 0.506 | 0.395 | 0.487 | 0.608 | 0.602 | 0.465 |
| StyleGAN-XL | 0.209 | 0.209 | 0.999 | 0.601 | 0.546 | 0.490 | 0.552 | 0.603 | 0.676 | 0.294 |
| DDIM | 0.819 | 0.819 | 0.363 | 0.999 | 0.550 | 0.633 | 0.589 | 0.939 | 0.877 | 0.964 |
| DiT | 0.795 | 0.795 | 0.341 | 0.891 | 0.995 | 0.773 | 0.994 | 0.855 | 0.976 | 0.979 |
| Pixart | 0.934 | 0.934 | 0.655 | 0.646 | 0.524 | 0.999 | 0.532 | 0.994 | 0.570 | 0.587 |
| SiT | 0.940 | 0.940 | 0.577 | 0.958 | 0.985 | 0.853 | 0.995 | 0.957 | 0.942 | 0.982 |
| SD2.1 | 0.823 | 0.823 | 0.687 | 0.929 | 0.492 | 0.878 | 0.539 | 0.999 | 0.811 | 0.998 |
| VQGAN | 0.356 | 0.356 | 0.787 | 0.709 | 0.579 | 0.506 | 0.595 | 0.835 | 0.999 | 0.999 |
| RDDM | 0.289 | 0.288 | 0.508 | 0.560 | 0.480 | 0.527 | 0.492 | 0.736 | 0.599 | 0.999 |

Table 14: Part of quantitative results of OvA (Protocol-4): **Testing on FS (CDF)** and Training on DF40 (FF). FS (FF) denotes all FS data within the FF domain, and DF40 (FF) denotes the combination results of FS (CDF), FR (CDF), and EFS (CDF).

| | UniFace | BlendFace | MobileSwap | FaceSwap | e4s | FaceDancer | FSGAN | InSwap | SimSwap |
|---|---|---|---|---|---|---|---|---|---|
| UniFace | 0.994 | 0.887 | 0.703 | 0.426 | 0.867 | 0.599 | 0.639 | 0.556 | 0.877 |
| BlendFace | 0.908 | 0.995 | 0.854 | 0.429 | 0.571 | 0.408 | 0.785 | 0.621 | 0.798 |
| MobileSwap | 0.467 | 0.843 | 0.999 | 0.525 | 0.431 | 0.340 | 0.764 | 0.542 | 0.686 |
| faceswap | 0.524 | 0.628 | 0.664 | 0.999 | 0.562 | 0.476 | 0.787 | 0.565 | 0.524 |
| e4s | 0.680 | 0.588 | 0.566 | 0.520 | 0.999 | 0.674 | 0.561 | 0.558 | 0.462 |
| facedancer | 0.694 | 0.651 | 0.571 | 0.254 | 0.739 | 0.997 | 0.602 | 0.551 | 0.535 |
| fsgan | 0.445 | 0.694 | 0.696 | 0.498 | 0.453 | 0.374 | 0.973 | 0.299 | 0.375 |
| InSwap | 0.684 | 0.827 | 0.835 | 0.486 | 0.396 | 0.400 | 0.670 | 0.967 | 0.902 |
| SimSwap | 0.879 | 0.865 | 0.781 | 0.466 | 0.505 | 0.409 | 0.416 | 0.553 | 0.997 |
| DaGAN | 0.227 | 0.368 | 0.602 | 0.549 | 0.496 | 0.699 | 0.706 | 0.328 | 0.298 |
| FOMM | 0.298 | 0.438 | 0.683 | 0.571 | 0.530 | 0.560 | 0.690 | 0.491 | 0.374 |
| HypterReenact | 0.532 | 0.657 | 0.662 | 0.380 | 0.682 | 0.611 | 0.680 | 0.489 | 0.502 |
| TPSM | 0.298 | 0.396 | 0.602 | 0.470 | 0.461 | 0.623 | 0.691 | 0.372 | 0.303 |
| FS_vid2vid | 0.463 | 0.520 | 0.571 | 0.536 | 0.613 | 0.746 | 0.616 | 0.587 | 0.468 |
| MCNet | 0.334 | 0.402 | 0.585 | 0.462 | 0.472 | 0.650 | 0.635 | 0.416 | 0.309 |
| LIA | 0.636 | 0.650 | 0.561 | 0.321 | 0.666 | 0.745 | 0.771 | 0.591 | 0.355 |
| Wav2Lip | 0.877 | 0.841 | 0.437 | 0.416 | 0.558 | 0.642 | 0.616 | 0.506 | 0.828 |
| MRAA | 0.457 | 0.546 | 0.644 | 0.562 | 0.617 | 0.562 | 0.714 | 0.617 | 0.383 |
| OneShot | 0.282 | 0.415 | 0.689 | 0.575 | 0.600 | 0.599 | 0.618 | 0.524 | 0.389 |
| Pirender | 0.297 | 0.399 | 0.620 | 0.396 | 0.612 | 0.560 | 0.614 | 0.410 | 0.304 |
| Sadtalker | 0.134 | 0.225 | 0.620 | 0.365 | 0.311 | 0.413 | 0.551 | 0.306 | 0.188 |
| StyleGAN2 | 0.552 | 0.629 | 0.622 | 0.699 | 0.488 | 0.500 | 0.612 | 0.567 | 0.607 |
| StyleGAN3 | 0.552 | 0.628 | 0.623 | 0.699 | 0.486 | 0.499 | 0.610 | 0.566 | 0.606 |
| StyleGAN-XL | 0.452 | 0.515 | 0.580 | 0.549 | 0.444 | 0.385 | 0.429 | 0.484 | 0.486 |
| DDIM | 0.561 | 0.530 | 0.570 | 0.549 | 0.813 | 0.588 | 0.446 | 0.513 | 0.529 |
| DiT | 0.939 | 0.691 | 0.500 | 0.517 | 0.807 | 0.828 | 0.526 | 0.669 | 0.661 |
| Pixart | 0.558 | 0.478 | 0.378 | 0.451 | 0.544 | 0.535 | 0.455 | 0.462 | 0.497 |
| SiT | 0.874 | 0.651 | 0.600 | 0.587 | 0.816 | 0.782 | 0.650 | 0.654 | 0.564 |
| SD2.1 | 0.546 | 0.522 | 0.478 | 0.363 | 0.616 | 0.593 | 0.488 | 0.482 | 0.454 |
| VQGAN | 0.557 | 0.731 | 0.525 | 0.391 | 0.587 | 0.472 | 0.498 | 0.582 | 0.489 |
| RDDM | 0.482 | 0.547 | 0.527 | 0.579 | 0.410 | 0.496 | 0.525 | 0.496 | 0.484 |

Table 15: Part of quantitative results of OvA (Protocol-4): **Testing on FR (CDF)** and Training on DF40 (FF). FR (FF) denotes all FR data within the FF domain, and DF40 (FF) denotes the combination results of FS (CDF), FR (CDF), and EFS (CDF).

| | DaGAN | FOMM | Hyperreenact | TPSM | FS_vid2vid | MCNet | LIA | Wav2Lip | MRAA | OneShot | Pirender | Sadtalker |
|---|---|---|---|---|---|---|---|---|---|---|---|---|
| UniFace | 0.604 | 0.615 | 0.762 | 0.518 | 0.668 | 0.667 | 0.578 | 0.6522 | 0.474 | 0.573 | 0.489 | 0.447 |
| BlendFace | 0.585 | 0.626 | 0.737 | 0.457 | 0.671 | 0.636 | 0.683 | 0.670 | 0.547 | 0.625 | 0.579 | 0.386 |
| MobileSwap | 0.677 | 0.877 | 0.694 | 0.464 | 0.762 | 0.715 | 0.532 | 0.317 | 0.527 | 0.902 | 0.737 | 0.481 |
| faceswap | 0.621 | 0.716 | 0.312 | 0.436 | 0.615 | 0.631 | 0.668 | 0.498 | 0.673 | 0.746 | 0.733 | 0.448 |
| e4s | 0.743 | 0.837 | 0.429 | 0.589 | 0.793 | 0.769 | 0.558 | 0.414 | 0.671 | 0.84 | 0.735 | 0.492 |
| facedancer | 0.503 | 0.628 | 0.459 | 0.390 | 0.575 | 0.553 | 0.601 | 0.478 | 0.647 | 0.64 | 0.580 | 0.393 |
| fsgan | 0.584 | 0.727 | 0.959 | 0.491 | 0.619 | 0.660 | 0.602 | 0.415 | 0.508 | 0.601 | 0.665 | 0.444 |
| InSwap | 0.616 | 0.774 | 0.441 | 0.547 | 0.744 | 0.651 | 0.647 | 0.503 | 0.679 | 0.786 | 0.555 | 0.512 |
| SimSwap | 0.468 | 0.497 | 0.539 | 0.365 | 0.467 | 0.497 | 0.332 | 0.547 | 0.245 | 0.565 | 0.411 | 0.345 |
| DaGAN | 0.997 | 0.981 | 0.976 | 0.939 | 0.999 | 0.997 | 0.692 | 0.325 | 0.811 | 0.962 | 0.950 | 0.716 |
| FOMM | 0.936 | 0.999 | 0.971 | 0.793 | 0.991 | 0.949 | 0.757 | 0.350 | 0.976 | 0.994 | 0.989 | 0.678 |
| HypterReenact | 0.699 | 0.827 | 0.999 | 0.527 | 0.845 | 0.756 | 0.639 | 0.435 | 0.714 | 0.791 | 0.792 | 0.431 |
| TPSM | 0.989 | 0.989 | 0.966 | 0.916 | 0.985 | 0.991 | 0.687 | 0.339 | 0.874 | 0.981 | 0.971 | 0.739 |
| FS_vid2vid | 0.963 | 0.970 | 0.982 | 0.909 | 0.998 | 0.967 | 0.812 | 0.472 | 0.958 | 0.975 | 0.972 | 0.630 |
| MCNet | 0.990 | 0.973 | 0.972 | 0.918 | 0.997 | 0.992 | 0.747 | 0.355 | 0.894 | 0.958 | 0.951 | 0.686 |
| LIA | 0.990 | 0.923 | 0.964 | 0.913 | 0.997 | 0.994 | 0.989 | 0.530 | 0.984 | 0.878 | 0.886 | 0.469 |
| Wav2Lip | 0.628 | 0.406 | 0.649 | 0.601 | 0.616 | 0.666 | 0.592 | 0.962 | 0.272 | 0.341 | 0.381 | 0.462 |
| MRAA | 0.901 | 0.995 | 0.932 | 0.760 | 0.985 | 0.933 | 0.889 | 0.417 | 0.995 | 0.988 | 0.977 | 0.632 |
| OneShot | 0.933 | 0.998 | 0.949 | 0.816 | 0.996 | 0.941 | 0.692 | 0.358 | 0.961 | 0.998 | 0.995 | 0.725 |
| Pirender | 0.946 | 0.995 | 0.980 | 0.788 | 0.994 | 0.962 | 0.733 | 0.330 | 0.963 | 0.996 | 0.999 | 0.654 |
| Sadtalker | 0.792 | 0.916 | 0.691 | 0.756 | 0.923 | 0.791 | 0.185 | 0.228 | 0.381 | 0.915 | 0.876 | 0.863 |
| StyleGAN2 | 0.689 | 0.723 | 0.590 | 0.556 | 0.719 | 0.701 | 0.618 | 0.572 | 0.630 | 0.718 | 0.691 | 0.537 |
| StyleGAN3 | 0.689 | 0.723 | 0.592 | 0.555 | 0.719 | 0.701 | 0.619 | 0.571 | 0.630 | 0.720 | 0.692 | 0.538 |
| StyleGAN-XL | 0.549 | 0.564 | 0.456 | 0.485 | 0.545 | 0.561 | 0.520 | 0.463 | 0.506 | 0.567 | 0.496 | 0.410 |
| DDIM | 0.494 | 0.691 | 0.502 | 0.345 | 0.480 | 0.540 | 0.571 | 0.397 | 0.591 | 0.705 | 0.695 | 0.380 |
| DiT | 0.707 | 0.748 | 0.549 | 0.592 | 0.716 | 0.699 | 0.693 | 0.625 | 0.687 | 0.751 | 0.664 | 0.417 |
| Pixart | 0.506 | 0.570 | 0.717 | 0.478 | 0.489 | 0.522 | 0.493 | 0.506 | 0.596 | 0.598 | 0.524 | 0.489 |
| SiT | 0.689 | 0.893 | 0.771 | 0.532 | 0.742 | 0.716 | 0.801 | 0.518 | 0.881 | 0.871 | 0.790 | 0.405 |
| SD2.1 | 0.580 | 0.624 | 0.519 | 0.399 | 0.583 | 0.609 | 0.688 | 0.479 | 0.570 | 0.598 | 0.440 | 0.334 |
| VQGAN | 0.580 | 0.694 | 0.404 | 0.460 | 0.618 | 0.615 | 0.648 | 0.545 | 0.575 | 0.733 | 0.669 | 0.421 |
| RDDM | 0.632 | 0.633 | 0.469 | 0.530 | 0.728 | 0.650 | 0.545 | 0.517 | 0.513 | 0.622 | 0.430 | 0.482 |

Table 16: Part of quantitative results of OvA (Protocol-4): **Testing on EFS (CDF)** and Training on DF40 (FF). EFS (FF) denotes all EFS data within the FF domain, and DF40 (FF) denotes the combination results of FS (CDF), FR (CDF), and EFS (CDF).

| | StyleGAN2 | StyleGAN3 | StyleGAN-XL | ddim | DiT | Pixart | SiT | SD2.1 | VQGAN | RDDM |
|---|---|---|---|---|---|---|---|---|---|---|
| UniFace | 0.697 | 0.639 | 0.592 | 0.784 | 0.544 | 0.709 | 0.568 | 0.882 | 0.703 | 0.999 |
| BlendFace | 0.498 | 0.309 | 0.326 | 0.457 | 0.585 | 0.360 | 0.602 | 0.747 | 0.895 | 0.984 |
| MobileSwap | 0.578 | 0.550 | 0.498 | 0.598 | 0.495 | 0.175 | 0.511 | 0.444 | 0.570 | 0.997 |
| faceswap | 0.538 | 0.480 | 0.334 | 0.540 | 0.470 | 0.428 | 0.539 | 0.476 | 0.570 | 0.997 |
| e4s | 0.345 | 0.298 | 0.469 | 0.638 | 0.494 | 0.336 | 0.488 | 0.808 | 0.754 | 0.997 |
| facedancer | 0.487 | 0.601 | 0.492 | 0.711 | 0.469 | 0.387 | 0.544 | 0.716 | 0.551 | 0.956 |
| fsgan | 0.551 | 0.330 | 0.789 | 0.536 | 0.417 | 0.479 | 0.408 | 0.879 | 0.372 | 0.933 |
| InSwap | 0.441 | 0.342 | 0.487 | 0.393 | 0.514 | 0.258 | 0.549 | 0.449 | 0.584 | 0.993 |
| SimSwap | 0.522 | 0.406 | 0.507 | 0.552 | 0.456 | 0.237 | 0.422 | 0.340 | 0.567 | 0.990 |
| DaGAN | 0.408 | 0.304 | 0.675 | 0.206 | 0.466 | 0.320 | 0.424 | 0.625 | 0.207 | 0.985 |
| FOMM | 0.477 | 0.431 | 0.610 | 0.240 | 0.631 | 0.223 | 0.630 | 0.615 | 0.547 | 0.865 |
| HypterReenact | 0.588 | 0.469 | 0.795 | 0.515 | 0.490 | 0.506 | 0.436 | 0.648 | 0.681 | 0.853 |
| TPSM | 0.370 | 0.379 | 0.744 | 0.199 | 0.517 | 0.289 | 0.507 | 0.580 | 0.342 | 0.973 |
| FS_vid2vid | 0.550 | 0.581 | 0.723 | 0.346 | 0.662 | 0.436 | 0.662 | 0.746 | 0.629 | 0.996 |
| MCNet | 0.312 | 0.389 | 0.653 | 0.215 | 0.470 | 0.405 | 0.467 | 0.627 | 0.294 | 0.974 |
| LIA | 0.739 | 0.905 | 0.575 | 0.398 | 0.370 | 0.671 | 0.439 | 0.992 | 0.352 | 0.999 |
| Wav2Lip | 0.206 | 0.182 | 0.305 | 0.534 | 0.597 | 0.445 | 0.563 | 0.680 | 0.868 | 0.995 |
| MRAA | 0.607 | 0.608 | 0.616 | 0.409 | 0.683 | 0.468 | 0.714 | 0.853 | 0.632 | 0.902 |
| OneShot | 0.556 | 0.507 | 0.754 | 0.305 | 0.654 | 0.157 | 0.637 | 0.363 | 0.703 | 0.870 |
| Pirender | 0.499 | 0.631 | 0.696 | 0.344 | 0.622 | 0.242 | 0.612 | 0.471 | 0.787 | 0.912 |
| Sadtalker | 0.430 | 0.683 | 0.828 | 0.168 | 0.379 | 0.176 | 0.369 | 0.257 | 0.081 | 0.162 |
| StyleGAN2 | 0.590 | 0.994 | 0.517 | 0.551 | 0.491 | 0.425 | 0.505 | 0.558 | 0.599 | 0.811 |
| StyleGAN3 | 0.590 | 0.994 | 0.518 | 0.550 | 0.491 | 0.424 | 0.504 | 0.558 | 0.598 | 0.810 |
| StyleGAN-XL | 0.384 | 0.211 | 0.810 | 0.467 | 0.486 | 0.405 | 0.541 | 0.504 | 0.676 | 0.833 |
| DDIM | 0.588 | 0.554 | 0.318 | 0.999 | 0.481 | 0.474 | 0.539 | 0.737 | 0.673 | 0.944 |
| DiT | 0.824 | 0.744 | 0.315 | 0.922 | 0.986 | 0.862 | 0.992 | 0.858 | 0.912 | 0.999 |
| Pixart | 0.528 | 0.730 | 0.571 | 0.684 | 0.584 | 0.999 | 0.572 | 0.971 | 0.554 | 0.934 |
| SiT | 0.986 | 0.954 | 0.400 | 0.973 | 0.969 | 0.935 | 0.991 | 0.973 | 0.858 | 0.978 |
| SD2.1 | 0.456 | 0.441 | 0.551 | 0.650 | 0.477 | 0.869 | 0.506 | 0.999 | 0.543 | 0.999 |
| VQGAN | 0.254 | 0.257 | 0.434 | 0.578 | 0.531 | 0.408 | 0.520 | 0.522 | 0.999 | 0.999 |
| RDDM | 0.354 | 0.345 | 0.429 | 0.589 | 0.558 | 0.546 | 0.542 | 0.787 | 0.693 | 0.999 |

Table 17: Ablation study regarding the impact of the fake region.

| Whole | Face | Mouth | Nose | Eye |
|---|---|---|---|---|
| 0.952 / 0.953 | 0.730 / 0.700 | 0.477 / 0.454 | 0.483 / 0.458 | 0.470 / 0.454 |

Table 18: Results of evaluating three models (*i.e.*, I3D, CLIP-base, and CLIP-large) trained on different variants of the proposed DF40 on other *previous* deepfake datasets.

| Model | Training Set | Testing Set | | | | | | |
|---|---|---|---|---|---|---|---|---|
| | | CDF-v2 [42] | FF-DF [15] | FF-F2F [72] | FF-FS [21] | FF-NT [71] | FaceShifter [39] | DF1.0 [30] |
| I3D (video-level AUC) | FS (FF) | 0.863 | 0.980 | 0.939 | 0.999 | 0.879 | 0.979 | 0.875 |
| | FR (FF) | 0.684 | 0.917 | 0.846 | 0.374 | 0.895 | 0.876 | 0.807 |
| | FS (FF) + FR (FF) | 0.764 | 0.970 | 0.926 | 0.994 | 0.928 | 0.953 | 0.929 |
| CLIP-base | FS (FF) | 0.823 | 0.965 | 0.678 | 0.985 | 0.585 | 0.932 | 0.945 |
| | FR (FF) | 0.649 | 0.669 | 0.572 | 0.457 | 0.682 | 0.625 | 0.947 |
| | EFS (FF) | 0.553 | 0.618 | 0.604 | 0.634 | 0.539 | 0.653 | 0.590 |
| | DF40 (FF) | 0.799 | 0.932 | 0.698 | 0.973 | 0.683 | 0.889 | 0.937 |
| CLIP-large | FS (FF) | 0.905 | 0.989 | 0.777 | 0.991 | 0.608 | 0.963 | 0.965 |
| | FR (FF) | 0.698 | 0.756 | 0.614 | 0.487 | 0.738 | 0.626 | 0.983 |
| | EFS (FF) | 0.616 | 0.733 | 0.634 | 0.730 | 0.571 | 0.772 | 0.739 |
| | DF40 (FF) | 0.891 | 0.978 | 0.753 | 0.973 | 0.717 | 0.927 | 0.986 |

## A.5  Further Analysis Results

### A.5.1  More Results of Different Fake Regions

We create a self-reconstruction image (SRI) for each original real image, where the SRI images act as fakes with a global forgery pattern (entire image synthesis). We then perform a blending operation to replace the reconstructed face with the original face, creating a localized fake (L-SRI), similar to other typical localized face forgery methods. We consider five common types of fake regions: whole, face, mouth, nose, and eyes. Results in Tab. 17 quantitatively verify the significance of the fake region in detection.

### A.5.2  Generalization to Non-face Domain AIGCs

We also evaluate **non-face domain data** to determine if models trained on face-domain data can transfer to non-face-domain detection. From Tab. 19, we can observe the following: **(1)** The model trained on EFS (FF) demonstrates the highest average AUC value (0.815) when tested on non-face domain GenImage deepfakes. This suggests that the model, when trained on EFS, learns features related to forgery traces rather than features specific to face content. This ability to generalize from face AIGCs to non-face AIGCs indicates the robustness of the model and its potential to detect deepfakes across different domains.

**(2)** The models trained on FS (FF) and FR (FF) show lower average AUC values (0.564 and 0.493, respectively) when tested on non-face domain GenImage deepfakes. This suggests that these models may be learning features more specific to face content or the particular manipulation techniques (face swap or drive) used in their training sets, limiting their ability to generalize to non-face AIGCs.

**(3)** The model trained on DF40 (FF), which includes a combination of different deepfake techniques, shows a relatively high average AUC value (0.746). While this is lower than the model trained on EFS (FF), it still indicates some level of generalization to non-face AIGCs. This suggests that training on a diverse set of deepfake techniques can help the model learn more general forgery-related features.

In summary, these results emphasize the importance of the training set in deepfake detection. Models trained on EFS (FF) appear to learn more general forgery-related features, enabling them to perform better on non-face AIGCs. Models trained on FS (FF) and FR (FF), on the other hand, maybe learning more technique-specific or face-specific features, limiting their performance on non-face AIGCs.

Table 19: Evaluation of training CLIP-large on different variants of the proposed DF40 (face domain) to detect GenImage [95] (**non-face domain**), pure AIGCs.

| Model | Training Set | Testing Set | | | | | | | | |
|---|---|---|---|---|---|---|---|---|---|---|
| | | ADM | BigGAN | GLide | MidJourney | SD-v4 | SD-v5 | Vqdm | Wukong | Avg. |
| CLIP-large | FS (FF) | 0.589 | 0.679 | 0.407 | 0.447 | 0.513 | 0.494 | 0.806 | 0.574 | 0.564 |
| | FR (FF) | 0.695 | 0.595 | 0.230 | 0.505 | 0.437 | 0.417 | 0.549 | 0.519 | 0.493 |
| | EFS (FF) | 0.912 | 0.992 | 0.906 | 0.591 | 0.729 | 0.72 | 0.956 | 0.713 | 0.815 |
| | DF40 (FF) | 0.911 | 0.967 | 0.736 | 0.571 | 0.630 | 0.614 | 0.882 | 0.660 | 0.746 |

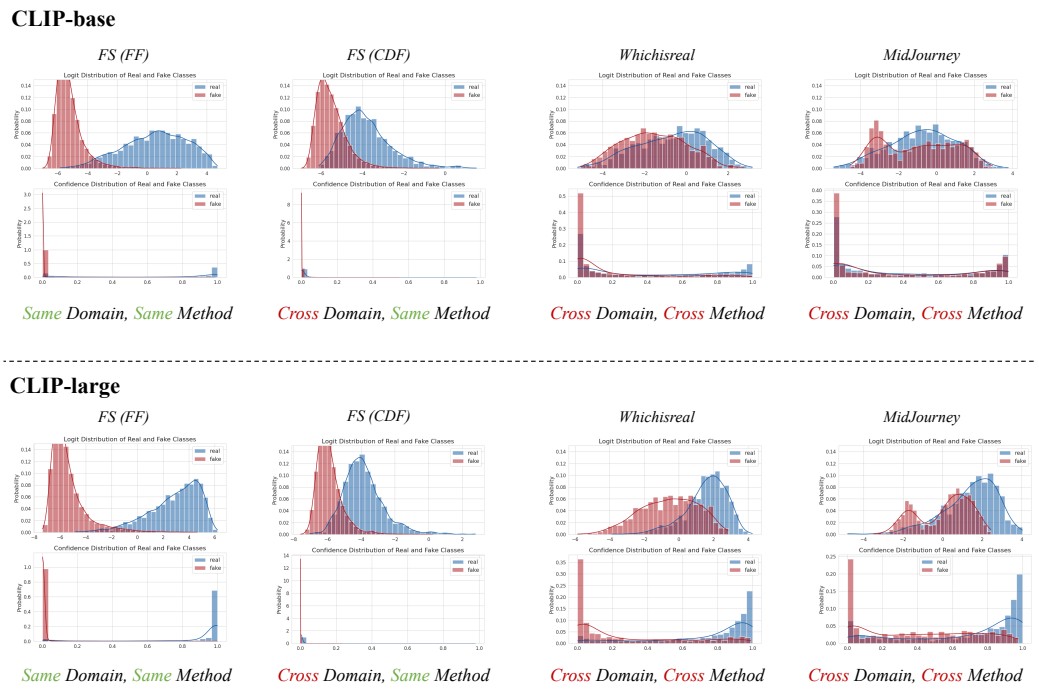

Figure 15: Logits and confidence analysis for comparing CLIP-base and CLIP-large. We show that CLIP-large can learn more robust real-people features, thereby generalizing well in unseen forgeries.

This highlights the need for more research into training methods to help models generalize better across different deepfake techniques and domains.

### A.5.3 Generalization to External/Previous Deepfake Datasets

We also evaluate the model trained on DF40 to detect previous deepfake datasets. Results in Tab. 18 show that **(1)** For the I3D detection method, training on FS (FF) and FR (FF) combined (FS (FF) + FR (FF)) results in better performance across most testing sets compared to training on FS (FF) or FR (FF) individually. This suggests that combining face-swapping and face-reenactment datasets for training improves the generalization capabilities of the I3D method. **(2)** Among the CLIP-base detection methods, training on DF40 (FF) leads to better performance across most testing sets compared to training on FS (FF), FR (FF), or EFS (FF) individually. This indicates that incorporating diverse deepfake techniques in the training set improves the generalization capabilities of the CLIP-base method. **(3)** The same trend is observed for the CLIP-large detection methods, where training on DF40 (FF) results in better performance across most testing sets compared to training on FS (FF), FR (FF), or EFS (FF) individually. This further supports the importance of training on diverse deepfake techniques to enhance the generalization capabilities of the detection methods. **(4)** The CLIP-large detection method generally outperforms the CLIP-base method across most testing sets when trained on the same training sets. This suggests that the larger model size of CLIP-large contributes to better deepfake detection performance.

### A.5.4 Comparative Analysis of CLIP-base and CLIP-large for Deepfake Detection

**t-SNE visualization:** Through t-SNE visualizations, we observe that the distribution of real images (blue points) for CLIP-large is significantly more compact than CLIP-base, where the points are more loosely scattered. This tight clustering among the blue points for CLIP-large indicates that it can more effectively capture the finer details and specific characteristics of different real images. Additionally, CLIP-large demonstrates a superior ability to learn the distinct features of various datasets. When performing dataset classification, we notice that the decision boundaries between different real domains and between real and fake domains can be orthogonal. This orthogonality

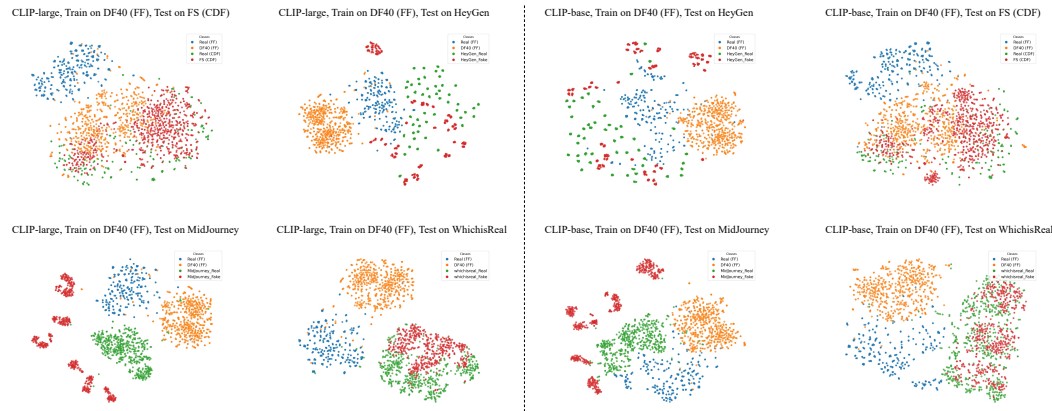

Figure 16: t-SNE visualizations for comparing CLIP-base and CLIP-large. We show that CLIP-large can learn more robust real-people features, thereby generalizing well in unseen forgeries.

is achieved without the need for explicitly adding a loss to align these boundaries, as suggested in the literature [70]. This implies that CLIP-large inherently distinguishes between domain-specific and domain-agnostic features, potentially due to insufficient real data, which makes it challenging to learn truly common features. Instead of attempting to decouple these features explicitly, CLIP-large identifies which features are domain-independent more efficiently.

**Logits Distribution Analysis:** The logits distributions provide further insights into the superior performance of CLIP-large. The logits for real images learned by CLIP-large are more concentrated and "tall and narrow," whereas those for CLIP-base are more "short and wide." This indicates that CLIP-large has a better grasp of the characteristics of real images, which enhances its ability to detect fakes. As evidenced by the reduced overlap between the real and fake logits distributions, CLIP-large's deep understanding of real images translates into a clearer differentiation between real and fake, improving detection accuracy. This suggests that better comprehending real images directly contributes to an improved understanding and detection of fake images.

Overall, the analysis highlights that CLIP-large outperforms CLIP-base in generalization for deepfake detection due to its superior feature learning capabilities. The compact clustering in t-SNE visuals and the focused logits distributions underscore its ability to distinguish between real and fake images more effectively. This enhanced understanding allows it to define more precise and orthogonal decision boundaries, leading to improved performance in detecting deepfakes across diverse datasets.

### A.5.5 Analysis of "Anomalous Values" in Experiments

In our previous experimental results summarized in Tab. 5 (protocol-3), we observed an anomalous outcome where the Xception model, trained on FF (FR), achieved an AUC of less than 0.2 on the StyleCLIP dataset. Given that an AUC of 0.5 indicates performance by chance, this result can be particularly perplexing. Here, we would like to delve deeper and clarify this observation in detail, as follows:

**Result explanation:** A very low prediction logits of the fake class make the 0.006 AUC possible: Conceptually, AUC can be formulated as $AUC = P(P_p > P_n)$, where $P_p$ and $P_n$ represent the probabilities of positive (fake, labeled as 1) and negative (real, labeled as 0) samples, respectively. In our case, the AUC of 0.006 can be attributed to the model assigning very low logits (*e.g.,* 10-7) to the fake class compared to the real class (*e.g.,* 10-2), as evidenced by the logit distribution in Fig. 17. According to the AUC formulation, when $P_p$ (logits for fake samples) is predominantly lower than $P_n$ (logits for real samples), the AUC approaches 0. Intuitively, this result indicates that the model perceives the fake test samples (in this case, StyleCLIP) as more likely to be real than the actual real samples. As a result, the ROC curve lies close to the x-axis, leading to an AUC of 0.006.

Xception, Train on FR (FF), Test on StyleCLIP

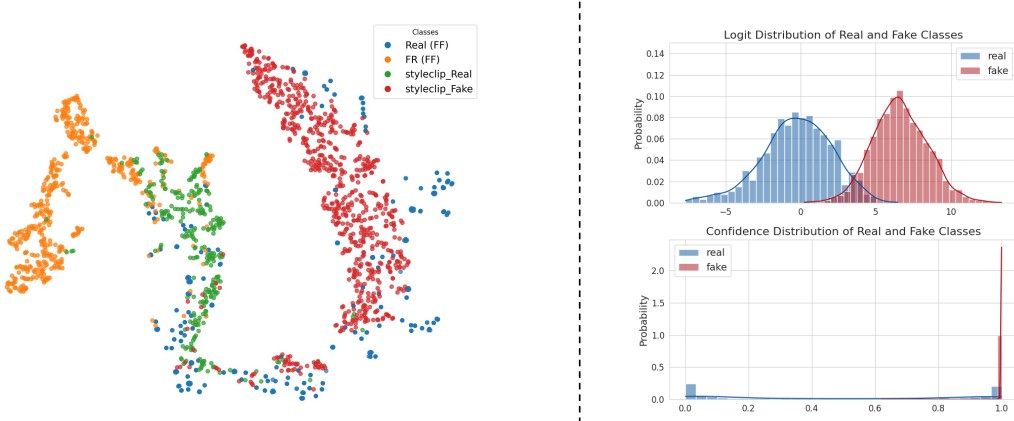

Figure 17: t-SNE visualization and logits analysis for analyzing the model, which is trained on FR (FF) and tested on StyleCLIP, achieving only about 0.2 AUC. See the text for our analysis and more details.

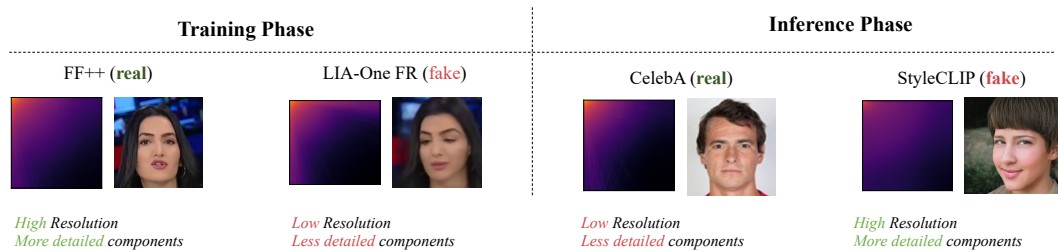

Figure 18: Frequency visualizations (std) of both the training and testing phases, for real and fake, respectively.

**Underlying reason:** Unaligned resolution between the real and fake classes introduces a model bias: To understand the underlying reason behind it, we provide further analysis from the resolution perspective. Specifically, we observe that the training real data (FF++) obviously contains more high-frequency components (higher resolution) compared to the training fake data (FR). This noticeable resolution gap between real and fake data could be inevitably captured by the model [56], leading the model to potentially adopt a trivial solution: "Low resolution implies fake." In contrast, during testing, the fake data (StyleCLIP) contains significantly more high-frequency components (higher resolution) than the testing real data used for testing. We provide a frequency visualization as evidence to validate this claim. As shown in Fig. 18, the testing fake (StyleCLIP) contains much more high-frequency details than the testing real. This bias leads the model to make an inconsistent decision, resulting in the anomaly in AUC. In our case, intuitively, since the biased model "observes" StyleCLIP carrying more high-frequency details with higher resolution than the real, the model then "thinks" StyleCLIP is more real than the actual real, eventually resulting in the 0.006 AUC.

### A.5.6 Comparison of Video Model and Image Model

Here, we would like to provide a deeper analysis comparing a video method (I3D) and an image method (Xception) using confidence distribution analysis.

Specifically, we obtain the prediction logits for each testing image from both models and apply the softmax function to obtain the confidence scores (ranging from 0 to 1). The visualization results of the confidence distribution for Xception and I3D can be seen in Fig. 19. As illustrated, we observe that I3D exhibits less uncertainty when making decisions for detection, while Xception shows higher uncertainty. This may suggest that there is limited discriminative information for Xception to perform

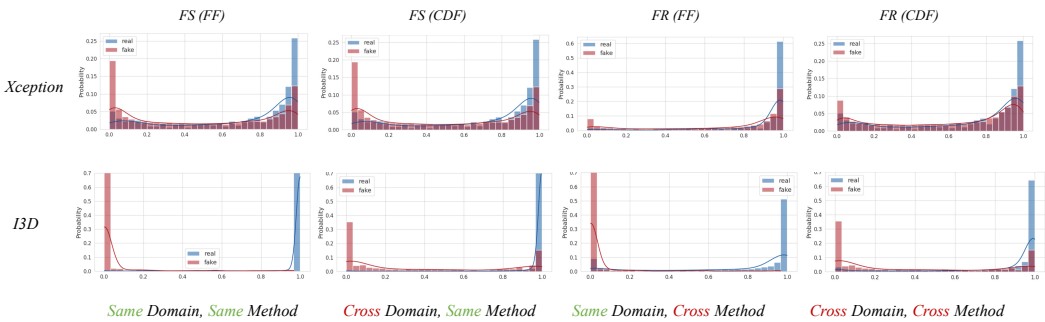

Figure 19: Confidence distribution comparison between Xception (image model) and I3D (video model). We show that I3D has less uncertainty in making real/fake classification decisions, while Xception has higher uncertainty.

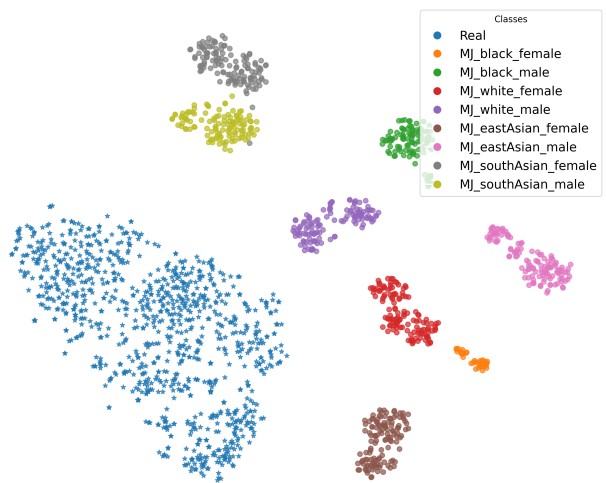

Figure 20: t-SNE visualization for illustrating the detection results of CLIP-large on the MidJourney dataset. We show that (fake) people of different races and genders can group into different clusters, highlighting that CLIP-large has already learned rich information about the real people's attributions.

detection based on a single frame. In contrast, the temporal information (many frames within a video) provides more informative clues for I3D's detection. This could be attributed to the fact that most (advanced) FS and FR methods cannot ensure perfect consistency at the temporal level, but they can maintain high realism at the image level, making detection more challenging for image-based detectors.

### A.5.7 Visualizations of the Real People Features.

Here, we explore which types of people tend to cluster in the MJ dataset and show the reasons behind this clustering. In Fig. 7 of our main paper (the result of CLIP large on MidJourney), we indeed notice that different clusters of fake images tend to form distinct groups rather than grouping together. This is an interesting observation, and we provide further exploration and analysis of this phenomenon:

**First**, we create fake generated data for MidJourney using different demographic compositions. We select people from different races and genders, including four races (white, black, east Asian, and south Asian) for both males and females, resulting in eight types in total. **Second**, we label all eight types with different labels. Along with the real label (real data is randomly selected from FFHQ), we have nine labels in total. We use t-SNE for visualization. Results in Fig. 20 show that the clusters in t-SNE correspond to different attributes/demographics. We observe that people with different races and genders are implicitly separated by the CLIP-large model. This indicates that CLIP-large contains rich real-world information from its large-scale pre-training, making CLIP-large able to have

accurate representations of races and genders. In contrast, Results in Fig. 7 (the result of Xception on MidJourney) shows that Xception cannot separate people of different races or genders. This suggests that Xception might only learn forgery-specific low-level signals without the ability to learn semantic facial attributes.

### A.5.8 Artifacts of Deepfake Forgeries in Frequency

**Brief Introduction:** Inspired by [77], we adopt a similar approach to visualize the average frequency spectra of each dataset. The purpose is to examine the artifacts generated by deepfake forgeries. Our methodology involves computing various statistics for image datasets, either in grayscale or across each color channel separately, and visualizing these statistics using heatmaps. We use two metrics to evaluate the results: *mean* and *std*. (**1**) The mean represents the average value of the DCT coefficients across the images in the dataset. A higher mean value means that, on average, the images are brighter or have more detailed features. For deepfake images, this might indicate that the synthetic process adds extra brightness or emphasizes certain facial details. On the other hand, a lower mean value suggests that the images are generally darker or smoother, possibly missing some fine details. By comparing the mean values of deepfake images to those of real images, we can see how well the fake images replicate the overall brightness and texture of real human faces. (**2**) The standard deviation (std) measures the spread or variability of the DCT coefficients around the mean. A higher standard deviation means more variation in the textures and details within the images. For deepfakes, this could point to inconsistencies or artifacts from the generation process, making the images easier to spot as fake. Conversely, a lower standard deviation indicates that the images are more similar, which might suggest a more uniform generation process but could also mean the images lack the natural diversity seen in real faces. We can assess how consistent and realistic the deepfake images are compared to real human faces by looking at the standard deviation.

**Mean Analysis:** We observe that many GAN-generated images, such as those from StyleGAN, often exhibit a checkerboard or grid-like pattern of bright spots on the mean heatmap. This phenomenon occurs because of the upsampling methods used in the GAN architecture, which can introduce such artifacts during the image generation process. These patterns are typically a result of the transposed convolution operations, which can cause uneven overlaps and thus create these visible grid artifacts. Moreover, it appears that different forgery methods can share similar patterns. For instance, face-swapping techniques like SimSwap also exhibit these GAN-like checkerboard artifacts. This similarity indicates that these methods generate the entire image using AI without a blending process. This contrasts with other Deepfake methods, such as those used in Celeb-DF, which involve blending and thus do not produce the same checkerboard artifacts.

**Std Analysis:** The standard deviation heatmaps reveal additional artifacts. For example, methods like DDPM and StarGAN also display checkerboard artifacts in the standard deviation maps. Interestingly, while StyleGAN exhibits these checkerboard patterns in the mean heatmaps, it does not show similar artifacts in the standard deviation maps. This discrepancy suggests that while the upsampling process affects StyleGAN's mean values, the overall variability or consistency across multiple generated images is not influenced similarly. This could imply that StyleGAN maintains a certain level of uniformity regarding pixel variance, even though the generation process distorts the mean values. Additionally, we observe radial artifacts in the standard deviation heat maps through methods such as HeyGen, DeepFaceLab, and DaGAN. These radial patterns indicate a different type of artifact, possibly resulting from how these methods handle image transformations and blending processes. The presence of these radial artifacts in the standard deviation maps suggests inconsistencies in the generation process, where certain regions of the images exhibit more variability than others.

### A.6 Dataset Publication

### A.6.1 Hosting Platform & Links

We host our DF40 dataset on GitHub (https://github.com/YZY-stack/DF40). Our project page is https://yzy-stack.github.io/homepage_for_df40/. All the codes, datasets, and checkpoints are released at the above links. Our dataset with documentation and associated codes (for reproduction of our experimental results) is maintained under the Creative Commons NonCommercial

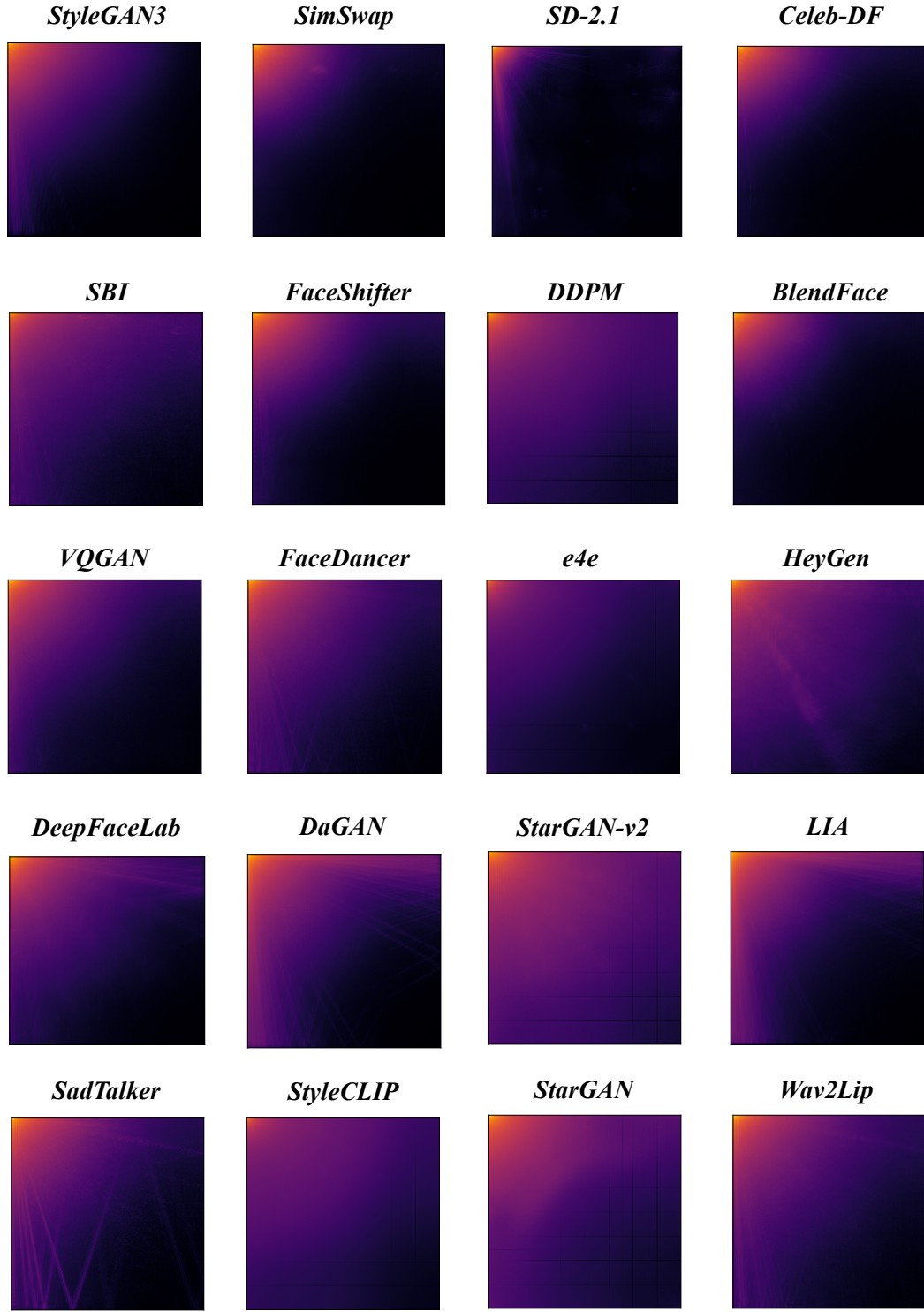

Figure 21: Std of the frequency analysis for visualizing different deepfake methods from the frequency domain.

### A.6.2 Controlled Access:

To control the random distribution, we have already implemented a controlled access system (a Google form) for the DF40 dataset, where users may require authorization or meet specific prerequisites to gain access (see https://docs.google.com/forms/d/e/1FAIpQLSedrbobvH3JtCZR7wDzH4S9olikmwL67x7AawckgRXKF_T2Mg/viewform). This approach, similar to protocols used with previous deepfake datasets like FF++ and Celeb-DF, can help restrict indiscriminate access. Additionally, acknowledging the evolving nature of threats, one of our strategies is the continuous refinement and adaptation of the tools and detection methods associated with the DF40 dataset. A static dataset is more vulnerable to exploitation; therefore, keeping our platform adaptive is essential.

### A.6.3 Discrimination, Bias, and Fairness:

Please note that the DF40 fake data is created entirely based on the real data, meaning that the demographic composition of the datasets is the same as the original datasets. We chose this setting to align with previous works. Specifically, most prior deepfake detection studies train their models on four deepfake methods from FF++ and test them on one fake method from CDF. This consideration motivated us to choose FF++ and CDF as the real data sources for generating corresponding fake data for alignment.

However, as shown in [74], these datasets might not be fair enough to create a completely unbiased detector without any explicit constraints and supervision. To address this potential concern, we have proposed several solutions to address the fairness issue, including: a. We have added the facial attribute annotations (such as race and gender) for each video on our GitHub page (github.com/YZY-stack/DF40/tree/main/Annotations_for_Facial_Attributes). By providing access to these attributes, users can develop fairer and less biased deepfake detectors using our datasets; b. Most existing studies on fairness are based on the FF++ and CDF datasets, the same datasets we utilized for developing ours. This suggests that existing fairness methods can be also used in our dataset. In summary, the existing fairness studies such as [31] that enforce fairness constraints based on the annotations of each video in FF++, can directly apply their methods to our dataset for improved fairness.

