# OpenReview forum: "DF40: Toward Next-Generation Deepfake Detection"
_NeurIPS.cc/2024/Datasets_and_Benchmarks_Track — NeurIPS 2024 Track Datasets and Benchmarks Poster_

### Official Review · Reviewer_RsPZ · 2024-07-08
**Comments of the paper: DF40: Toward Next-Generation Deepfake Detection**

**Rating:** 9
**Confidence:** 5
**Correctness:** I do not see any correctness problems…
**Clarity:** This paper is well-written.

**Review:**

Overall, this paper makes a positive contribution to the field. It offers insightful findings and raises important open questions. The content is clear, original, and significant. However, there are some limitations, primarily concerning the clarification of details about the dataset. These include the dataset's publication plan, the different dataset variants, and detailed results of their analysis. Additionally, the paper could be further improved by incorporating more video detectors in the deepfake domain.

**Strengths:**

1. Comprehensive Benchmarking: The DF40 dataset introduces a new benchmark that includes 40 distinct deepfake techniques, covering face-swapping, face-reenactment, entire face synthesis, and face editing. This coverage ensures a more thorough evaluation of deepfake detection methods and promotes the development of more generalizable detectors. Also, DF40 utilizes 4 standard evaluation protocols and conducts over 2,000 evaluations. This extensive analysis leads to 8 new findings, providing further understanding of the strengths and weaknesses of current detection methods and opening up several research questions to inspire future work in the field.

2. Inclusion of SOTA Techniques and Open-Set Evaluation: The dataset includes the latest and most advanced deepfake generation techniques, such as DiT, PixArt-a, and popular software like DeepFaceLab and HeyGen. By incorporating these SOTA methods, DF40 ensures that the detection algorithms are tested against the most realistic and sophisticated forgeries, making the evaluations more relevant to real-world scenarios. Also, DF40 introduces an open-set evaluation protocol to test models on unknown forgery methods and domains. This simulates real-world conditions where new and unseen deepfake techniques continuously emerge.

3. Analysis of Data and Method Impact Toward Generalization: The research provides a detailed analysis of how different forgery methods and data domains contribute to the discriminative forgery artifacts, influencing the model's generalization performance. This analysis, supported by causal graphs and t-SNE visualizations, offers valuable insights into the factors affecting deepfake detection accuracy and helps guide the development of more effective and generalizable detection algorithms.

**Additional Feedback:**

I look forward to the author's response to my concern. If the response is satisfactory, I will consider further improving my rating. I would also like to discuss this good work in more detail with the authors.

**Documentation:**

I do not find detailed documentation for the dataset on the provided GitHub. I hope the author can provide more comprehensive instructions on how to use the proposed dataset.

**Ethics:**

I do not find any obvious and serious ethic problems in this paper.

**Limitations:**

1. Clarification of Different Face-Swapping Methods: There is a lack of clarification distinguishing various face-swapping methods, specifically identifying which belong to the "DF family" and which do not. This distinction is missing in Table 2. Additionally, there needs to be an analysis of the generalization performance across DF-family and non-DF-family face-swapping methods.

2. About real-people image quality: The quality of images in the real-person dataset (FF and CDF) is too blurry, which may impact the results. Consider using a dataset with clearer facial images. The effect of varying image clarity on the performance of face-swapping methods should be evaluated, as the clarity of real-person data can significantly influence the outcomes.

3. Impact of Super-Resolution: The influence of super-resolution on fake images should be investigated. Assess whether super-resolution significantly affects the detection and analysis of fake images.

4. Further Visualization and Analysis of Real-Person Data: More visualization and analysis of real-person data are needed. For instance, investigate which real people tend to cluster in the MJ dataset and the reasons behind this clustering. Additionally, analyze if different races are grouped together and examine the clustering patterns in the HeyGen dataset.

5. Analysis of Forgery Regions: The DF40 dataset provides deepfake data with varying forgery regions (e.g., whole image synthesis, face-focused, mouth-only). Evaluate the impact of different forgery regions on detection performance. Determine whether methods trained on face forgery data can be effectively utilized to detect mouth forgery data.

6. Detailed explanation about the dataset publication: Clarify the publication details of the DF40 dataset. Currently, there is no visible release plan. Provide a detailed plan for making the DF40 dataset open-source.

7. Clear clarification about their dataset variants: There are multiple variants of the proposed dataset, such as DF40 (FF), DF40 (CDF), FS (FF), and EFS (FF). A table listing these variants should be included for better clarification and understanding.

8. About the video detector: Results of the I3D video detector should be included and compared with the results of image detectors. This comparison will provide a comprehensive evaluation of the performance of video detectors versus image detectors.

**Opportunities For Improvement:**

See the Limitations part. I specifically hope the author can provide (1) further analysis for image quality and real-people distribution; (2) further clarification about the dataset publication and variants. Furthermore, the implementation of more video deepfake detectors would be a bonus.

**Relation To Prior Work:**

Yes, the authors have shown a clear comparison in Table 1 to compare with previous works.

**Summary And Contributions:**

The paper introduces DF40, a new deepfake detection dataset featuring 40 techniques for face synthesis and generation. The paper is well-written, with a clear motivation and background introduction. I mostly agree with the perspectives presented throughout the article. In particular, the proposed evaluation protocols and new insights will be highly valuable for future deepfake detection research.

---

> ### Author Rebuttal · Authors · 2024-08-24
>
> We sincerely appreciate Reviewer RsPZ for their insightful suggestion and valuable comments. We are encouraged by their recognition of our benchmark and important contribution to the field. Below, we address the reviewer's concerns in detail, point by point.
>
> ---
>
> > Exploring which types of people tend to cluster in the MJ dataset and the reasons behind this clustering.
>
> **Answer:** Thank you for your insightful question and detailed observation. In Figure 7 (the result of CLIP large on MidJourney), we indeed notice that different clusters of fake images tend to form distinct groups rather than grouping together. This is an interesting observation, and we provide further exploration and analysis of this phenomenon:
> - **First**, we create fake generated data for MidJourney using different demographic compositions. We select people from different races and genders, including four races (white, black, east Asian, and south Asian) for both males and females, resulting in eight types in total.
> - **Second**, we label all eight types with different labels. Along with the real label (real data is randomly selected from FFHQ), we have nine labels in total. We use t-SNE for visualization.
>   - Results in **Figure 1 of the attached PDF** show that the clusters in t-SNE correspond to different attributes/demographics. We observe that people with different races and genders are implicitly separated by the CLIP-large model. This indicates that CLIP-large contains rich real-world information from its large-scale pre-training, making CLIP-large able to have accurate representations of races and genders.
> - In contrast, Figure 7 (the result of **Xception on MidJourney**) shows that Xception cannot separate people of different races or genders. This suggests that Xception might only learn forgery-specific low-level signals without the ability to learn semantic facial attributes.
>
> > Lack analysis of the impact of different forgery regions on detection performance.
>
> **Answer:** Thanks for your question. We design the following experiments to explore the impact of different forgery regions on detection performance:
> - We create a self-reconstruction image (SRI) for each original real image, where the SRI images act as fakes with a global forgery pattern (entire image synthesis). We then perform a blending operation to replace the reconstructed face with the original face, creating a localized fake (L-SRI), similar to other typical localized face forgery methods. We consider five common types of fake regions: whole, face, mouth, nose, and eyes.
> - The results in **Table 1 of the attached PDF** below quantitatively demonstrate the significance of the fake region in detection.
>
>
> > The impact of real image quality and resolution on the detection results.
>
> **Answer:** Thank you for your detailed observation. The resolution of images can significantly impact detection results. Specifically, the high resolution adds more high-frequency details to an image, which may lead to model bias in detection (see **Figure 2 of Rebuttal PDF to reviewer H9dn**).
> - To illustrate, we observe that the training real data (FF++) contains more high-frequency components (higher resolution) compared to the training fake data (FR). This noticeable resolution gap between real and fake data could be inadvertently captured by the model, causing it to adopt a trivial solution: "Low resolution implies fake." Conversely, during testing, the fake data (StyleCLIP) contains significantly more high-frequency components (higher resolution) than the real data used for testing. This bias leads the model to make inconsistent decisions, resulting in a low AUC.
>
>
> > Results of the I3D video detector should be compared with the results of image detectors.
>
> **Answer:** Thank you for your suggestion. We would like to provide a deeper analysis comparing a video method (I3D) and an image method (Xception) using confidence distribution analysis.
> - Specifically, we obtain the prediction logits for each testing image from both models and apply the softmax function to obtain the confidence scores (ranging from 0 to 1). The visualization results of the confidence distribution for Xception and I3D can be seen in **Figure 2 of the attached PDF**.
> - As illustrated, we observe that I3D exhibits **less uncertainty** when making decisions for detection, while Xception shows higher uncertainty. This may suggest that there is limited discriminative information for Xception to perform detection based on a single frame. In contrast, the temporal information (many frames within a video) provides more informative clues for I3D's detection. This could be attributed to the fact that most (advanced) FS and FR methods cannot ensure perfect consistency at the temporal level, but they can maintain high realism at the image level, making detection more challenging for image-based detectors.
>
>
> > Exploring the impact of super-resolution on fake images.
>
> **Answer:** Thank you for your insightful suggestion. Following your recommendation, we use GFPGAN to perform super-resolution on the self-reconstruction images (SRI). Results in **Table 2 of the attached PDF** show that super-resolution has a **significant impact** on fake images, particularly for EFS and FE methods. As demonstrated in the table, models trained on FS and EFS exhibit more than a 20% point improvement for the SRI after applying super-resolution. This suggests that super-resolution operations may introduce noticeable generative artifacts that can be detected by models trained on EFS and FE.
>
>
> > Clear clarification about the dataset variants.
>
> **Answer:** Thanks for your kind mention. We will list a table to summarize the different variants of DF40 used in this paper, for better clarity.

---

> > ### Comment · Reviewer_RsPZ · 2024-08-29
> >
> > I appreciate the authors detailed responses, which well addressed my concerns. I have promoted my rating from 8 to 9.

---

### Official Review · Reviewer_1Hg1 · 2024-07-09
**This paper gives some important contributions to deepfake detection field, but some limitations should be carefuuly adressed.**

**Rating:** 9
**Confidence:** 5
**Correctness:** All the claims and evaluation design …
**Clarity:** This work is well-written and easy to…

**Review:**

⁃ pros (details can be seen in Strengths part):
(1) quality: the overall quality is relatively very high in the whole field. The contribution (see Strengths part) is good. The writing style and paper structure is well-designed. The evaluation and analysis in both manuscript and supplementary seem to be comprehensive. The paper provides insightful findings and raises important open questions.
(2) clarity: the paper is well-written, with clear motivation, background introduction, and analysis. It is easy to follow from start to end, with engaging metaphors that enhance readability.
(3) originality and significance: this paper provides a new benchmarking dataset with high originality.
⁃ cons (details can be seen in Limitations part):
(1) lack of more analysis for video methods: the paper only considers I3D for evaluation, lacking a broader analysis with more video detectors.
(2) lack of evaluations for AIGC-based detectors: there are no results for detectors trained solely on AIGC data tested on DF40.
(3) More detailed description for the dataset and dataset publication: the paper lacks detailed discussion on the dataset's release and access.
(4) ethic problem: there is no discussion of potential ethical issues, which is important for a face dataset work.

**Strengths:**

⁃Datasets: DF40 offers following advantages over previous datasets: (1) diversity: DF40 includes 40 distinct forgery methods, providing greater diversity than previous datasets; (2) realism: DF40 uses the latest methods, including DiT and popular software like DeepFaceLab and HeyGen, ensuring state-of-the-art deepfake detection; (3) alignment: 31 methods are used to generate deepfakes on FF++ and CDF domains, ensuring aligned data domains for evaluation.
⁃Evaluations and Protocols: (1) The authors provide relatively comprehensive evaluations based on the proposed dataset. Some are new to the field, e.g., train on face-swapping and testing on entire face synthesis. It is true that most existing works perform training and testing solely on face-swapping (train on FF and test on CDF/DFD/DFDC); (2) the authors provide evaluations on aligned settings (e.g., same domain but different forgery).
⁃Analysis and Findings: (1) Logits and Confidence Analysis: This new tool can analyze the distribution for both real and fake classes, offering insights for future work; (2) Novel Findings: Several new and reasonable findings are presented, contributing to the field.
⁃Writing Style: The paper is engaging and well-structured, with good logic and attractive metaphors.

**Additional Feedback:**

Overall, this is definitely a high-quality paper in the deepfake detection domain with good writing, clear motivation, and thorough evaluation and analysis. The proposed dataset is crucial for developing future deepfake detectors. Addressing the outlined limitations will further enhance the paper.

**Documentation:**

A more detailed plan for dataset publication is not sufficient (see Limitations.6)

**Ethics:**

Concerns about safety, security, discrimination, bias, and fairness are raised. A clear clarification of these ethical issues is suggested.

**Limitations:**

⁃ About the ethic: The paper lacks a detailed discussion on potential ethical issues. Adding a paragraph on this topic is recommended.
⁃ The term "Sub-Types" in Table 1 is not appropriate. "Highlight" would be a more suitable term.
⁃ More detailed description for the dataset: (1) In Line 105-114, the authors only provide the description for the scope and scale of dataset, lacking more discussion about the training and testing data split. For instance, do you have validation data? Can we use the DF40 (CDF) for training? (2) for the original data, why use these datasets to serve as the real data? There is lack more solid motivations.
⁃ Lack of more analysis for video-level detectors: it is true that the authors provide comprehensive analysis for image-based detectors but lack the analysis for video-level methods. It is needed to implement more video-level detectors (not limited to I3D) and combine them to analysis using DF40. Also, the more insightful analysis at the video level is another bonus.
⁃ Lack of evaluations for AIGC-based detectors: the authors use 7 detectors from face forgery detection field but miss the detectors from AIGC detection (like DIRE). The absence of evaluations for AIGC-based detectors (like DIRE) raises questions about their generalizability to face forgeries.
⁃ Lack of more detailed information for the data publication: the authors may provide a dataset form for the users to access their data, ensuring the usage of their dataset. Also, the authors should write more detailed instructions in their GitHub repo. The current one lacks instruction of how to use them for training and testing, and re-producing these results reported in their paper.
⁃ More detailed implementation for fine-tuning the EFS data should be discussed.
⁃ Minor typos:
(1) Changing the “Total number of EFS: 5” to “Total number of FE: 5” in Figure 1.
(2) the structure order of Section 6 (Supplementary) is different from the actual orders in the supplementary.

**Opportunities For Improvement:**

⁃ deeper analysis for video-level detectors.
⁃ more evaluations for AIGC-based detectors.
⁃ more detailed clarification for the dataset information and publication.
⁃ (extra bonus) Exploring the combination of DF40 (FF) with the original FF++ dataset for improved training.
⁃ (extra bonus) Detailed analysis of the differences between using real faces and real data for training.

**Relation To Prior Work:**

Although I can see the forgery diversity, scale and realism of the proposed datasets compared to previous works, there is no clear comparison of the proposed work over previous work. It is suggested to give a clear comparison in the manuscript.

**Summary And Contributions:**

This paper introduces DF40, a new deepfake detection dataset comprising 40 deepfake techniques for face synthesis and generation. The authors propose 4 evaluation protocols utilizing 8 different detection models, generating over 2000 evaluation results. These results are thoroughly visualized and analyzed, revealing several findings and open questions. Based on the quality and contribution, I believe this work can be the top 15% of accepted papers in the deepfake detection field.

---

> ### Author Rebuttal · Authors · 2024-08-22
>
> We sincerely appreciate Reviewer 1Hg1 for their valuable comments and are encouraged by their recognition of our novelty and contributions. We also acknowledge one of the biggest concerns of reviewer 1Hg1 is the **dataset publication and detailed information**. Below, we address them in detail, point-by-point.
>
> ---
>
> > The paper lacks a detailed discussion on potential ethical issues.
>
> **Answer:** Thanks. We are grateful to the reviewer for highlighting this significant concern. We aim to address it with a systematic and comprehensive response:
>
> 1. **Identification of the Ethical Concern:** The primary ethical issue, as highlighted, is the potential for malicious actors to misuse our DF40 dataset to develop advanced deepfake detectors and refine their deepfake generators, ultimately enabling them to evade state-of-the-art detection tools.
>     * **Inherent challenge with deepfake datasets:** The core dilemma faced by any deepfake dataset created for positive purposes, including our DF40 dataset, is its dual potential. While such datasets are crucial for advancing research and developing effective countermeasures against deepfakes, their transparency can also inadvertently offer a roadmap for malicious actors.
> 2. **Potential Solutions and Forward Path:**
>     * **Controlled Access:** We have already implemented a controlled access system (a **Google form**) for the DF40 dataset, where users may require authorization or meet specific prerequisites to gain access (see `https://docs.google.com/forms/d/e/1FAIpQLSedrbobvH3JtCZR7wDzH4S9olikmwL67x7AawckgRXKF_T2Mg/viewform`). This approach, similar to protocols used with previous deepfake datasets like FF++ and Celeb-DF, can help restrict indiscriminate access.
>     * **Dynamic Evolution of DF40:** Acknowledging the evolving nature of threats, one of our strategies is the continuous refinement and adaptation of the tools and detection methods associated with the DF40 dataset. A static dataset is more vulnerable to exploitation; therefore, keeping our platform adaptive is essential.
>
> In conclusion, we sincerely appreciate the reviewer's insights and agree on the importance of addressing these concerns. While no system can be entirely foolproof, we aim to develop the DF40 dataset into a resilient and valuable tool in the ongoing fight against deepfakes, while remaining aware of its potential vulnerabilities. We plan to include this discussion in the revised supplementary materials.
>
>
>
> > More details about the training and testing data split.
>
> **Answer:** Thanks for your kind mention. Actually, we have provided detailed information for the split, **as shown in Lines 639-647 of the supplementary**. We would like to clarify the training and testing split with a more general explanation:
> - In **Protocol-1**, **Protocol-2**, and **Protocol-4**, we use "XX" (FF) for training, and "XX" (FF) or "XX" (CDF) for testing. This means that CDF is only used for testing not training. This aligns with the previously widely used setting, which is training on FF++ and testing on CDF, allowing previous works to seamlessly and conveniently extend their evaluation on our evaluation protocols.
> - In **Protocol-3**, we consider a more challenging (black-box) setting: testing the models on unknown data domains and fake methods. In this setting, we use FF++ domains for training and do not use CDF for testing.
>
> We will add these details to our revision. Thanks for your kind mention.
>
>
> > For the original data, why use these datasets to serve as the real data?
>
> **Answer:** Thanks for your kind mention. Actually, we have already provided a detailed introduction (e.g., leading institutions, dataset links, and official copyright issues) of the original data used in our dataset, as shown in Section 2.4 of the supplementary. The key motivation for using FF++ and CDF to create aligned fake data is their wide use in many previous deepfake detection works, as also highlighted in ref[1].
>
> ref[1]: DeepfakeBench, NIPS 2023 D&B track.
>
>
> > More detailed implementation for fine-tuning the EFS data should be discussed.
>
> **Answer:** Thanks for your kind mention. Please note the fine-tuning details for each EFS method can vary. In Algorithm 3 of the supplementary, we provide a general understanding of the fine-tuning process. Following your suggestion, we will add the detailed process for each method in the revision.
>
>
> > Lack of more detailed information for the data publication.
>
> **Answer:** Thanks for your kind mention. Data publication is an important part of a dataset work. In this paper, we have provided detailed information for this in **Lines 956-974 of the supplementary**. Here, we would like to further clarify that:
> - **License and distribution:** Our dataset with documentation and associated codes (for the reproduction of our experimental results) is maintained under the **CC BY-NC 4.0 license** (`https://creativecommons.org/licenses/by-nc/4.0/`). The term of use should be **only for research and educational purposes**, not for any commercial purposes.
>   - For restricting and controlling the distribution, we have already built a **Google form** that the users must fill to access our dataset (see the form from the link: `https://docs.google.com/forms/d/e/1FAIpQLSedrbobvH3JtCZR7wDzH4S9olikmwL67x7AawckgRXKF_T2Mg/viewform`). Users may require authorization or meet specific prerequisites to gain access.
> - **Homepage:** Furthermore, we have already made our dataset available on an official website with detailed instructions and official download links (see `https://yzy-stack.github.io/homepage_for_df40/`).
>
> > Minor typos: (1) Changing the “Total number of EFS: 5” to “Total number of FE: 5” in Figure 1. (2) the structure order of Section 6 (Supplementary) is different from the actual orders in the supplementary.
>
> **Answer:** Thanks for your kind mention. We have clarified all of them in the revision.

---

> > ### Comment · Reviewer_1Hg1 · 2024-08-30
> >
> > The authors' detailed response has fully addressed my concerns, and my score of 9 shows my strong support for this paper.

---

### Official Review · Reviewer_H9dn · 2024-07-23
**Clear contribution with comprehensive experiments**

**Rating:** 6
**Confidence:** 5
**Correctness:** The proposed Dataset, DF40, is constr…

**Review:**

Overall, the proposed paper is high quality, and the arguments are clearly stated. Please see “Opportunities for improvement,” “Limitations,” “Correctness,” “Clarity,” “Documentation,” and “Additional feedback.” below for more details.

**Strengths:**

(1) The proposed benchmark compensates for the limitations (size, diversity, and composition) of existing deepfake datasets.
(2) Results can contribute to the future development of deepfake research by presenting valuable findings and setting new research questions.
(3) Through comprehensive experiments, the author(s) demonstrate the superiority of DF40 and the correlation between existing forgery methods and data domains.
(4) The author(s) address possible ethical issues by specifying ethical considerations.

**Additional Feedback:**

[Questions]
- line 40~50: In the existing face forgery detection methods, there are approaches such as frequency artifact-based in addition to the blending-based detector; why focus only on the blending-based detector?
- line 141~143: When fine-tuning the generative model, how and for what purpose is it tuned, e.g., to produce results similar to real data in FF++ and CDF when a target is input?
- line 249: What is meant by 'orthogonal'?
- Why is FE not considered in the [4.2 Evaluation] section?

[Suggestions for improvement]
- Supplementary material line 480, 487: The numbering of tables and figures is missing.

**Clarity:**

Overall, it's well-written but lacks in the following areas.
(1) The figures are mentioned in the text in a different order than they are numbered. Numbering them in the same order as the flow of the text would help the reader understand them better.
(2) Lines 198~200: 'The results ~ StyleClip.' -> It would be better to mention the table (i.e., Table 5) that shows the results covered in those lines.
(3) A few typo errors should be fixed.
- line 201: that forgery (i) artifacts -> that (i) forgery artifacts
- line 259~260: 'can we ~ and FE?' -> italic

**Documentation:**

The authors provide details of DF40 in the Supplementary material, as well as plans for dataset distribution and licensing.

**Ethics:**

No.

**Limitations:**

(1) References for some claims should be provided.
- Line 37: 'into two sub-fields: ~ detection.' -> The division into two sub-fields needs to be clarified on what basis, and a reference should be added.
- lines 56-57: 'highly popular ~ real-world deepfakes;' -> Adding a reference to the fact that DeepFaceLab and HeyGen are indeed popular would strengthen the argument that they contain real-world data.
- line 72: 'popular ~ Roop' -> Same as above.
- line 112: 'widely used ~ DeepFaceLab.' -> Same as above.

(2) The most recent methodology for the baseline model utilized in our experiments was proposed in 2022, which is hardly the most recent methodology. Adding a newer methodology that is not used in DeepfakeBenchmark may improve the reliability of the experimental results.

(3) Detailed descriptions of the figures and tables presented in this paper should be added to facilitate readers' understanding.
- Table 2: Among the 9 'Test Only' data, CollabDiff and e4e are labeled as using FF++ and CDF as 'Real Data Source,' but it is unclear why they are categorized as 'unknown domain data.' (See Note-1)
- Figure 2: The notation is somewhat confusing and needs to be unified or accompanied by clarification (e.g., if \(\theta_e)) and \(\G) mean the same thing, it needs to be unified).
- Figure 2: Referred to as (a)-(d) in the text, Figure 2 needs to have (a)-(d) labeled, which can be confusing and needs to be appropriately labeled.
- Figure 6: Same as above.

**Opportunities For Improvement:**

Overall, the paper is well-written, but it could be improved by adding additional explanations for the following points.
- Table 3: 'Performance drop' shows that the FS-based results are relatively less different than those of FR and EFS. A more detailed explanation of why this is the case would help the reader understand the results.
- Table 5: Reasons for the AUC (0.006) result for Xception for StyleClip should be added: Achieving an AUC of 0.006 on a binary classification problem implies deficient performance, and an analysis of this result is needed.

**Relation To Prior Work:**

he authors clearly state the limitations of previous research related to this study and provide a rationale for how they addressed them from each perspective.

**Summary And Contributions:**

This paper proposed DF40, which consists of 40 distinct deepfake technologies. Through comprehensive experiments on seven representative deepfake detection models in four standard evaluation protocols (i.e., face-swapping (FS), face-reenactment (FR), entire face synthesis (EFS), and face editing (FE)) settings, the author(s) emphasize the need and importance of DF40. This paper also contributes to further deepfake research by presenting eight findings and three new research questions.

---

> ### Author Rebuttal · Authors · 2024-08-20
>
> We sincerely appreciate Reviewer H9dn for their detailed suggestions and are encouraged by their recognition of our work. Below, we address each of the reviewer's concerns in detail, point by point.
>
> ---
>
> ### ***Part-1: Opportunities For Improvement***
> > Table 3: 'Performance drop' shows that the FS-based results are relatively less different than those of FR and EFS. A more detailed explanation of why this is the case would help the reader understand the results.
>
> **Answer:** Thanks for your kind mention. We would like to provide a deeper analysis of why 'Performance drop' shows that the FS-based results are relatively less different than those of EFS. One reason is that **FS can include both localized and global forgeries, while EFS is limited to global forgeries, making EFS less diverse in scope compared to FS**.
>   -  Specifically, FS may contain fake regions within both the facial area and the entire image, whereas EFS involves forgeries that affect the entire image. This could simplify the fake patterns in EFS than those in FS. As a result, training a model on EFS might not capture all forgery patterns, especially the  *region-specific* fake artifacts present in FS.
>   - **We conduct two experiments to verify our claim:** ***(1) Quantitatively,*** we generate self-reconstruction images (SRI) for each original real image, where SRI images represent fakes with a global forgery pattern (entire image synthesis). We then perform a blending operation to replace the reconstructed face with the original face, creating a localized fake (L-SRI), similar to other typical localized face forgery methods. We consider five typical types of fake regions: whole, face, nose, eyes, and mouth. Results in **Table 1 of the attached PDF** quantitatively verify the significance of the fake region.
> ***(2) Qualitatively,*** we use t-SNE to visualize the latent distribution of 'real', 'whole-fake', and 'face-fake' images. The t-SNE results (**Figure 1 of the attached PDF**) show that a well-trained EFS detection model can easily distinguish the original SRI from real images. However, the localized version (L-SRI) appears much more similar to real images, making it more challenging to detect compared to the SRI. This experiment further confirms our claim, qualitatively.
>
>
> > Table 5: Reasons for the AUC (0.006) result for Xception for StyleClip should be added: Achieving an AUC of 0.006 on a binary classification problem implies deficient performance, and an analysis of this result is needed.
>
> **Answer:** Thanks for your very good question and observation. Actually, we have already noticed this 'anomaly value' during our experiment and conducted visualizations to investigate it further, as shown in **Figure 12 and Lines 933-955 of the supplementary.** Here, we would like to delve deeper and clarify this observation in detail, as follows:
> - **(*Result explanation*) A very low prediction logits of the fake class make the 0.006 AUC possible:** Conceptually, AUC can be formulated as $AUC=P(P_p>P_n)$, where $P_p$ and $P_n$ represent the probabilities of positive (fake, labeled as 1) and negative (real, labeled as 0) samples, respectively. In our case, the AUC of 0.006 can be attributed to the model assigning very low logits (e.g., 10-7) to the fake class compared to the real class (e.g., 10-2), as evidenced by the logit distribution in Figure 12 of the supplementary. According to the AUC formulation, when $P_p$ (logits for fake samples) is predominantly lower than $P_n$ (logits for real samples), the AUC approaches 0. Intuitively, this result indicates that the model perceives the fake test samples (in this case, StyleCLIP) as more likely to be real than the actual real samples. As a result, the ROC curve lies close to the x-axis, leading to an AUC of 0.006.
> - **(*Underlying reason*) Unaligned *resolution* between the real and fake classes introduces a model bias:** To understand the underlying reason behind it, we provide further analysis from the resolution perspective. Specifically, we observe that the training real data (FF++) obviously contains more high-frequency components (higher resolution) compared to the training fake data (FR). This noticeable resolution gap between real and fake data could be inevitably captured by the model (ref[1]), leading the model to potentially adopt a trivial solution: "*Low resolution implies fake.*" In contrast, during testing, the fake data (StyleCLIP) contains significantly more high-frequency components (higher resolution) than the testing real data used for testing. We provide a frequency visualization as evidence to validate this claim. As shown in **Figure 2 of the attached PDF**, the testing fake (StyleCLIP) contains much more high-frequency details than the testing real. This bias leads the model to make an inconsistent decision, resulting in the anomaly in AUC. In our case, intuitively, since the biased model "observes" StyleCLIP carrying more high-frequency details with higher resolution than the real, the model then "thinks" StyleCLIP is more real than the actual real, eventually resulting in the 0.006 AUC.
>
> *ref[1]: F3Net, ECCV 2020.*
>
> ---
>
> ### ***Part-2: Limitations & Clarity:***
> > References for some claims should be provided; Detailed descriptions of the figures and tables should be added.
>
> **Answer:** Thanks for the kind mention and detailed review. Following your suggestion, we have added appropriate references for each claim and detailed descriptions of figures and tables mentioned by the reviewer.
>
> > Adding a newer detection methodology for evaluation.
>
> **Answer:** Thanks. Following your suggestion, we have added one just-accepted ECCV 2024 detector (ref[2], provided by the author) for evaluation. Detailed results and discussion can be found in **Figure 3 of the attached PDF**.
>
> *ref[2]: Fake It till You Make It: Curricular Dynamic Forgery Augmentations towards General Deepfake Detection, ECCV 2024.*

---

> ### Author Response · Authors · 2024-08-20
> **Additional response by Authors**
>
> Due to the limited space, we provide an additional response to address the additional feedback from the reviewer H9dn.
>
> ---
>
> ### ***Part-3: Additional Feedback***
> > line 40~50: why focus only on the blending-based detector? What about the frequency-based?
>
> **Answer:** Thanks for your valuable comment.
> - **Reason explanation:** In Lines 40-50, we mainly use the blending-based detector as an *example* to illustrate our claim (in Line 39) that: *most existing detectors are limited to detecting specific types of deepfakes.* More importantly, in the field of face forgery detection, blending has been shown to be one of the most effective approaches and achieved SoTA performance on many existing deepfake datasets, as evidenced by ref[1,2]. By focusing on blending, we aim to point out that without our DF40 dataset, we might only see "the tip of the iceberg" and mistakenly assume that blending is "all you need" to address most deepfake challenges. However, our work reveals that the generalization capabilities of blending-based detection methods are limited, indicating that there is still significant room for improvement in the deepfake detection field.
> - **For frequency-based detectors:** We have implemented two typical frequency-based detection methods in deepfake detection: SPSL and SRM. We conducted all evaluations and analyses based on the two models. Inspired by your comment, we plan to extend a further analysis specifically for the frequency detectors. Thanks for your comment!
>
> *ref[1]: SBI, CVPR 2022;* *ref[2]: LSDA, CVPR 2024.*
>
>
> > line 141~143: When fine-tuning the generative model, how and for what purpose is it tuned?
>
> **Answer:** Thanks for your valuable mention. Actually, this is one of the key contributions of our DF40 dataset. We would like to clarify the following points:
> - ***Why* did we fine-tune the generative model of EFS?** The primary motivation is to create an **aligned data domain and fake methods** for evaluation. We have **two main reasons** for pursuing this:
>   - **Alignment with previous evaluation protocols:** Most previous deepfake detection studies have followed a widely-used protocol for selecting state-of-the-art detectors: training models on four deepfake methods from FF++ and testing them on one fake method from CDF. This led us to question whether we could expand this approach by implementing more fake methods on FF++ (beyond the original four) and CDF (testing on more than just one specific method). This consideration motivated us to choose FF++ and CDF as the real data sources for generating corresponding fake data.
>   - **Isolating the individual effects on detection results:** Simply following the protocol of training on FF++ and testing on CDF would not allow us to isolate the individual influences of data domains and fake methods on the final detection results. Additionally, we noticed that many previous studies have performed cross-manipulation evaluations—training on one specific method and testing on others within the same data domain (FF++). However, the limited diversity of fake methods in previous datasets restricted cross-manipulation evaluations to just 4-5 different methods, most of which were face-swapping techniques. This limitation motivated us to create a significantly more diverse dataset to enable more comprehensive cross-manipulation evaluations.
> - ***How* did we fine-tune the generative model of EFS?** Please note the fine-tuning details for each EFS method can vary. In Algorithm 3 of the supplementary, we provide a general understanding of the fine-tuning process. Following your suggestion, we will add the detailed process for each method in the revision.
>
> > line 249: What is meant by 'orthogonal'?
>
> **Answer:** Thanks for your mention. The 'orthogonal' in Line 249 means that the decision boundary of data domains and fake methods are orthogonal. In our case, from the t-SNE of Figure 7, we can see that  CLIP-large can "separate and align" different forgeries and data domains in the "WhichisReal" test data. This indicates that CLIP-large can learn informative and discriminative features to separate different data domains and fake methods without explicit supervision and constraints.
>
> > Why is FE not considered in the [4.2 Evaluation] section?
>
> **Answer:** Thank you for your kind mention. Directly using the original FE methods (without fine-tuning) in [4.2 Evaluation] could lead to unaligned and unfair evaluations since the other three types (FS, FR, and EFS) are aligned. Specifically:
> - Fine-tuning FE methods requires detailed and accurate facial annotations (such as age, race, and gender) for each image, which are not readily available.
> - Directly using FE can result in high-resolution images that are noticeably different from the real images in FF++. This discrepancy could introduce a resolution bias, as mentioned in my earlier Rebuttal.

---

### Official Review · Reviewer_nz7T · 2024-07-25
**A Large-Scale Deepfake Dataset with a Variety in Generation Techniques**

**Rating:** 4
**Confidence:** 5
**Correctness:** The construction of the dataset is so…
**Clarity:** The paper is well written.

**Review:**

Quality: The work is of high quality, demonstrating a thorough understanding of the current limitations in deepfake detection and addressing them with a well-constructed benchmark.

Clarity: The paper is well-written and clear.

Originality: The originality of this work lies in the creation of the DF40 dataset, which significantly expands the range of deepfake techniques included in a single benchmark. The introduction of 40 distinct deepfake methods, including recent and emerging techniques, showcases the innovative aspect of this work.

Significance: The significance of this work is substantial, as it provides a much-needed resource for the deepfake detection community. The DF40 dataset not only enhances the scope of research in this area but also sets a new standard for evaluating deepfake detection methods.

**Strengths:**

+ The paper is written well and is easy to understand.
+ The dataset includes 40 generation techniques which makes it comprehensive.
+ The paper discusses about the ethical and societal implications.
+ The analysis presented in the paper is interesting.

**Additional Feedback:**

If the authors address the concerns, I'll happy to improve my rating.

**Documentation:**

The details are available.

**Ethics:**

No such issues.

**Limitations:**

Yes.

**Opportunities For Improvement:**

- While the assertion made at line 39 is accurate, it should be supported with appropriate citations to strengthen the argument.
- Findings 1 and 2 would benefit from the inclusion of qualitative analysis to substantiate the claims presented. The qualitative analysis can be misleading.
- If Finding 5 is accurate, it may challenge the underlying rationale for the dataset’s creation. The authors must discuss this in detail.

**Relation To Prior Work:**

The discussion mentions how the work is different from previous contributions.

**Summary And Contributions:**

The paper introduces a novel comprehensive benchmark for deepfake detection. The authors propose a dataset which includes 40 distinct deepfake techniques, significantly surpassing existing datasets in diversity and scale. This dataset covers a broad range of deepfake methods, including face-swapping, face-reenactment, entire face synthesis, and face editing. The paper identifies limitations in current deepfake detection methods, such as the outdated and limited diversity of training datasets, which hinder the generalization of detection algorithms to real-world scenarios.

---

> ### Author Rebuttal · Authors · 2024-08-19
>
> We sincerely thank Reviewer nz7T for their thoughtful and constructive feedback, and for recognizing the substantial contributions to the deepfake detection community of our paper. Below, we address each of the reviewer's concerns in detail, point by point.
>
> ---
>
> > *While the assertion made at line 39 is accurate, it should be supported with appropriate citations to strengthen the argument.*
>
> **Answer:** Thanks for your suggestion. **We would like to enhance this claim with two references (ref[1] and ref[2]) in Line 39 of the manuscript.**
> - To illustrate, ref[1], which introduces a detector for localized face forgery detection, highlights its limitation in detecting whole-image synthesis, such as GAN-generated images, as shown by its 0.6911 AUC on StyleGAN. This limitation arises because ref[1] and similar techniques are designed to detect blending artifacts, **assuming that fake images involve a blending process** to differentiate between real and fake. However, in cases of **whole image synthesis, where no blending occurs**, these techniques fall short, even though the image remains fake. A similar claim can also be found in ref[2].
> - These two references support our argument in Line 39 of the manuscript.
>
> *ref[1]: SBI, CVPR 2022;* *ref[2]: LSDA, CVPR 2024.*
>
> ---
>
> > *Findings 1 and 2 would benefit from the inclusion of qualitative analysis to substantiate the claims presented. The qualitative analysis can be misleading.*
>
> **Answer:** Thanks for your comment. **We would like to provide further qualitative and quantitative analysis for *Finding-1* and *Finding-2*:**
> - **From *Finding-1*,** we observe an asymmetric performance drop among different forgery types (by Protocol-1). Specifically, model training on FS yields higher results (around 0.8) on EFS, while training on EFS only achieves about 0.6 on FS. To understand the underlying reason for this finding, we provide the following detailed explanations:
>   - **One possible explanation is that FS can exhibit both localized and global forgeries, while EFS is restricted to global forgeries, making EFS less diverse in scope compared to FS.** In other words, FS may include fake regions both within the facial area and across the entire image, whereas EFS only involves forgeries affecting the entire image. This makes the fake patterns in EFS potentially simpler than those in FS. As a result, training a model on EFS might not be sufficient to capture all forgery patterns, particularly the region-aware fake artifacts present in FS. For example, some localized FS (such as DeepFakes of FF++) only contain fake artifacts within the facial region, but a model trained on EFS might be biased to "expect" artifacts across the entire image, including the background, resulting in a lower result on FS.
>   - **To verify our above claim, we design and conduct **quantitative** and **qualitative** verification experiments:** ***(1) Quantitatively,*** we create the *self-reconstruction image* (SRI) for each original real image, where the SRI images act as fakes with a global forgery pattern (entire image synthesis). Then, we perform a blending operation to replace this reconstruction face with the original face, creating a localized fake (L-SRI), just like other typical localized face forgery methods. Here, we consider five typical types of fake regions: whole, face, mouth, nose, and eyes. Results in **Table 1 of the attached PDF** below quantitatively verify the *significance* of the fake region in detection.
> ***(2) Qualitatively,*** To provide further qualitative analysis, we consider using t-SNE for visualizing the latent distribution of 'real', 'whole-fake', and 'face-fake'. The t-SNE (**Figure 1 of the attached PDF**) results show that a well-trained EFS detection model can easily distinguish the whole-fake from real images. However, the localized version, including face-fake and mouth-fake, appears much more similar to real images, making it more challenging to detect compared to the whole-fake. This experiment qualitatively verifies the significance of the fake region in detection.
> - **For *Finding-2*,** our observation is that existing SoTA detectors, which generally achieve higher results on previous datasets like Celeb-DF compared to the baseline, **perform comparably or even slightly worse than the baseline** in our evaluation. In this study, we chose SoTA detectors such as SRM, SPSL, RECCE, and RFM for evaluation. These detectors all utilize Xception as their baseline or backbone but achieve results comparable to Xception (see Tables 3, 4, and 5 in our manuscript for details). Notably, Xception basically "beats" all these SoTAs when training on EFS (FF) and testing on other DF40 forgery variants (see **Table 2 of the attached PDF** for our summary).
>
> ---
>
>
> > If Finding 5 is accurate, it may challenge the underlying rationale for the dataset’s creation. The authors must discuss this in detail.
>
> **Answer:** Thanks. **We provide detailed discussion as follows:**
> - **On the one hand, Finding-5 is accurate** because we have indeed observed that: most FR forgery methods demonstrate high AUC scores in the heatmap (see Figure 3 of the manuscript).
> - **On the other hand, this does not mean that collecting FR methods is without value**, for two main reasons:
>     - **First**, we were able to achieve this finding thanks to our extensive collection of FR methods (we collected 12 methods) and the evaluation protocols we proposed. Without these efforts, this finding would not have been possible.
>     - **Second, more importantly,** training on different FR methods can lead to significantly different results when applied to other forgery types, as demonstrated in **Table 3 of the attached PDF**. This indicates that even though these FR methods may share some transferable patterns, the similarity of fake content with other types (like EFS) can vary greatly depending on the specific FR method used. So the collection of (many) FR methods can still be valuable.

---

> > ### Comment · Reviewer_nz7T · 2024-08-30
> >
> > I appreciate the authors’ efforts in addressing my initial concerns. However, after reviewing the feedback from other reviewers, several critical issues remain unresolved:
> >
> > - Bias Analysis: The authors have not provided an analysis to determine if the proposed dataset introduces any inherent biases. It is crucial to include statistics on facial attributes such as gender and skin tone to ensure fairness.
> >
> > - Balancing across different attributes: Without a balanced representation of these attributes, the models trained on this dataset are likely to exhibit biased behavior [1]. The authors must address this by providing detailed statistics and ensuring balance.
> >
> > - Environmental Impact: The environmental impact of the dataset, particularly in terms of emissions, has not been discussed. The authors should quantify and report the emissions associated with the dataset’s creation and usage.
> >
> > - Novelty, Contribution, and Longevity of the Dataset: The authors need to clarify the steps taken to ensure that this work is not merely another incremental addition to the existing body of deepfake datasets. A discussion on the unique contributions and advancements made by this dataset is necessary. Additionally, the authors should outline how this dataset can be updated and maintained to support future needs in recognizing deepfakes, especially those generated by new or unseen techniques.
> >
> > - Unimodal dataset: The proposed dataset is not multimodal and could potentially be replicated by curating multiple existing datasets. The authors should elaborate on how this dataset stands out in terms of its single modality and what unique value it brings to the field.

---

> ### Author Response · Authors · 2024-08-31
> **We appreciate our rebuttal has addressed your initial concerns, and here we provide additional comment for your newly raised concerns**
>
> We sincerely appreciate that our rebuttal has addressed your initial concerns, and we also appreciate the reviewer for recognizing our paper:
>
> - (1) A **novel comprehensive benchmark** for deepfake detection;
>
> - (2) DF40 **significantly** surpasses existing datasets in **diversity and scale**;
>
> - (3) **Interesting analysis** using this work;
>
> - (4) DF40 provides a **much-needed resource** for the deepfake detection community;
>
> - (5) DF40 not only enhances the scope of research in this area but also **sets a new standard** for evaluating deepfake detection methods.
>
> Below, we provide further clarification and a response to the newly raised concerns by the reviewer.
>
>
> ---
>
>
> > **Bias Analysis (concern-1) and Balancing across different attributes (concern-2).**
>
> Thanks for your valuable question. Our DF40's fake data is created based on existing publicly available datasets, aligning its demographic with these original data. Consequently, the fairness issues existing in their data are also reflected in our dataset.
>
> However, it is important to note that:
> - **(1) Using publicly available datasets for creating fakes is highly meaningful, though these datasets might be potentially biased:** Fake data of our DF40 dataset is created by using existing publicly available real datasets such as FF++ and CDF. The rationale behind this setting is to align with previous works. Specifically, most prior deepfake detection studies train their models on a limited number of fake types (four) from FF++ and test them on one fake type from CDF. This consideration led us to select FF++ and CDF as the real data sources to create a **much more diverse range of fake types for both training and testing.**
> - **(2) Training on a biased deepfake dataset could still yield an unbiased deepfake detector:** There are two scenarios for this:
>   - **a.** Models have been pre-trained on large-scale datasets. CLIP is a good example to illustrate, as evidenced in *Figure 1 of the attached PDF of the rebuttal for RsPZ*, where CLIP can clearly identify fake data with different races and genders. This suggests that CLIP does NOT consider race and gender as discriminative features for distinguishing real and fake, making it possibly unbiased;
>   - **b.** A conventional CNN such as PatchForensics which takes only the local patch as input, cannot "see" the global facial attributes. So training this model on a biased dataset might not be biased.
> - **(3) The main goal of our work is to create a generalizable deepfake detector, not a fairness model:** Fairness and generalization can be *two different concepts in machine learning*. A fairness model might not achieve generalizable results in deepfake detection tasks. In our work, we primarily focus on generalization and have considered several ways to aid in developing an unbiased deepfake detector (see the solution below).
>
> In line with the reviewer's suggestion, we have also contemplated **several solutions** to address the issue:
> - **(1) Add the annotations of facial attributes for each video:** This enables the development of more fair and unbiased deepfake detectors using our datasets. By doing so, we can obtain the facial attributes like race and gender of each video, thereby enabling us to add constraints for creating unbiased models;
> - **(2) Provide more analysis tools to confirm whether the detector has learned biased information:** We intend to extract facial attributes such as gender from each video and then conduct a **correlation analysis** to determine the relationship between this attribute and the probability of being a fake.
>
> ---
>
> > **Environmental Impact (concern-3).**
>
> Thanks. **We have checked all benchmarks and datasets in this field and did not find any calculations or estimates of carbon emissions**. We would like to estimate it roughly:
> - The calculation of carbon emissions usually involves energy consumption and the corresponding carbon emission coefficient. According to the official info of NVIDIA-V100 (`https://www.nvidia.cn/data-center/v100/`), the average power during operation is **300 watts**. The carbon emission coefficient is taken as 0.997 kilograms of carbon dioxide per kilowatt-hour. Then the carbon emissions of the V100 within one hour can be calculated by: 0.3 kilowatts × 1 hour × 0.997 kilograms of carbon dioxide per kilowatt-hour = **0.2991**.
>
> - **For the creation phase:** creating one fake image needs 1s and one video for 3min. Since we have 0.1M videos and 0.1M images (see Table 2 of the manuscript), the carbon emission can be estimated using the above formulation: **82.92** and **1495.5** kilograms of carbon dioxide for fake images and videos;
> - **For the training phase:** Training one model roughly takes one hour in one epoch. So training 10 models by 10 epochs needs 100 hours. Thus, the total carbon emission can be estimated as **299.1** kilograms of carbon dioxide.
>
> For each specific method, we will provide more detailed computations in the revision.

---

> > ### Comment · Reviewer_nz7T · 2024-09-01
> >
> > I appreciate the authors’ prompt response to my comments. After reviewing their reply, I remain concerned that the current structure of the dataset exacerbates significant ethical issues related to bias, as also highlighted by the ethics reviewer.
> >
> > The authors seem to suggest that training on any dataset with the goal of generalizability will inherently produce a fair detector. However, extensive research has shown that it is the developers’ responsibility to address all potential biases and ensure that the trained model is unbiased. This has been empirically demonstrated in the case of deepfake detectors, where models trained on existing datasets without careful consideration of fairness tend to exhibit biased predictions [1,2,3,4,5]. Such issues should not be left for the model to resolve autonomously; it is imperative that developers ensure models are trained on fair data to be reliable in real-world applications.
> >
> > This year, over 70 countries held elections in 2024 [6], leading to a significant increase in the dissemination of deepfakes across major social media platforms [7]. These countries include politicians from diverse racial, gender, and age groups. Relying on a biased deepfake detector to manage this surge and prevent widespread misinformation is untenable.
> >
> > Upon further reflection, I believe the proposed dataset is an incremental curation to existing deepfake datasets. Given its scale, it is likely to perpetuate biases in the deepfake detectors trained on it [1], which could then be deployed in real-world scenarios. To mitigate this concern, I strongly recommend that the authors balance the dataset across various subgroups, even if this results in a reduction in its overall size.
> >
> > In light of these considerations, I regret to inform you that I must lower my rating from 5 to 4.
> >
> > Reference:
> > [1] Trinh, L., & Liu, Y. (2021). An examination of fairness of ai models for deepfake detection, IJCAI 2021.
> > [2] Lin, L., He, X., Ju, Y., Wang, X., Ding, F., & Hu, S. Preserving fairness generalization in deepfake detection, CVPR 2024.
> > [3] Ju, Y., Hu, S., Jia, S., Chen, G. H., & Lyu, S. Improving fairness in deepfake detection, WACV 2024.
> > [4] Nadimpalli, A. V., & Rattani, A. GBDF: gender balanced deepfake dataset towards fair deepfake detection, ICPR 2022.
> > [5] Wang, T., Zhao, J., Yatskar, M., Chang, K. W., & Ordonez, V. Balanced datasets are not enough: Estimating and mitigating gender bias in deep image representations, ICCV 2019.
> > [6] https://edition.cnn.com/2024/07/08/world/global-elections-2024-maps-charts-dg/index.html.
> > [7] https://www.washingtonpost.com/technology/2024/04/23/ai-deepfake-election-2024-us-india/.

---

> ### Author Response · Authors · 2024-08-31
> **Additional response by Authors**
>
> Due to the limited space, we provide an additional response to handle the **remaining two concerns that reviewer nz7T has newly put forward.**
>
> ---
>
> > **Unique Contribution and Longevity of the Dataset (concern-4).**
>
> Thanks for your kind mention! We would like to clarify it from **two aspects**: **(1)** What is the unique contribution of our dataset over previous works? **(2)** How our dataset can be updated and maintained to support future deepfakes?
>
> **(1) Unique Contributions of Our Dataset Compared to Previous Works:**
> - **Contribution-1: Aligned data domains and fake methods.** DF40 provides alignment between fake methods and data domains. Most methods of our dataset are generated under the FF++ and CDF domains. This point provides two obvious advantages:
>   - **Advantage-1**: It aligns with previous evaluation settings, facilitating further extension evaluation and making it easier for others to follow existing research protocols.
>   - **Advantage-2**: It helps analyze the individual impacts of the data domain and fake method separately, leading to the discovery that they collaboratively contribute to the final detection results as shown in the causal graph in Figure 5 of the manuscript.
>
> - **Contribution-2: DF40 consists of 40 different deepfake techniques covering various fake types**, including representative and state-of-the-art methods such as 10 face-swapping methods, 13 face-reenactment methods, 12 entire face synthesis methods, and 5 face editing methods. This enables the detection of current state-of-the-art deepfakes and AIGCs.
>
> - **Contribution-3: Provision of Multiple Standard Evaluation Protocols and Settings.** Most existing detection works evaluate only one type, like face-swapping types. In contrast, our DF40 dataset allows for 4 standard protocols to be performed for more comprehensive evaluations to identify the universal deepfake model.
>
>
> **(2) How Our Dataset Can Be Updated and Maintained to Support Future Deepfakes:**
> - Recognizing the constantly evolving nature of deepfakes, we are committed to continuously updating our DF40 dataset by incorporating the latest deepfake methods. As the name indicates, currently it contains 40 different methods for creating fakes. In the future, we plan to expand its scope to DF40w10 or DF40w20 and so on. Each updated version will include a variety of new deepfake methods. Given that a static dataset is more susceptible to being exploited, it is crucial to keep our platform adaptive.
>
> ---
>
> > **About multimodal dataset (concern-5).**
>
> We appreciate the valuable questions. We would like to clarify it as follows:
>
> - **Our proposed standard evaluation protocols cannot be replicated by curating multiple existing datasets:** Our work proposes Protocol-1 (cross-manipulation evaluation) and Protocol-2 (cross-domain evaluation). These protocols are designed to ***explore the distinct impact of fake methods and data domains.*** To implement them, alignment between fake methods and data domains is essential. This means keeping one factor unchanged while varying the other to understand their individual contributions to the final results.
>   - In DF40, we have achieved this alignment among fake methods, enabling these evaluation protocols. Notably, we fine-tuned entire-face-synthesis (EFS) deepfake methods such as DDIM and DiT under FF++ and CDF to ensure this alignment. To our knowledge, ***we are the first to achieve this alignment between methods and domains.***
>
> - **Our dataset has already included the multimodal deepfake methods:** Our dataset already encompasses three multimodal fake methods: Wav2Lip, SadTalker, and HeyGen, which generate both fake video and audio. The audio and text modalities for their generation are sourced from the LRS3 dataset (refer to Lines 177-178 of the supplementary).
>   - In our experiments, we discovered that these methods mainly focus on manipulating the mouth region, unlike other face-swapping methods that focus on the entire face. This is due to the inherent nature of multimodal deepfake methods which center around manipulating expression and mouth movement, relating to the "talking head" research topic. We have provided an analysis of the impact of different fake regions (whole-fake, face-fake, mouth-fake) in **Table 1 and Figure 1 of the attached PDF.**

---

### Decision · Program_Chairs · 2024-09-26

**Decision:**

Accept (Poster)

**Comment:**

This paper proposes a new comprehensive benchmark that has the potential to revolutionize the current deepfake detection field. Although the paper received diverse comments, one reviewer initially provided a negative review and did not respond later. The Area Chair double-checked the rebuttal and agreed with the authors' responses. Hence, I recommend acceptance.